# PSD Representations for Effective Probability Models

**Alessandro Rudi**
Inria, École normale supérieure,
CNRS, PSL Research University, Paris, France
`alessandro.rudi@inria.fr`

**Carlo Ciliberto**
Department of Computer Science
University College London, London, UK
`c.ciliberto@ucl.ac.uk`

## Abstract

Finding a good way to model probability densities is key to probabilistic inference. An ideal model should be able to concisely approximate any probability while being also compatible with two main operations: multiplications of two models (product rule) and marginalization with respect to a subset of the random variables (sum rule). In this work, we show that a recently proposed class of positive semi-definite (PSD) models for non-negative functions is particularly suited to this end. In particular, we characterize both approximation and generalization capabilities of PSD models, showing that they enjoy strong theoretical guarantees. Moreover, we show that we can perform efficiently both sum and product rule in closed form via matrix operations, enjoying the same versatility of mixture models. Our results open the way to applications of PSD models to density estimation, decision theory and inference.

## 1 Introduction

Modeling probability distributions is a key task for many applications in machine learning [17, 5]. To this end, several strategies have been proposed in the literature, such as adopting mixture models (e.g. Gaussian mixtures) [5], exponential models [32], implicit generative models [11, 13] or kernel (conditional) mean embeddings [16]. An ideal probabilistic model should have two main features: $i$) efficiently perform key operations for probabilistic inference, such as sum rule (i.e. marginalization) and product rule [5] and, $ii$) concisely approximate a large class of probabilities. Finding models that satisfy these two conditions is challenging and current methods tend to tackle only one of the two. Exponential and implicit generative models have typically strong approximation properties (see e.g. [32, 29]) but cannot easily perform operations such as marginalization. On the contrary, mixture models are designed to efficient integrate and multiply probabilities, but tend to require a large number of components to approximate complicated distributions.

In principle, mixture models would offer an appealing strategy to model probability densities since they allow for efficient computations when performing key operations such as sum and product rule. However, these advantages in terms of computations, come as a disadvantage in terms of expressiveness: even though mixture models are universal approximators (namely they can approximate arbitrarily well any probability density), they require a significant number $n$ of observations and of components to do that. Indeed it is known that models that are non-negative mixtures of non-negative components lead to learning rates that suffer from the curse of dimensionality. For example, when approximating probabilities on $\mathbb{R}^d$, kernel density estimation (KDE) [21] with non-negative components has rates as slow as $n^{-2/(4+d)}$ (see, e.g. [38, page 100]), and cannot improve with the regularity (smoothness) of the target probability (see e.g. [9, Thm. 16.1] for impossibility results in the one dimensional case).

In the past decades, this limitation has been overcome by removing either the non-negativity of the weights of the mixture (leading to RBF networks [27][35]), or the non-negativity of the components

35th Conference on Neural Information Processing Systems (NeurIPS 2021).

| | Non-negative | Sum Rule | Product Rule | Concise Approximation | Optimal Learning | Efficient Sampling |
|---|---|---|---|---|---|---|
| **Linear Models** | ✗ | ✓ | ✓ | ✓ | ✓ | ✗ |
| **Mean Embeddings** | ✗ | ✓ | ✓ | ✓ | ✓ | ✗ |
| **Mixture Models** | ✓ | ✓ | ✓ | ✗ | ✗ | ✓ |
| **Exponential Models** | ✓ | ✗ | ✓ | ✓ | ✓ | ✗ |
| **PSD Models** | ✓ (see [14]) | ✓ (Prop. 1) | ✓ (Prop. 2) | ✓ (Thm. 6) | ✓ (Thm. 7) | ✓ (see [15]) |

Table 1: Summary of the main desirable properties for a probability model.

used in the mixture (leading to KDE with oscillating kernel [35]), or both. On the positive side this allows to achieve a learning rate of $n^{-\beta/(2\beta+d)}$, for $\beta$-times differentiable densities on $\mathbb{R}^d$. Note that such rate is *minimax optimal* [10] and overcomes the curse of dimensionality when $\beta \geq d$. Additionally the resulting model is very *concise*, since only $m = O(n^{d/\beta})$ centers are necessary to achieve the optimal rate [36]. However, on the negative side, *the resulting model is not a probability*, since it may attain negative values, and so it cannot be used where a proper probability is required.

In this paper, we show that the positive semidefinite (PSD) models, recently proposed in [14], offer a way out of this dichotomy. By construction PSD models generalize mixture models by allowing also for negative weights, while still guaranteeing that the resulting function is non-negative everywhere. Here, we prove that they get the best of both worlds: expressivity (optimal learning and approximation rates with concise models) and flexibility (exact and efficient sum and product rule).

**Contributions.** The main contributions of this paper are:

*i)* Showing that PSD models can perform exact sum and product rules in terms of efficient matrix operations (Sec. 2.1).

*ii)* Characterizing the approximation and learning properties of PSD models with respect to a large family of probability densities (Sec. 3).

*iii)* Providing a "compression" method to control the number of components of a PSD model (introduced in Sec. 2.2 and analyzed in Sec. 3.3).

*iv)* Discussing a number of possible applications of PSD models (Sec. 4).

**Summary of the results.** Table 1 summarizes all the desirable properties that a probability model should have and which of them are satisfied by state-of-the-art estimators. Among these we have: *non-negativity*, the model should be point-wise non-negative; *sum and product rule*, the model should allow for efficient computation of such operations between probabilities; *concise approximation*, a model with the optimal number of components $m = O(\varepsilon^{-d/\beta})$ is enough to approximate with error $\varepsilon$ a non-negative function in a wide family of smooth $\beta$-times differentiable functions [36]; *optimal learning*, the model achieves optimal learning rates of order $n^{-\beta/(2\beta+d)}$ when learning the unknown target probability from i.i.d. samples [35]; *efficient sampling* the model allows to extract i.i.d. samples without incurring in the curse of dimensionality in terms of computational complexity. We note that for PSD models, this last property has been recently investigated in [15]. We consider under the umbrella of *linear models* all the models which corresponds to a mixture $\sum_{j=1}^{m} w_j f_j(x)$ of functions $f_j : \mathcal{X} \to \mathbb{R}$, with either $w_j \geq 0, \ \forall j = 1, \ldots, m$ or $f_j \geq 0, \ \forall j = 1, \ldots, m$ or both. We denote by *mixture models* the mixture models defined as above, where $w_j \geq 0, f_j \geq 0, \ \forall j = 1, \ldots, m$. By *mean embeddings*, we denote the kernel mean embedding estimator framework from [30] see also [16]). With *exponential models* we refer to models of the form $\exp(\sum_{j=1}^{m} w_j f_j(x))$ with $w_j \in \mathbb{R}, f_j : X \to \mathbb{R}$ [32]. With *PSD models* we refer to the framework introduced in [14] and studied in this paper. We note that this work contributes in showing that the latter are the only probability models to satisfy all requirements (we recall that non-negativity and efficient sampling have been shown respectively in [14] and [15]).

**Notation.** We denote by $\mathbb{R}^d_{++}$ the space vectors in $\mathbb{R}^d$ with positive entries, $\mathbb{R}^{n \times d}$ the space of $n \times d$ matrices, $\mathbb{S}^n_+ = \mathbb{S}_+(\mathbb{R}^n)$ the space of positive semidefinite $n \times n$ matrices. Given a vector $\eta \in \mathbb{R}^d$, we denote $\text{diag}(\eta) \in \mathbb{R}^{d \times d}$ the diagonal matrix associated to $\eta$. We denote by $A \circ B$

and $A \otimes B$ respectively the entry-wise and Kronecker product between two matrices $A$ and $B$. We denote by $\|A\|_F, \|A\|, \det(A), \text{vec}(A)$ and $A^\top$ respectively the Frobenius norm, the operator norm (i.e. maximum singular value), the determinant, the (column-wise) vectorization of a matrix and the (conjugate) transpose of $A$. With some abuse of notation, where clear from context we write element-wise products and division of vectors $u, v \in \mathbb{R}^d$ as $uv, u/v$. Given two matrices $X \in \mathbb{R}^{n \times d_1}, Y \in \mathbb{R}^{n \times d_2}$ with same number of rows, we denote by $[X, Y] \in \mathbb{R}^{n \times (d_1 + d_2)}$ their concatenation row-wise. The term $\mathbf{1}_n \in \mathbb{R}^n$ denotes the vector with all entries equal to 1.

## 2 PSD Models

Following [14], in this work we consider the family of positive semi-definite (PSD) models, namely non-negative functions parametrized by a feature map $\phi : \mathcal{X} \to \mathcal{H}$ from an input space $\mathcal{X}$ to a suitable feature space $\mathcal{H}$ (a separable Hilbert space e.g. $\mathbb{R}^q$) and a linear operator $\mathsf{M} \in \mathbb{S}_+(\mathcal{H})$, of the form

$$f(x \,; \mathsf{M}, \phi) = \phi(x)^\top \mathsf{M}\, \phi(x). \tag{1}$$

PSD models offer a general way to parametrize non-negative functions (since $\mathsf{M}$ is positive semidefinite, $f(x \,; \mathsf{M}, \phi) \geq 0$ for any $x \in \mathcal{X}$) and enjoy several additional appealing properties discussed in the following. In this work, we will focus on a special family of models of the form (1) to parametrize probability densities over $\mathcal{X} = \mathbb{R}^d$. In particular, we will consider the case where: $i)$ $\phi = \phi_\eta : \mathbb{R}^d \to \mathcal{H}_\eta$ is a feature map associated to the Gaussian kernel [26] $k_\eta(x, x') = \phi_\eta(x)^\top \phi_\eta(x) = e^{-(x-x')^\top \text{diag}(\eta)(x-x')}$, with $\eta \in \mathbb{R}^d_{++}$ and, $ii)$ the operator $\mathsf{M}$ lives in the span of $\phi(x_1), \ldots, \phi(x_n)$ for a given set of points $(x_i)_{i=1}^n$, namely there exists $A \in \mathbb{S}^n_+$ such that $\mathsf{M} = \sum_{ij} A_{ij} \phi(x_i) \phi(x_j)^\top$. We define a *Gaussian PSD model* by specializing the definition in (1) as

$$f(x \,; A, X, \eta) = \sum_{i,j=1}^n A_{i,j} k_\eta(x_i, x) k_\eta(x_j, x), \qquad \forall \, x \in \mathbb{R}^d \tag{2}$$

in terms of the coefficient matrix $A \in \mathbb{S}^n_+$, the base points matrix $X \in \mathbb{R}^{n \times d}$, whose $i$-th row corresponds to the point $x_i$ for each $i = 1, \ldots, n$ and kernel parameter $\eta$. In the following, given two base point matrices $X \in \mathbb{R}^{n \times d}$ and $X' \in \mathbb{R}^{m \times d}$, we denote by $K_{X,X',\eta} \in \mathbb{R}^{n \times m}$ the kernel matrix with entries $(K_{X,X',\eta})_{ij} = k_\eta(x_i, x'_j)$ where $x_i, x'_j$ are the $i$-th and $j$-th rows of $X, X'$ respectively. When clear from context, in the following we will refer to Gaussian PSD models as PSD models.

**Remark 1** (PSD models generalize Mixture models). *Mixture models (a mixture of Gaussian distributions) are a special case of PSD models. Let $A = \text{diag}(a)$ be a diagonal matrix of $n$ positive weights $a \in \mathbb{R}^n_{++}$. We have $f(x \,; A, X, \eta/2) = \sum_{i=1}^n A_{ii} k_{\eta/2}(x_i, x)^2 = \sum_{i=1}^n a_i k_\eta(x_i, x)$.*

**Remark 2** (PSD models allow negative weights). *From (2), we immediately see that PSD models generalize mixture models by allowing also for negative weights: e.g., $f(\cdot \,; A, X, \eta)$ with $A = (1, -\frac{1}{2}; -\frac{1}{2}, \frac{1}{4}) \in \mathbb{S}^2_+, \eta = 1, X = (x_1; x_2)$ and $x_1 = 0, x_2 = 2$, corresponding to $f(x \,; A, X, \eta) = e^{-2x^2} + \frac{1}{4} e^{-2(x-2)^2} - \frac{1}{e} e^{-2(x-1)^2}$, i.e. a mixture of Gaussians with also negative weights.*

### 2.1 Operations with PSD models

In Sec. 3 we will show that PSD models can approximate a wide class of probability densities, significantly outperforming mixture models. Here we show that this improvement does not come at the expenses of computations. In particular, we show that PSD models enjoy the same flexibility of mixture models: $i)$ they are closed with respect to key operations such as marginalization and multiplication and $ii)$ these operations can be performed efficiently in terms of matrix sums/products. The derivation of the results reported in the following is provided in Appendix F. They follow from well-known properties of the Gaussian function.

**Evaluation.** Evaluating a PSD model in a point $x_0 \in \mathcal{X}$ corresponds to $f(x = x_0 \,; A, X, \eta) = K_{X,x_0,\eta}^\top A K_{X,x_0,\eta}$. Moreover, partially evaluating a PSD in one variable yields

$$f(x, y = y_0 \,; A, [X, Y], (\eta_1, \eta_2)) = f(x \,; B, X, \eta_1) \quad \text{with} \quad B = A \circ (K_{Y,y_0,\eta_2} K_{Y,y_0,\eta_2}^\top). \tag{3}$$

Note that $f(x \,; B, X, \eta_1)$ is still a PSD model since $B$ is positive semidefinite.

**Sum Rule (Marginalization and Integration).** The integral of a PSD model can be computed as

$$\int f(x\,;\,A,X,\eta)\,dx = c_{2\eta}\,\mathrm{Tr}(A\,K_{X,X,\frac{\eta}{2}}) \qquad \text{with} \qquad c_\eta = \int k_\eta(0,x)\,dx, \qquad (4)$$

where $c_\eta = \pi^{d/2}\det(\mathrm{diag}(\eta))^{-1/2}$. This is particularly useful to model probabiliy densities with PSD models. Let $Z = \int f(x\,;\,A,X,\eta)dx$, then the function $f(x\,;\,A/Z,X,\eta) = \frac{1}{Z}f(x\,;\,A,X,\eta)$ is a probability density. Integrating only one variable of a PSD model we obtain the sum rule.

**Proposition 1** (Sum Rule – Marginalization). *Let* $X \in \mathbb{R}^{n\times d}$, $Y \in \mathbb{R}^{n\times d'}$, $A \in \mathbb{S}_+(\mathbb{R}^n)$ *and* $\eta \in \mathbb{R}^d_{++}, \eta' \in \mathbb{R}^{d'}_{++}$. *Then, the following integral is a PSD model*

$$\int f(x,y\,;\,A,[X,Y],(\eta,\eta'))\,dx = f(y\,;\,B,Y,\eta'), \qquad \text{with} \qquad B = c_{2\eta}\,A \circ K_{X,X,\frac{\eta}{2}}, \quad (5)$$

The result above shows that we can efficiently marginalize a PSD model with respect to one variable by means of an entry-wise multiplication between two matrices.

**Remark 3** (Integration and marginalization on the hypercube). *The integrals in* (4) *and* (5) *can be performed also on when* $\mathcal{X}$ *is a hypercube* $H = \prod_{t=1}^{d}[a_t, b_t]$ *rather than the entire space* $\mathbb{R}^d$. *This leads to a closed form, where the matrix* $K_{X,X,\frac{\eta}{2}}$ *is replaced by a suitable* $K_{X,X,\frac{\eta}{2},H}$ *that can be computed with same number of operations (the full form of such matrix is reported in* Appendix F*).*

**Product Rule (Multiplication).** Multiplying two probabilities is key to several applications in probabilistic inference [5]. The family of PSD models is closed with respect to this operation.

**Proposition 2** (Multiplication). *Let* $X \in \mathbb{R}^{n\times d_1}$, $Y \in \mathbb{R}^{n\times d_2}$, $Y' \in \mathbb{R}^{m\times d_2}$, $Z \in \mathbb{R}^{m\times d_3}$, $A \in \mathbb{S}^n_+$, $B \in \mathbb{S}^m_+$ *and* $\eta_1 \in \mathbb{R}^{d_1}_{++}$, $\eta_2, \eta_2' \in \mathbb{R}^{d_2}_{++}$, $\eta_3 \in \mathbb{R}^{d_3}_{++}$. *Then*

$$f(x,y\,;\,A,[X,Y],(\eta_1,\eta_2))f(y,z\,;\,B,[Y',Z],(\eta_2',\eta_3)) = f(x,y,z\,;\,C,W,\eta), \qquad (6)$$

*is a PSD model, where* $C = (A \otimes B) \circ \big(\mathrm{vec}(K_{Y,Y',\tilde{\eta}_2})\mathrm{vec}(K_{Y,Y',\tilde{\eta}_2})^\top\big)$, *with* $\tilde{\eta}_2 = \frac{\eta_2\eta_2'}{\eta_2+\eta_2'}$, *base matrix* $W = [X \otimes \mathbf{1}_m,\ \frac{\eta_2}{\eta_2+\eta_2'}Y \otimes \mathbf{1}_m + \frac{\eta_2'}{\eta_2+\eta_2'}\mathbf{1}_n \otimes Y',\ \mathbf{1}_n \otimes Z]$ *and* $\eta = \big(\eta_1, \eta_2 + \eta_2', \eta_3\big)$.

We note that, despite the heavy notation, multiplying two PSD is performed via simple operations such as tensor and entry-wise product between matrices. In particular, we note that $X \otimes \mathbf{1}_m \in \mathbb{R}^{nm\times d_1}$ and $\mathbf{1}_n \otimes Z \in \mathbb{R}^{nm\times d_3}$ correspond respectively to the $nm \times d_1$-matrix containing $m$ copies of each row of $X$ and the $nm \times d_3$-matrix containing $n$ copies of $Z$. Finally, $\eta Y \otimes \mathbf{1}_m + \eta'\mathbf{1}_n \otimes Y$ is the $nm \times d_2$ base point matrix containing all possible sums $(\eta y_i + \eta' y_j')_{i,j=1}^{n,m}$ of points from $Y$ and $Y'$.

**Reduction.** As observed above, when performing operations with PSD models, the resulting base point matrix might be of the form $X \otimes \mathbf{1}_m$ (e.g. if we couple the multiplication in (6) with marginalization as in a Markov transition, see the corollary below). In these cases we can reduce the PSD model to have only $n$ base points (rather than $nm$), as follows

$$f(x\,;\,A,X \otimes \mathbf{1}_m,\eta) = f(x\,;\,B,X,\eta) \qquad \text{with} \qquad B = (I_n \otimes \mathbf{1}_m^\top)A(I_n \otimes \mathbf{1}_m), \qquad (7)$$

where $A \in \mathbb{S}^{nm}_+$ and $I_n \in \mathbb{R}^{n\times n}$ is the $n \times n$ identity matrix. The reduction operation is useful to avoid the dimensionality of the PSD model grow unnecessarily. This is for instance the case of a Markov transition.

**Corollary 3** (Markov Transition). *Let* $X \in \mathbb{R}^{n\times d_1}$, $Y \in \mathbb{R}^{n\times d_2}$, $Y' \in \mathbb{R}^{m\times d_2}$, $A \in \mathbb{S}^n_+$, $B \in \mathbb{S}^m_+$ *and* $\eta_1 \in \mathbb{R}^{d_1}_{++}$, $\eta_2, \eta_2' \in \mathbb{R}^{d_2}_{++}$. *Then*

$$\int f(x,y\,;\,A,[X,Y],(\eta_1,\eta_2))f(y\,;\,B,Y',\eta_2')\,dy = f(x\,;\,C,X,\eta_1), \qquad (8)$$

*with* $C \in \mathbb{S}^n_+$ *obtained by applying in order,* Prop. 2, Prop. 1 *and reduction* (7).

We remark that the result of a Markov transition retains the same base point matrix $X$ and parameters $\eta$ of the transition kernel. This is thanks to the reduction operation in (7), which avoids the resulting matrix $C$ to be $nm \times nm$. This fact is particularly useful in applications that require multiple Markov transitions, such as in hidden Markov models (see also Sec. 4).

## 2.2 Compression of a PSD model

From Prop. 2 we note that the multiplication operation can rapidly yield a large number of base points and thus incur in high computational complexity in some settings. It might therefore be useful to have a method to reduce the number of base points while retaining essentially the same model. To this purpose, here we propose a dimensionality reduction strategy. In particular, given a set of points $\tilde{x}_1, \ldots, \tilde{x}_m \in \mathbb{R}^d$, we leverage the representation of the PSD model in terms of reproducing kernel Hilbert spaces [3] to use powerful sketching techniques as *Nyström projection* (see, e.g., [40]), to project the PSD model on a new PSD model now based only on the new points. Given $A \in \mathbb{S}^n_+$, $X \in \mathbb{R}^{n \times d}$, $\eta \in \mathbb{R}^d_{++}$. Let $\tilde{X} \in \mathbb{R}^{m \times d}$ be the base point matrix whose $j$-rows corresponds to the point $\tilde{x}_j$. The *compression* of $f(\cdot\,;\,A, X, \eta)$ corresponds to

$$f(x\,;\,\tilde{A}, \tilde{X}, \eta) \qquad \text{with} \qquad \tilde{A} = BAB^\top \in \mathbb{S}^m_+, \quad B = K^{-1}_{\tilde{X}, \tilde{X}, \eta} K_{\tilde{X}, X, \eta} \in \mathbb{R}^{m \times n}, \qquad (9)$$

which it is still a PSD model since the matrix $BAB^\top \in \mathbb{S}^m_+$. We not that even with a rather simple strategy to choose the new base points $\tilde{x}_1, \ldots, \tilde{x}_m$ – such as uniform sampling – compression is an effective tool to reduce the computational complexity of costly operations. In particular, in Sec. 3.3 we show that a compressed model with only $O(t \operatorname{polylog}(1/\varepsilon))$ centers (instead of $t^2$) can $\varepsilon$-approximate the product of two PSD models with $t$ points each.

## 3 Representation power of PSD models

In this section we study the theoretical properties of Gaussian PSD models. We start by showing that they admit concise approximations of the target density (in the sense discussed in the introduction to this paper and of Table 1). We then proceed to studying the setting in which we aim to learn an unknown probability from i.i.d. samples. We conclude the section by characterizing the approximation properties of the compression operation introduced in Sec. 2.2.

### 3.1 Approximation properties of Gaussian PSD models

We start the section recalling that Gaussian PSD models are universal approximators for probability densities. In particular, the following result restates [Thm. 2 14] for the case of probabilities.

**Proposition 4** (Universal consistency – Thm. 2 in [14]). *The Gaussian PSD family is a universal approximator for probabilities that admit a density.*

The result above is not surprising since Gaussian PSD models generalize classical Gaussian mixtures (see Remark 1), which are known to be universal [5]. We now introduce a mild assumption that will enable us to complement Prop. 4 with approximation and learning results. In the rest of the section we assume that $\mathcal{X} = (-1, 1)^d$ (or more generally an open bounded subset of $\mathbb{R}^d$ with Lipschitz boundary). Here $L^\infty(\mathcal{X})$ and $L^2(\mathcal{X})$ denote respectively the space of essentially bounded and square-integrable functions over $\mathcal{X}$, while $W_2^\beta(\mathcal{X})$ denotes the *Sobolev space* of functions whose weak derivatives up to order $\beta$ are square-integrable on $\mathcal{X}$ (see [1] or Appendix A for more details).

**Assumption 1.** *Let $\beta > 0, q \in \mathbb{N}$. There exists $f_1, \ldots, f_q \in W_2^\beta(\mathcal{X}) \cap L^\infty(\mathcal{X})$, such that the density $p : \mathcal{X} \to \mathbb{R}$ satisfies*

$$p(x) = \sum_{j=1}^q f_j(x)^2, \qquad \forall x \in \mathcal{X}. \qquad (10)$$

The assumption above is quite general and satisfied by a wide family of probabilities, as discussed in the following proposition. The proof is reported in Appendix D.1

**Proposition 5** (Generality of Assumption 1). *The assumption above is satisfied by*

*(a) any probability density $p \in W_2^\beta(\mathcal{X}) \cap L^\infty(\mathcal{X})$ and strictly positive on $[-1, 1]^d$,*

*(b) any exponential model $p(x) = e^{-v(x)}$ with $v \in W_2^\beta(\mathcal{X}) \cap L^\infty(\mathcal{X})$,*

*(c) any mixture model of Gaussians or, more generally, of exponential models from (b),*

*(d) any $p$ that is $\beta + 2$-times differentiable on $[-1, 1]^d$, with a finite set of zeros, all in $(-1, 1)^d$, and with positive definite Hessian in each zero. E.g. $p(x) \propto x^2 e^{-x^2}$.*

*Moreover if $p$ is $\beta$-times differentiable over $[-1,1]^d$, then it belongs to $W_2^\beta(\mathcal{X}) \cap L^\infty(\mathcal{X})$.*

We note that in principle the class $\mathcal{C}_{\beta,d} = W_2^\beta(\mathcal{X}) \cap L^\infty(\mathcal{X})$ is larger than the Nikolskii class usually considered in density estimation [10] when $\beta \in \mathbb{N}$, but Assumption 1 imposes a restriction on it. However, Prop. 5 shows that the restriction imposed by Assumption 1 is quite mild, since it the resulting space includes all probabilities that are $\beta$-times differentiable and strictly positive (it also contains probabilities that are not strictly positive but have some zeros, see [24]). We can now proceed to the main result of this work, which characterizes the approximation capabilities of PSD models.

**Theorem 6** (Conciseness of PSD Approximation). *Let $p$ satisfy Assumption 1. Let $\varepsilon > 0$. There exists a Gaussian PSD model of dimension $m \in \mathbb{N}$, i.e., $\hat{p}_m(x) = f(x\,;\, A_m, X_m, \eta_m)$, with $A_m \in \mathbb{S}_+^m$ and $X_m \in \mathbb{R}^{m \times d}$ and $\eta_m \in \mathbb{R}_{++}^d$, such that*

$$\|p - \hat{p}_m\|_{L^2(\mathcal{X})} \leq \varepsilon, \quad \text{with} \quad m = O(\varepsilon^{-d/\beta}(\log \tfrac{1}{\varepsilon})^{d/2}). \tag{11}$$

The proof of Thm. 6 is reported in Appendix D.3. According to the result, the number of base points needed for a PSD model to approximate a density up to precision $\varepsilon$ depends on its smoothness (the smoother the better) and matches the bound $m = \varepsilon^{-d/\beta}$ that is optimal for function interpolation [18], corresponding to models allowing for negative weights, and is also optimal for convex combinations of oscillating kernels [36].

## 3.2 Learning a density with PSD models

In this section we study the capabilities of PSD models to estimate a density from $n$ samples. Let $\mathcal{X} = (-1, 1)^d$ and let $p$ be a probability on $\mathcal{X}$. Denote by $x_1, \ldots, x_n$ the samples independently and identically distributed according to $p$, with $n \in \mathbb{N}$. We consider a Gaussian PSD estimator $\hat{p}_{n,m} = f(x\,;\, \hat{A}, \tilde{X}, \eta)$ that is built on top of $m$ additional points $\tilde{x}_1, \ldots, \tilde{x}_m$, sampled independenly and uniformly at random in $\mathcal{X}$. In particular, $\eta \in \mathbb{R}_{++}^d$, $\tilde{X} \in \mathbb{R}^{m \times d}$ is the base point matrix whose $j$-th row corresponds to the point $\tilde{x}_j$ and $\hat{A} \in \mathbb{S}_+^m$ is trained as follows

$$\hat{A} = \underset{A \in \mathbb{S}_+^m}{\operatorname{argmin}} \int_{\mathcal{X}} f(x\,;\, A, \tilde{X}, \eta)^2 dx - \frac{2}{n}\sum_{i=1}^n f(x_i\,;\, A, \tilde{X}, \eta) \,+\, \lambda\|K^{1/2}AK^{1/2}\|_F^2, \tag{12}$$

where $K = K_{\tilde{X}, \tilde{X}, \eta}$. Note that the functional is constituted by two parts. The first two elements are an empirical version of $\|f - p\|_{L^2(\mathcal{X})}^2$ modulo a constant independent of $f$ (and so not affecting the optimization problem), since $x_i$ are identically distributed according to $p$ and so $\frac{1}{n}\sum_{i=1}^n f(x_i) \approx \int f(x)p(x)dx$. The last term is a regularizer and corresponds to $\|K^{1/2}AK^{1/2}\|_F^2 = \operatorname{Tr}(AKAK)$, i.e. the Frobenius norm of $\mathsf{M} = \sum_{ij=1}^m A_{ij}\phi_\eta(\tilde{x}_i)\phi_\eta(\tilde{x}_j)$. The problem in (12) corresponds to a quadratic problem with a semidefinite constraint and can be solved using techniques such as Newton method [24] or first order dual methods [14]. We are now ready to state our result.

**Theorem 7.** *Let $n, m \in \mathbb{N}, \lambda > 0, \eta \in \mathbb{R}_{++}^d$ and $p$ be a density satisfying Assumption 1. With the definitions above, let $\hat{p}_{n,m}$ be the model $\hat{p}_{n,m}(x) = f(x\,;\, \hat{A}, \tilde{X}, \eta)$, with $\hat{A}$ the minimizer of (12). Let $\eta = n^{\frac{2}{2\beta+d}} \mathbf{1}_d$ and $\lambda = n^{-\frac{2\beta+2d}{2\beta+d}}$. When $m \geq C'n^{\frac{d}{2\beta+d}}(\log n)^d \log(C''n(\log n))$, the following holds with probability at least $1 - \delta$,*

$$\|p - \hat{p}_{n,m}\|_{L^2(\mathcal{X})} \leq Cn^{-\frac{\beta}{2\beta+d}}(\log n)^{d/2}, \tag{13}$$

*where constant $C$ depends only on $\beta, d$ and $p$ and the constants $C', C''$ depend only on $\beta, d$.*

The proof of Thm. 7 is reported in Appendix E.2. The theorem guarantees that under Assumption 1, Gaussian PSD models can achieve the rate $O(n^{-\beta/(2\beta+d)})$ – that is optimal for the $\beta$-times differentiable densities – while admitting a concise representation. Indeed, it needs a number $m = O(n^{d/(2\beta+d)})$ of base points, matching the optimal rate in [36]. When $\beta \geq d$, a model with $m = O(n^{1/3})$ centers achieves optimal learning rates.

## 3.3 The Effect of compression

We have seen in the previous section that Gaussian PSD models achieve the optimal learning rates, with concise models. However, we have seen in the operations section that multiplying two PSD

models of $m$ centers leads to a PSD model with $m^2$ centers. Here we study the effect of compression, to show that it is possible to obtain an $\varepsilon$-approximation of the product via a compressed model with $O(m \operatorname{polylog}(1/\varepsilon))$ centers. In the following theorem we analyze the effect in terms of the $L^\infty$ distance on a domain $[-1,1]^d$, induced by the compression, when using points taken independently and uniformly at random from the same domain.

Let $A \in \mathbb{S}_+^n$, $X \in \mathbb{R}^{n \times d}$, $\eta \in \mathbb{R}_{++}^d$, we want to study the compressibility of the PSD model $p(x) = f(x\,;\,A, X, \eta)$. Let $\tilde{X} \in \mathbb{R}^{m \times d}$ be the base point matrix whose $j$-rows corresponds to the point $\tilde{x}_j$ with $\tilde{x}_1, \ldots, \tilde{x}_m$ be sampled independently and uniformly at random from $[-1,1]^d$. Denote by $\tilde{p}_m(x)$ the PSD model $\tilde{p}_m(x) = f(x\,;\,\tilde{A}, \tilde{X}, \eta)$ where $\tilde{A}$ is the compression of $A$ via (9). We have the following theorem.

**Theorem 8** (Compression of Gaussian PSD models). *Let $\delta \in (0, 1]$, $\eta_+ = \max(1, \max_{i=1,\ldots,d} \eta_i)$. When $m$ satisfies*

$$m \geq O\left( \left( \eta_+^{1/2} \log \frac{\|A\| n}{\varepsilon} \right)^d \log \frac{1}{\delta} \right), \tag{14}$$

*then the following holds with probability at least $1 - \delta$,*

$$|p(x) - \tilde{p}_m(x)| \leq \varepsilon^2 + \varepsilon \sqrt{p(x)}, \qquad \forall x \in [-1,1]^d, \tag{15}$$

The proof of the theorem above is in Appendix C.1. To understand its relevance, let $\hat{p}_1$ be a PSD model trained via (12) on $n$ points sampled from $p_1$ and $\hat{p}_2$ trained from $n$ points sampled from $p_2$, where both $p_1, p_2$ satisfy Assumption 1 for the same $\beta$ and $m, \lambda, \eta$ are chosen as Thm. 7, in particular $m = n^{d/(2\beta+d)}$ and $\eta = \eta_+ \mathbf{1}_d$, $\eta_+ = n^{2/(2\beta+d)}$. Consider the model $\hat{p} = \hat{p}_1 \cdot \hat{p}_2$. By construction $\hat{p}$ has $m^2 = n^{2d/(2\beta+d)}$ centers, since it is the pointwise product of $\hat{p}_1, \hat{p}_2$ (see Prop. 2) and approximates $p_1 \cdot p_2$ with error $\varepsilon = n^{-\beta/(2\beta+d)} \operatorname{polylog}(n)$, since both $\hat{p}_1, \hat{p}_2$ are $\varepsilon$-approximators of $p_1, p_2$. Instead, by compressing $\hat{p}$, we obtain an estimator $\bar{p}$, that according to Thm. 8, achieves error $\varepsilon$ with a number of center

$$m' = O(\eta_+^{d/2} \operatorname{polylog}(1/\varepsilon)) = O(n^{d/(2\beta+d)} \operatorname{polylog}(1/\varepsilon)) = O(m \operatorname{polylog}(1/\varepsilon)). \tag{16}$$

Then $\bar{p}$ approximates $p_1 \cdot p_2$ at the optimal rate $n^{-\beta/(2\beta+d)}$, but with a number of centers $m'$ that is only $O(m \operatorname{polylog}(n))$, instead of $m^2$. This means that $\bar{p}$ is essentially as good as if we learned it from $n$ samples taken directly from $p_1 \cdot p_2$. This renders compression a suitable method to reduce the computational complexity of costly inference operations as the product rule.

## 4   Applications

PSD models are a strong candidate in a variety of probabilistic settings. On the one hand, they are computationally amenable to performing key operations such as sum and product rules, similarly to mixture models (Sec. 2.1). On the other hand, they are remarkably flexible and can approximate/learn (coincisely) a wide family of target probability densities (Sec. 3). Building on these properties, in this section we consider different possible applications of PSD models in practice.

### 4.1   PSD Models for Decision Theory

Decision theory problems (see e.g. [5] and references therein) can be formulated as a minimization

$$\theta_* = \operatorname*{argmin}_{\theta \in \Theta} L(\theta) = \mathbb{E}_{x \sim p}\, \ell(\theta, x), \tag{17}$$

where $\ell$ is a loss function, $\Theta$ is the space of target parameters (decisions) and $p$ is the underlying data distribution. When we can sample directly from $p$ – e.g. in supervised or unsupervised learning settings – we can apply methods such as stochastic gradient descent to efficently solve (17). However, in many applications, sampling from $p$ is challenging or computationally unfeasible. This is for instance the case when $p$ has been obtained via inference (e.g. it is the $t$-th estimate in a hidden Markov model, see Sec. 4.3) or it is fully known but has a highly complex form (e.g. the dynamics of a physical system). In contexts where sampling cannot be performed efficiently, it is advisable to consider alternative approaches. Here we propose a strategy to tackle (17) when $p$ can be modeled (or well-approximated) by a PSD model. Our method hinges on the following result.

**Proposition 9.** *Let $p(x) = f(x\,;\,A, X, \eta)$ with $X \in \mathbb{R}^{n \times d}$, $A \in \mathbb{S}_+^n$, $\eta \in \mathbb{R}_{++}^d$. Let $g : \mathbb{R}^d \to \mathbb{R}$ and define $c_{g,\eta}(z) = \int g(x) e^{-\eta \|x-z\|^2} \, dx$ for any $z \in \mathbb{R}^d$. Then*

$$\mathbb{E}_{x \sim p} \, g(x) \;=\; \mathrm{Tr}\big( (A \circ K_{X,X,\eta/2}) \, G \big) \qquad with \qquad G_{ij} = c_{g,2\eta}\big(\tfrac{x_i + x_j}{2}\big). \tag{18}$$

Thanks to Prop. 9 we can readily compute several quantities related to a PSD model such as its mean $\mathbb{E}_p[x] = X^\top b$ with $b = (A \circ K_{X,\eta/2}) \mathbf{1}_n$, its covariance or its characteristic function (see Appendix F for the explicit formulas and derivations). However, the result above is particularly useful to tackle the minimization in (17). In particular, since $\nabla L(\theta) = \mathbb{E}_{x \sim p} \nabla_\theta \ell(\theta, x)$, we can use Prop. 9 to directly compute the gradient of the objective function: it is sufficient to know how to evaluate (or approximate) the integral $c_{\nabla \ell_\theta, \eta}(z) = \int \nabla_\theta \ell(\theta, x) e^{-\eta \|x-z\|^2} \, dx$ for any $\theta \in \Theta$ and $z \in \mathbb{R}^d$. Then, we can use first order optimization methods, such as gradient descent, to efficiently solve (17). Remarkably, this approach works well also when we approximate $p$ with a PSD model $\hat{p}$. If $\ell$ is convex, since $\hat{p}$ is non-negative, the resulting $\hat{L}(\theta) = \mathbb{E}_{x \sim \hat{p}} \, \ell(\theta, x)$ is still a convex functional (see also the discussion on structured prediction in Sec. 4.2). This is not the case if we use more general estimators of $p$ that do not preserve non-negativity.

## 4.2 PSD Models for Estimating Conditional Probabilities

In supervised learning settings, one is typically interested in solving decision problems of the form $\min_{\theta \in \Theta} \mathbb{E}_{(x,y) \sim p} \ell(h_\theta(x), y)$ where $p$ is a probability over the joint input-output space $\mathcal{X} \times \mathcal{Y}$ and $h_\theta : \mathcal{X} \to \mathcal{Y}$ is a function parameterized by $\theta$. It is well-known [see e.g. 34] that the ideal solution of this problem is the $\theta_*$ such that for any $x \in \mathcal{X}$ the function $h_{\theta_*}(x) = \mathrm{argmin}_{z \in \mathcal{Y}} \mathbb{E}_{y \sim p(\cdot|x)} \ell(z, y)$ is the minimizer with respect to $z \in \mathcal{Y}$ of the conditional expectation of $\ell(z, y)$ given $x$. This leads to target functions that capture specific properties of $p$, such as moments. For instance, when $\ell$ is the squared loss, $h_{\theta_*}(x) = \mathbb{E}_{y \sim p(\cdot|x)} y$ corresponds to the conditional expectation of $y$ given $x$, while for $\ell$ the absolute value loss, $h_{\theta_*}$ recovers the conditional median of $p$.

In several applications, associating an input $x$ to a single quantity $h_\theta(x)$ in output is not necessarily ideal. For instance, when $p(y|x)$ is multi-modal, estimating the mean or median might not yield useful predictions for the given task. Moreover, estimators of the form $h_\theta$ require access to the full input $x$ to return a prediction, and therefore cannot be used when some features are missing (e.g. due to data corruption). In these contexts, an alternative viable strategy is to directly model the conditional probability. When using PSD models, conditional estimation can be performed in two steps, by first modeling the joint distribution $p(y, x) = f(y, x\,;\,A, [Y, X], (\eta, \eta'))$ (e.g. by learning it as suggested Sec. 3) and then use the operations in Sec. 2.1 to condition it with respect to $x_0 \in \mathcal{X}$ as

$$p(y|x_0) = \frac{p(y, x_0)}{p(x_0)} = f(y\,;\,B, Y, \eta) \quad \text{with} \quad B = \frac{A \circ (K_{X,x_0,\eta'} K_{X,x_0,\eta'}^\top)}{c_{2\eta} \mathrm{Tr}\big( A \circ (K_{X,x_0,\eta'} K_{X,x_0,\eta'}^\top) K_{Y,Y,\eta} \big)}. \tag{19}$$

In case of data corruption, it is sufficient to first marginalize $p(y, x)$ on the missing variables and then apply (19). Below we discuss a few applications of the conditional estimator.

**Conditional Expectation.** Conditional mean embeddings [31] are a well-established tool to efficiently compute the conditional expectation $\mathbb{E}_{y \sim p(\cdot|x_0)} \, g(y)$ of a function $g : \mathcal{Y} \to \mathbb{R}$. However, although they enjoy good approximation properties [16], they to not guarntee the resulting estimator to take only non-negative values. In contrast, when $p$ is a PSD model (or an approximation), we can apply Prop. 9 to $p(\cdot|x_0)$ in (19) and evaluate the conditional expectation of any $g$ for which we know how to compute (or approximate) the integral $c_{g,\eta}(z) = \int g(y) e^{-\eta \|y-z\|^2} \, dy$. In particular we have $\mathbb{E}_{y \sim p(\cdot|x)} \, g(y) = \mathrm{Tr}((B \circ K_{Y,Y,\eta/2}) G)$ with $B$ as in (19) and $G$ the matrix with entries $G_{ij} = c_{g,2\eta}\big(\tfrac{y_i + y_j}{2}\big)$. Remarkably, differently from conditional mean embeddings estimators, this strategy allows us to compute the conditional expectations also of functions $g$ not in $\mathcal{H}_\eta$.

**Structured Prediction.** Structured prediction identifies supervised learning problems where the output space $\mathcal{Y}$ has complex structures so that it is challenging to find good parametrizations for $h_\theta : \mathcal{X} \to \mathcal{Y}$ [4, 19]. In [7], a strategy was proposed to tackle these settings by first learning an approximation $\psi_\theta(z, x) \approx \mathbb{E}_{y \sim p(\cdot|x)} \ell(z, y)$ and then model $h_\theta(x) = \mathrm{argmin}_{z \in \mathcal{Y}} \psi_\theta(z, x)$. However, the resulting function $\psi_\theta(\cdot, x)$ is not guaranteed to be convex, even when $\ell$ is convex. In contrast, by combining the conditional PSD estimator in (19) with the reasoning in Sec. 4.1, we have a strategy that overcomes this issue: when $p(\cdot|x)$ is a PSD model approximating $p$, we can compute its gradient

---

**Algorithm 1** PSD Hidden Markov Model

---

**Input:** Transition $\hat{\tau}(x_+, x) = f(x_+, x\,;\, B, [X_+, X], (\eta_+, \eta))$, initial $\hat{p}(x_0) = f(x_0\,;\, A_0, X_0, \eta_0)$ and observation $\hat{\omega}(y, x) = f(y, x\,;\, C, [Y, X'], (\eta_{obs}, \eta'))$ distributions. $\tilde{X}$ as in Prop. 10. $\tilde{\eta} = \frac{\eta(\eta' + \eta_+)}{\eta + \eta' + \eta_+}$.

$\tilde{X}' = \frac{\eta}{\eta + \eta' + \eta_+} X \otimes \mathbf{1}_{nm} + \frac{\eta' + \eta_+}{\eta + \eta' + \eta_+} \mathbf{1}_n \otimes \tilde{X}$, and $\tilde{\eta}' = \frac{\eta' \eta_+}{\eta' + \eta_+}$

**For** any new observation $y_t$:

$\quad C_t = C \circ (K_{Y, y_t, \eta_{obs}}^\top K_{Y, y_t, \eta_{obs}})$ \qquad // Partial evaluation $\hat{\omega}_t(x_+) = \hat{\omega}(y = y_t, x_+)$

$\quad B_t = (B \otimes A_{t-1}) \circ \left( \mathrm{vec}(K_{X, \tilde{X}, \tilde{\eta}}) \mathrm{vec}(K_{X, \tilde{X}, \tilde{\eta}})^\top \right)$ \qquad // Product $\hat{\beta}_t(x_+, x) = \hat{\tau}(x_+, x) \hat{p}(x | y_{1:t-1})$

$\quad D_t = (I_n \otimes \mathbf{1}_{nm}^\top)(B_t \circ K_{\tilde{X}', \tilde{X}', \frac{\tilde{\eta}}{2}})(I_n \otimes \mathbf{1}_{nm})$ \qquad // Marginalization $\hat{\beta}_t(x_+) = \int \beta_t(x_+, x)\, dx$

$\quad E_t = (C_t \otimes D_t) \circ \left( \mathrm{vec}(K_{X_+, X', \frac{\tilde{\eta}'}{2}}) \mathrm{vec}(K_{X_+, X', \frac{\tilde{\eta}'}{2}})^\top \right)$ \qquad // Product $\hat{\pi}_t(x_+) = \hat{\omega}_t(x_+) \hat{\beta}_t(x_+)$

$\quad A_t = E_t / c_t$, with $c_t = c_{2(\eta' + \eta_+)} \mathrm{Tr}(E_t K_{\tilde{X}, \tilde{X}, \frac{\eta' + \eta_+}{2}})$ \qquad // Normalization $\hat{p}(x_+ | y_{1:t}) = \frac{\hat{\pi}_t(x_+)}{\int \hat{\pi}_t(x_+)\, dx}$

**Return** $f(x_t\,;\, A_t, \tilde{X}, \eta' + \eta_+)$

---

$\mathbb{E}_{y \sim p(\cdot | x)} \nabla_z \ell(z, y)$ as mentioned in Sec. 4.1 using Prop. 9. Moreover, if $\ell(\cdot, y)$ is convex, the term $\mathbb{E}_{y \sim p(\cdot | x)} \ell(\cdot, y)$ is also convex, and we can use methods such as gradient descent to find $h(x)$ exactly.

**Mode Estimation.** When the output distribution $p(y | x)$ is multimodal, having access to an explicit form for the conditional density can be useful to estimate its modes. This problem is typically non-convex, yet, when the output $y$ belongs to a small dimensional space (e.g. in classification or scalar-valued regression settings), efficient approximations exist (e.g. bisection).

### 4.3 Inference on Hidden Markov Models

We consider the problem of performing inference on hidden Markov models (HMM) using PSD models. Let $(x_t)_{t \in \mathbb{N}}$ and $(y_t)_{t \in \mathbb{N}}$ denote two sequences of states and observations respectively. For each $t \geq 1$, we denote by $x_{0:t} = x_0, \ldots, x_t$ and $y_{1:t} = y_1, \ldots, y_t$ and we assume that $p(x_t | x_{0:t-1}, y_{1:t-1}) = p(x_t | x_{t-1}) = \tau(x_t, x_{t-1})$ and $p(y_t | x_{0:t}, y_{1:t-1}) = p(y_t | x_t) = \omega(y_t, x_t)$ with $\tau : \mathcal{X} \times \mathcal{X} \to \mathbb{R}_+$ and $\omega : \mathcal{Y} \times \mathcal{X} \to \mathbb{R}_+$ respectively the transition and observation functions.

Our goal is to infer the distribution of possible states $x_t$ at time $t$, given all the observations $y_{1:t}$ and a probability $p(x_0)$ on the possible initial states. We focus on this goal for simplicity, but other forms of inferences are possible (e.g. estimating $x_{m+t}$, namely $m$ steps into the future or the past). We assume that the transition and observation functions can be well approximated by PSD models $\hat{\tau}$ and $\hat{\omega}$ (e.g. by learning them or known a-priori for the problem's dynamics). Then, given a PSD model estimate $\hat{p}(x_0)$ of the initial state $p(x_0)$, we can recursively define the sequence of estimates

$$\hat{p}(x_t | y_{t:1}) = \frac{\hat{\tau}(y_t, x_t) \int \omega(x_t, x_{t-1}) \hat{p}(x_t | y_{1:t-1})\, dx_{t-1}}{\int \hat{\tau}(y_t, x_t) \omega(x_t, x_{t-1}) \hat{p}(x_t | y_{1:t-1})\, dx_t dx_{t-1}}. \tag{20}$$

Note that when $\hat{\tau}, \hat{\omega}, \hat{p}(x_0)$ correspond to the real transition, observation and initial state probability, the formula above yields the exact distribution $p(x_t | y_{1:t})$ over the states at time $t$ (this follows directly by subsequent applications of Bayes' rule. See also e.g. [5]). If $\hat{\tau}, \hat{\omega}, \hat{p}(x_0)$ are PSD models, then each of the $\hat{p}(x_t | y_{t:1})$ is a PSD model recursively defined only in terms of the previous estimate and the operations introduced in Sec. 2.1. In particular, we have the following result.

**Proposition 10** (PSD Hidden Markov Models (HMM)). *Let $X_0 \in \mathbb{R}^{n_0 \times d}$, $X_+, X \in \mathbb{R}^{n \times d}$, $X' \in \mathbb{R}^{m \times d}$, $Y \in \mathbb{R}^{m \times d'}$, $A_0 \in \mathbb{S}_+^{n_0}$, $A \in \mathbb{S}_+^n$, $B \in \mathbb{S}_+^m$ and $\eta_0, \eta, \eta', \eta_+ \in \mathbb{R}_{++}^d$, $\eta_{obs} \in \mathbb{R}_{++}^{d'}$. Let*

$$\hat{\tau}(x_+, x) = f(x_+, x\,;\, B, [X_+, X], (\eta_+, \eta)), \qquad \hat{\omega}(y, x) = f(y, x\,;\, C, [Y, X'], (\eta_{obs}, \eta')), \tag{21}$$

*be approximate transition and observation functions. Then, given the initial state probability $\hat{p}(x_0) = f(x_0\,;\, A_0, X_0, \eta_0)$, for any $t \geq 1$, the estimate $\hat{p}$ in (20) is a PSD model of the form*

$$\hat{p}(x_t | y_{t:1}) = f(x_t\,;\, A_t, \tilde{X}, \eta' + \eta_+), \tag{22}$$

*where $\tilde{X} = \frac{\eta'}{\eta' + \eta_+} X' \otimes \mathbf{1}_n + \frac{\eta_+}{\eta' + \eta_+} \mathbf{1}_m \otimes X_+$ and $A_t$ is recursively obtained from $A_{t-1}$ as in Alg. 1.*

**Remark 4** (Sum-product Algorithm)**.** *Eq.* (20) *is an instance of the so-called sum-product algorithm, a standard inference method for graphical models [5] (of which HMMs are a special case). The application of the sum-product algorithm relies mainly on sum and product rules for probabilities (as is the case for HMMs in* (20)*). Hence, according to* Sec. 2.1*, it is highly compatible with PSD models.*

## 5  Discussion

In this work we have shown that PSD models are a strong candidate in practical application related to probabilistic inference. They satisfy both requirements for an ideal probabilistic model: $i$) they perform exact sum and product rule in terms of efficient matrix operations; $ii$) we proved that they can concisely approximate a wide range of probabilities.

**Future Directions.** We identify three main directions for future work: $i$) when performing inference on large graphical models (see Remark 4) the multiplication of PSD models might lead to an inflation in the number of base points. Building on our compression strategy, we plan to further investigate low-rank approximations to mitigate this issue. $ii$) An interesting problem is to understand how to efficiently sample from a PSD model. A first answer to this open question was recently given in [15]. $iii$) The current paper has a purely theoretical and algorithmic focus. In the future, we plan to investigate the empirical behavior of PSD models on the applications introduced in Sec. 4. Related to this, we plan to develop a library for operations with PSD models and make it available to the community.

**Acknowledgments.** A.R. acknowleges the support of the French government under management of Agence Nationale de la Recherche as part of the "Investissements d'avenir" program, reference ANR-19-P3IA-0001 (PRAIRIE 3IA Institute) and the support of the European Research Council (grant REAL 947908). C.C. acknowledges the support of the Royal Society (grant SPREM RGS\R1\201149) and Amazon.com Inc. (Amazon Research Award – ARA 2020)

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
