# Appendix

The appendix is organized as follows:

## A    Notation and definitions

We introduce basic notation and review results that will be useful in the following.

**Multi-index notation.** Let $\alpha \in \mathbb{N}_0^d$, $x \in \mathbb{R}^d$ and $f$ be an infinitely differentiable function on $\mathbb{R}^d$, we introduce the following notation

$$|\alpha| = \sum_{j=1}^d \alpha_i, \quad \alpha! = \prod_{j=1}^d \alpha_j!, \quad x^\alpha = \prod_{j=1}^d x_j^{\alpha_j}, \quad \partial^\alpha f = \frac{\partial^{|\alpha|} f}{\partial x_1^{\alpha_1} \cdots \partial x_d^{\alpha_d}}.$$

We introduce also the notation $D^\alpha$ that corresponds to the multivariate distributional derivative of order $\alpha$ and such that

$$D^\alpha f = \partial^\alpha f$$

for functions that are differentiable at least $|\alpha|$ times [1].

**Fourier Transform.** Given two functions $f, g : \mathbb{R}^d \to \mathbb{R}$ on some set $\mathbb{R}^d$, we denote by $f \cdot g$ the function corresponding to *pointwise product* of $f, g$, i.e.,

$$(f \cdot g)(x) = f(x)g(x), \quad \forall x \in \mathbb{R}^d.$$

Let $f, g \in L^1(\mathbb{R}^d)$ we denote the *convolution* by $f \star g$

$$(f \star g)(x) = \int_{\mathbb{R}^d} f(y)g(x - y)dy.$$

We now recall some basic properties, that will be used in the rest of the appendix.

**Proposition A.1** (Basic properties of the Fourier transform [39], Chapter 5.2.)**.**

(a) *There exists a linear isometry $\mathcal{F} : L^2(\mathbb{R}^d) \to L^2(\mathbb{R}^d)$ satisfying*

$$\mathcal{F}[f] = \int_{\mathbb{R}^d} e^{-2\pi i \, \omega^\top x} f(x) \, dx \quad \forall f \in L^1(\mathbb{R}^d) \cap L^2(\mathbb{R}^d),$$

*where $i = \sqrt{-1}$. The isometry is uniquely determined by the property in the equation above.*

(b) *Let $f \in L^2(\mathbb{R}^d)$, then $\|\mathcal{F}[f]\|_{L^2(\mathbb{R}^d)} = \|f\|_{L^2(\mathbb{R}^d)}$.*

(c) *Let $f \in L^2(\mathbb{R}^d), r > 0$ and define $f_r(x) = f(\frac{x}{r}), \forall x \in \mathbb{R}^d$, then $\mathcal{F}[f_r](\omega) = r^d \mathcal{F}[f](r\omega)$.*

(d) *Let $f, g \in L^1(\mathbb{R}^d)$, then $\mathcal{F}[f \cdot g] = \mathcal{F}[f] \star \mathcal{F}[g]$.*

(e) *Let $\alpha \in \mathbb{N}_0^d$, $f, D^\alpha f \in L^2(\mathbb{R}^d)$, then $\mathcal{F}[D^\alpha f](\omega) = (2\pi i)^{|\alpha|}\omega^\alpha \mathcal{F}[f](\omega)$, $\forall \omega \in \mathbb{R}^d$.*

(f) *Let $f \in L^1(\mathbb{R}^d) \cap L^2(\mathbb{R}^d)$, then $\|\mathcal{F}[f]\|_{L^\infty(\mathbb{R}^d)} \leq \|f\|_{L^1(\mathbb{R}^d)}$.*

(g) *Let $f \in L^\infty(\mathbb{R}^d) \cap L^2(\mathbb{R}^d)$, then $\|f\|_{L^\infty(\mathbb{R}^d)} \leq \|\mathcal{F}[f]\|_{L^1(\mathbb{R}^d)}$.*

**Reproducing kernel Hilbert spaces for translation invariant kernels..** We now list some important facts about reproducing kernel Hilbert spaces in the case of translation invariant kernels on $\mathbb{R}^d$. For this paragraph, we refer to [34, 39]. For the general treatment of positive kernels and Reproducing kernel Hilbert spaces, see [3, 34]. Let $v : \mathbb{R}^d \to \mathbb{R}$ such that its Fourier transform $\mathcal{F}[v] \in L^1(\mathbb{R}^d)$ and satisfies $\mathcal{F}[v](\omega) \geq 0$ for all $\omega \in \mathbb{R}^d$. Then, the following hold.

(a) The function $k : \mathbb{R}^d \times \mathbb{R}^d \to \mathbb{R}$ defined as $k(x, x') = v(x - x')$ for any $x, x' \in \mathbb{R}^d$ is a positive kernel and is called *translation invariant kernel*.

(b) The *reproducing kernel Hilbert space* (RKHS) $\mathcal{H}$ and its norm $\|\cdot\|_{\mathcal{H}}$ are characterized by

$$\mathcal{H} = \{f \in L^2(\mathbb{R}^d) \mid \|f\|_{\mathcal{H}} < \infty\}, \quad \|f\|_{\mathcal{H}}^2 = \int_{\mathbb{R}^d} \frac{|\mathcal{F}[f](\omega)|^2}{\mathcal{F}[v](\omega)} d\omega, \qquad \text{(A.1)}$$

(c) $\mathcal{H}$ is a separable Hilbert space, whose inner product $\langle \cdot, \cdot \rangle_{\mathcal{H}}$ is characterized by

$$\langle f, g \rangle_{\mathcal{H}} = \int_{\mathbb{R}^d} \frac{\mathcal{F}[f](\omega)\overline{\mathcal{F}[g](\omega)}}{\mathcal{F}[v](\omega)} d\omega.$$

In the rest of the paper, when clear from the context we will simplify the notation of the inner product, by using $f^\top g$ for $f, g \in \mathcal{H}$, instead of the more cumbersome $\langle f, g \rangle_{\mathcal{H}}$.

(d) The feature map $\phi : \mathbb{R}^d \to \mathcal{H}$ is defined as $\phi(x) = k(x - \cdot) \in \mathcal{H}$ for any $x \in \mathbb{R}^d$.

(e) The functions in $\mathcal{H}$ have the *reproducing property*, i.e.,

$$f(x) = \langle f, \phi(x) \rangle_{\mathcal{H}}, \quad \forall f \in \mathcal{H}, x \in \mathbb{R}^d, \qquad \text{(A.2)}$$

in particular $k(x', x) = \langle \phi(x'), \phi(x) \rangle_{\mathcal{H}}$ for any $x', x \in \mathbb{R}^d$.

We now introduce an important example of translation invariant kernel and the associated RKHS, that will be useful in our analysis.

**Example 1** (Gaussian Reproducing kernel Hilbert space). *Let $\eta \in \mathbb{R}_{++}^d$ and $k_\eta(x, x') = e^{-(x-x')^\top \mathrm{diag}(\eta)(x-x')}$, for $x, x' \in \mathbb{R}^d$ be the Gaussian kernel with precision $\eta$. The function $k_\eta$ is a translation invariant kernel, since $k_\eta(x, x') = v(x - x')$ with $v(z) = e^{-\|D^{1/2}z\|^2}$, $D = \mathrm{diag}(\eta)$ and $\mathcal{F}[v](\omega) = c_\eta e^{-\pi^2\|D^{-1/2}\omega\|^2}$, $c_\eta = \pi^{d/2}\det(D)^{-1/2}$, for $\omega \in \mathbb{R}^d$ is in $L^1(\mathbb{R}^d)$ and satisfies $\mathcal{F}[v](\omega) \geq 0$ for all $\omega \in \mathbb{R}^d$. The associated reproducing kernel Hilbert space $\mathcal{H}_\eta$ is defined according to (A.1), with norm*

$$\|f\|_{\mathcal{H}_\eta}^2 = \frac{1}{c_\eta} \int_{\mathbb{R}^d} |\mathcal{F}[f](\omega)|^2 e^{\pi^2\|D^{-1/2}\omega\|^2} d\omega, \qquad \forall f \in L^2(\mathbb{R}^d). \qquad \text{(A.3)}$$

*The inner product and the feature map $\phi_\eta$ are defined as in the discussion above.*

## A.1 Sobolev spaces

Let $\beta \in \mathbb{N}$, $p \in [1, \infty]$ and let $\Omega \subseteq \mathbb{R}^d$ be an open set. The set $L^p(\Omega)$ denotes the set of $p$-integrable functions on $\Omega$ for $p \in [1, \infty)$ and that of the essentially bounded on $\Omega$ when $p = \infty$. The set $W_p^\beta(\Omega)$ denotes the Sobolev space, i.e., the set of measurable functions with their distributional derivatives up to $\beta$-th order belonging to $L^p(\Omega)$,

$$W_p^\beta(\Omega) = \{f \in L^p(\Omega) \mid \|f\|_{W_p^\beta(\Omega)} < \infty\}, \quad \|f\|_{W_p^\beta(\Omega)}^p = \sum_{|\alpha| \leq \beta} \|D^\alpha f\|_{L^p(\Omega)}^p, \qquad \text{(A.4)}$$

where $D^\alpha$ denotes the distributional derivative. In the case of $p = \infty$,

$$\|f\|_{W_\infty^\beta(\Omega)} = \max_{|\alpha| \leq \beta} \|D^\alpha f\|_{L^\infty(\Omega)} \qquad \text{(A.5)}$$

We now recall some basic results about Sobolev spaces that are useful for the proofs in this paper. First we start by recalling the restriction properties of Sobolev spaces. Let $\Omega \subseteq \Omega' \subseteq \mathbb{R}^d$ be two open sets. Let $\beta \in \mathbb{N}$ and $p \in [1, \infty]$. By definition of the Sobolev norm above we have

$$\|g|_\Omega\|_{W_p^s(\Omega)} \leq \|g\|_{W_p^s(\Omega')},$$

and so $g|_\Omega \in W_p^s(\Omega)$ for any $g \in W_p^s(\Omega')$. Now we recall the extension properties of Sobolev spaces.

**Proposition A.2** (Stein total extension theorem, 5.24 in [1] or [23]). *Let $\Omega$ be a bounded open subset of $\mathbb{R}^d$ with locally Lipschitz boundary [1]. For any measurable function $h : \Omega \to \mathbb{R}$, there exists a function $\tilde{h} : \mathbb{R}^d \to \mathbb{R}$, such that $\tilde{h}|_\Omega = h$ almost everywhere on $\Omega$ and for any $\beta \geq 0$ and $p \in [1, \infty]$, the condition $\|h\|_{W_p^\beta(\Omega)} < \infty$ implies $\|\tilde{h}\|_{W_p^\beta(\mathbb{R}^d)} \leq C_{\Omega,\beta,p}\|h\|_{W_p^\beta(\Omega)}$.*

**Corollary A.3.** *Let $\mathcal{X} \subset \mathbb{R}^d$ be a non-empty open set with Lipschitz boundary. Let $\beta \in \mathbb{N}, p \in [1, \infty]$. Then for any function $f \in W_p^\beta(\mathcal{X}) \cap L^\infty(\mathcal{X})$ there exists an extension $\tilde{f}$ on $\mathbb{R}^d$, i.e. a function $\tilde{f} \in W_p^\beta(\mathbb{R}^d) \cap L^\infty(\mathbb{R}^d)$ such that*

$$f = f|_{\mathcal{X}} \text{ a.e. on} \mathcal{X}, \quad \|\tilde{f}\|_{L^\infty(\mathbb{R}^d)} \leq C\|f\|_{L^\infty(\mathcal{X})}, \quad \|\tilde{f}\|_{W_p^\beta(\mathbb{R}^d)} \leq C'\|f\|_{L^\infty(\mathcal{X})}. \tag{A.6}$$

*The constant $C$ depends only on $\mathcal{X}, d$, and the constant $C'$ only on $\mathcal{X}, \beta, d, p$*

**Proposition A.4.** *Let $\mathcal{X}$ be an open bounded set with Lipschitz boundary. Let $f$ be a function that is $m$ times differentiable on the closure of $\mathcal{X}$. Then there exists a function $\tilde{f} \in W_p^m(\mathcal{X}) \cap L^\infty(\mathcal{X})$ for any $p \in [1, \infty]$, such that $\tilde{f} = f$ on $\mathcal{X}$.*

*Proof.* A function $f$ that is $m$-times differentiable on the closure of $X$ belongs also to $W_\infty^m(\mathcal{X})$ since each derivative up to order $m$ is continuous and the set $\mathcal{X}$ is bounded. Then $f$ satisfies also $f \in W_p^m(\mathcal{X}) \cap L^\infty(\mathcal{X})$ since $W_\infty^m(\mathcal{X}) \subset L^\infty(\mathcal{X})$ and $W_\infty^m(\mathcal{X}) \subset W_p^\beta(\mathcal{X})$, by construction, for bounded $\mathcal{X}$ and any $p \in [1, \infty]$. $\qquad\square$

The following proposition provides a useful characterization of the space $W_2^\beta(\mathbb{R}^d)$

**Proposition A.5** (Characterization of the Sobolev space $W_2^k(\mathbb{R}^d)$, [39]). *Let $k \in \mathbb{N}$. The norm of the Sobolev space $\|\cdot\|_{W_2^k(\mathbb{R}^d)}$ is equivalent to the following norm*

$$\|f\|'^2_{W_2^k(\mathbb{R}^d)} = \int_{\mathbb{R}^d} |\mathcal{F}[f](\omega)|^2 (1 + \|\omega\|^2)^k \, d\omega, \quad \forall f \in L^2(\mathbb{R}^d) \tag{A.7}$$

*and satisfies*

$$\tfrac{1}{(2\pi)^{2k}}\|f\|^2_{W_2^k(\mathbb{R}^d)} \leq \|f\|'_{W_2^k(\mathbb{R}^d)} \leq 2^{2k}\|f\|^2_{W_2^k(\mathbb{R}^d)}, \quad \forall f \in L^2(\mathbb{R}^d) \tag{A.8}$$

*Moreover, when $k > d/2$, then $W_2^k(\mathbb{R}^d)$ is a reproducing kernel Hilbert space.*

*Proof.* Consider first the seminorm $|g|^2_{W_2^t(\mathbb{R}^d)} = \sum_{|\alpha| \leq t} \|D^\alpha g\|^2_{L^2(\mathbb{R}^d)}$. We have that $\|g\|^2_{W_2^k(\mathbb{R}^d)} = \sum_{t=0}^k |g|^2_{W_2^t(\mathbb{R}^d)}$. Now let $0 \leq t \leq k$. By using the properties of the Fourier transform (in particular, the Plancherel theorem and the transform of a distributional derivative Prop. A.1) we have that

$$|g|^2_{W_2^t(\mathbb{R}^d)} = \sum_{|\alpha|=t} \|D^\alpha g\|^2_{L^2(\mathbb{R}^d)} = \sum_{|\alpha|=t} \|(2\pi i)^\alpha \omega^\alpha \mathcal{F}[g](\omega)\|^2_{L^2(\mathbb{R}^d)} = \int \sum_{|\alpha|=t} (2\pi\omega)^{2\alpha} |\mathcal{F}[g](\omega)|^2 d\omega.$$

Now note that, by the multinomial theorem, $\|2\pi\omega\|^{2t} = (2\pi\omega_1^2 + \cdots + 2\pi\omega_d^2)^k = \sum_{|\alpha|=k} \binom{k}{\alpha}(2\pi\omega)^\alpha$. Since $1 \leq \binom{t}{\alpha} \leq 2^t$ for any $\alpha \in \mathbb{N}_0^d, |\alpha| = t$, then $2^{-t}\|2\pi\omega\|^{2t} \leq \sum_{|\alpha|=t}(2\pi\omega)^{2\alpha} \leq \|2\pi\omega\|^{2t}$, so

$$|g|^2_{W_2^t(\mathbb{R}^d)} \leq (2\pi)^{2t} \int \|\omega\|^{2t} |\mathcal{F}[g](\omega)|^2 d\omega \leq 2^t |g|^2_{W_2^t(\mathbb{R}^d)}. \tag{A.9}$$

Since, $(1 + \|\omega\|^2)^k = \sum_{t=0}^k \binom{k}{t}\|\omega\|^{2k}$ and so $\sum_{t=0}^k \|\omega\|^{2t} \leq (1 + \|\omega\|^2)^k \leq 2^k \sum_{t=0}^k \|\omega\|^{2t}$, then

$$\|g\|_{W_2^k(\mathbb{R}^d)} = \sum_{t=0}^k |g|^2_{W_2^t(\mathbb{R}^d)} \leq (2\pi)^{2k} \sum_{t=0}^k \int \|\omega\|^{2t} |\mathcal{F}[g](\omega)|^2 d\omega \leq (2\pi)^{2k} \int (1+\|\omega\|^2)^k |\mathcal{F}[g](\omega)|^2,$$

moreover

$$\int (1+\|\omega\|^2)^k |\mathcal{F}[g](\omega)|^2 \leq 2^k \sum_{t=0}^{k} (2\pi)^{2t} \int \|\omega\|^{2t} |\mathcal{F}[g](\omega)|^2 \leq 2^{2k} \sum_{t=0}^{k} |g|^2_{W_2^k(\mathbb{R}^d)} = 2^{2k} \|g\|_{W_2^k(\mathbb{R}^d)}.$$

To conclude, we recall that when $k > d/2$ the space $W_2^k(\mathbb{R}^d)$ endowed with the equivalent norm $\|\cdot\|'_{W_2^k(\mathbb{R}^d)}$ is a reproducing kernel Hilbert space [39]. $\qquad\square$

# B  Useful linear operators in RKHS

Consider the space $\mathcal{G}_\eta = \mathcal{H}_\eta \otimes \mathcal{H}_\eta = \{v \otimes v' \mid v, v' \in \mathcal{H}_\eta\}$ endowed with the inner product $\langle u \otimes u', v \otimes v' \rangle_{\mathcal{G}_\eta} = \langle u, v \rangle_{\mathcal{H}_\eta} \langle u', v' \rangle_{\mathcal{H}_\eta}$ for any $u, u', v, v' \in \mathcal{H}_\eta$. Denote by vec the unitary map that maps the Hilbert-Schmidt operators on $\mathcal{H}_\eta$ in vectors in $\mathcal{H}_\eta^\otimes$. In particular, for any $u, v \in \mathcal{H}_\eta$, we have $\text{vec}(uv^\top) = v \otimes u$, moreover for any $\mathsf{M}, \mathsf{M}' : \mathcal{H}_\eta \to \mathcal{H}_\eta$ with finite Hilbert-Schmidt norm

$$\langle \text{vec}(\mathsf{M}), \text{vec}(\mathsf{M}') \rangle_{\mathcal{G}_\eta} = \text{Tr}(\mathsf{M}^*\mathsf{M}'), \quad \langle \text{vec}(\mathsf{M}), v \otimes u \rangle_{\mathcal{G}_\eta} = v^\top \mathsf{M} u. \tag{B.1}$$

Now denote by $\psi_\eta$ the feature map $\psi_\eta(x) = \phi_\eta(x) \otimes \phi_\eta(x)$ for any $x \in \mathbb{R}^d$. We define the operator $Q \in \mathbb{S}_+(\mathcal{G}_\eta)$ and the vectors $\hat{v}, v \in \mathcal{G}_\eta$ as follows

$$Q = \int_{\mathcal{X}} \psi_\eta(x)\psi_\eta(x)^\top dx, \quad \hat{v} = \frac{1}{n}\sum_{i=1}^{n} \psi_\eta(x_i), \quad v = \int_{\mathcal{X}} \psi_\eta(x)p(x)dx. \tag{B.2}$$

Define the operator $S : \mathcal{G}_\eta \to L^2(\mathcal{X})$ as

$$Sf = \langle \psi_\eta(\cdot), f \rangle_{\mathcal{G}_\eta} \in L^2(\mathcal{X}), \quad \forall f \in \mathcal{G}_\eta \tag{B.3}$$

$$S^*\alpha = \int \alpha(x)\psi_\eta(x)dx \in \mathcal{G}_\eta, \quad \forall \alpha \in L^2(\mathcal{X}). \tag{B.4}$$

Note, in particular, that $Q$ and $v$ are characterized by

$$Q = S^*S, \quad v = S^*p. \tag{B.5}$$

Given $\tilde{x}_1, \ldots, \tilde{x}_m \in \mathbb{R}^d$ define the operator $\tilde{Z} : \mathcal{H}_\eta \to \mathbb{R}^m$ as $\tilde{Z} = (\phi_\eta(\tilde{x}_1)^\top, \ldots, \phi_\eta(\tilde{x}_m)^\top)$, in particular we have

$$\tilde{Z}u = (\phi_\eta(\tilde{x}_1)^\top u, \ldots, \phi_\eta(\tilde{x}_m)^\top u), \forall u \in \mathcal{H}_\eta$$
$$\tilde{Z}^*\alpha = \sum_{i=1}^{m} \phi_\eta(\tilde{x}_i)\alpha_i, \forall \alpha \in \mathbb{R}^m, \tag{B.6}$$

In particular, note that for any $A \in \mathbb{R}^{m \times m}$

$$\tilde{Z}^*A\tilde{Z} = \sum_{i,j=1}^{m} A_{i,j}\phi_\eta(\tilde{x}_i)\phi_\eta(\tilde{x}_j)^\top, \quad \tilde{Z}\tilde{Z}^* = K_{\tilde{X},\tilde{X},\eta}. \tag{B.7}$$

Given $\tilde{Z} : \mathcal{H}_\eta \to \mathbb{R}^m$, define the associated projection operator $\tilde{P} : \mathcal{H}_\eta \to \mathcal{H}_\eta$ on the range of the adjoint $\tilde{Z}^*$. In particular, note that

$$\tilde{P} = \tilde{Z}^* K_{\tilde{X},\tilde{X},\eta}^{-1} \tilde{Z} \tag{B.8}$$

indeed, since $\tilde{Z}\tilde{Z}^* = K_{\tilde{X},\tilde{X},\eta}$ and it is invertible for any $\eta \in \mathbb{R}_{++}^d$, then

$$\tilde{P}^2 = \tilde{Z}^* K_{\tilde{X},\tilde{X},\eta}^{-1} \tilde{Z}\tilde{Z}^* K_{\tilde{X},\tilde{X},\eta}^{-1} \tilde{Z}^* = \tilde{Z}^* K_{\tilde{X},\tilde{X},\eta}^{-1} K_{\tilde{X},\tilde{X},\eta} K_{\tilde{X},\tilde{X},\eta}^{-1} \tilde{Z} = \tilde{Z}^* K_{\tilde{X},\tilde{X},\eta}^{-1} \tilde{Z} = \tilde{P}. \tag{B.9}$$

and

$$\tilde{P}\tilde{Z}^* = \tilde{Z}^* K_{\tilde{X},\tilde{X},\eta}^{-1} \tilde{Z}\tilde{Z}^* = \tilde{Z}^* K_{\tilde{X},\tilde{X},\eta}^{-1} K_{\tilde{X},\tilde{X},\eta} = \tilde{Z}^*. \tag{B.10}$$

and analogously $\tilde{Z}\tilde{P} = \tilde{Z}$. This implies also that $\tilde{P}\phi_\eta(\tilde{x}_i) = \phi_\eta(\tilde{x}_i)$ for any $i = 1, \ldots, m$.

## C   Compression of a PSD model

Let $\mathcal{X} \subset \mathbb{R}^d$ be an open set with Lipschitz boundary, contained in the hypercube $[-R, R]^d$ with $R > 0$. Given $\tilde{x}_1, \ldots, \tilde{x}_m \in \mathcal{X}$ be $m$ points in $[-R, R]^d$. Define the base point matrix $\tilde{X} \in \mathbb{R}^{m \times d}$ to be the matrix whose $j$-th row is the point $\tilde{x}_j$. The following result holds. We introduce the so called *fill distance* [39]

$$h = \max_{x \in [-R,R]^d} \min_{z \in \tilde{X}} \|x - z\|, \tag{C.1}$$

In the next lemma we specialize Theorem 4.5 of [22], to obtain explicit constants in terms of $R$ and of our $\eta$. In particular, we identify the scale parameter $\sigma = \min(R, 1/\sqrt{\max_i \eta_i})$. This is interesting since it shows the effect of the precision $\eta$ of the kernel (if it was a Gaussian probability, it variance would scale exactly as $1/\sqrt{\eta}$).

**Lemma C.1** (Norm of functions with scattered zeros). *Let* $T = (-R, R)^d$ *and* $\eta \in \mathbb{R}_{++}^d$. *Let* $u \in \mathcal{H}_\eta$ *satisfying* $u(\tilde{x}_1) = \cdots = u(\tilde{x}_m) = 0$. *There exists three constants* $c, C', C''$ *depending only on* $d$ *(and in particular, independent from* $R, \eta, u, \tilde{x}_1, \ldots, \tilde{x}_m$*), such that, when* $h \leq \sigma/C'$*, then,*

$$\|u\|_{L^\infty(T)} \leq C q_\eta \, e^{-\frac{c\sigma}{h} \log \frac{c\sigma}{h}} \, \|u\|_{\mathcal{H}_\eta}, \tag{C.2}$$

*with* $q_\eta = \det(\frac{1}{\eta_+} \mathrm{diag}(\eta))^{-1/4}$ *and* $\sigma = \min(R, \frac{1}{\sqrt{\eta_+}})$ *and* $\eta_+ = \max_{i=1,\ldots,d} \eta_i$.

*Proof.* By Theorem 4.3 of [22] there exists two constants $B_d, B_d'$ depending only on $d$ (and independent from $R, \tilde{x}_j$), such that for any $k \in \mathbb{N}, k > d/2 + 1$ and $u \in W_2^k(T)$ satisfying $u(\tilde{x}_1) = \cdots = u(\tilde{x}_m) = 0$, the following holds,

$$\|u\|_{L^\infty(T)} \leq \frac{B_d^k k^{k-d/2}}{k!} \, h^{k-d/2} \, |u|_{W_2^k(T)}, \tag{C.3}$$

when $kh \leq R/B_d'$. Here the seminorm $|u|_{W_2^k(T)}$, by using the multinomial notation recalled in [Appendix A](#), corresponds to $|u|_{W_2^k(T)}^2 = \sum_{|\alpha|=k} \|D^\alpha u\|_{L^2(T)}^2$. Our goal is to apply the result above to $f \in \mathcal{H}_\eta$. First we recall that $\mathcal{H}_\eta \subset W_2^k(\mathbb{R}^d)$ for any $k$ [22]. Then, since $T \subset \mathbb{R}^d$, we have $|f|_{W_2^k(T)}^2 = \sum_{|\alpha|=k} \|D^\alpha f\|_{L^2(T)}^2 \leq \sum_{|\alpha|=k} \|D^\alpha f\|_{L^2(\mathbb{R}^d)}^2 = |f|_{W_2^k(\mathbb{R}^d)}^2$. Then, using [(A.9)](#) we have $|f|_{W_2^k(\mathbb{R}^d)}^2 \leq (2\pi)^{2k} \int \|\omega\|^{2k} |\mathcal{F}[f](\omega)|^2 d\omega$. Now, denote by $D$ the matrix $D = \mathrm{diag}(\eta)$ and $c_\eta = \pi^{d/2} \det(D)^{-1/2}$. By the characterization of the norm $\|\cdot\|_{\mathcal{H}_\eta}$ in terms of the Fourier transform reported in [Example 1](#), we have

$$\|f\|_{W_2^k(\mathbb{R}^d)}^2 \leq (2\pi)^{2k} \int \|\omega\|^{2k} |\mathcal{F}[f](\omega)|^2 d\omega \tag{C.4}$$

$$= \int \|\omega\|^{2k} c_\eta (2\pi)^{2k} e^{-\pi^2 \|D^{-1/2}\omega\|^2} \frac{1}{c_\eta} e^{\pi^2 \|D^{-1/2}\omega\|^2} |\mathcal{F}[f](\omega)|^2 d\omega \tag{C.5}$$

$$\leq c_\eta (2\pi)^{2k} \sup_{t \in \mathbb{R}^d} \|t\|^{2k} e^{-\pi^2 \|D^{-1/2}t\|^2} \int \frac{1}{c_\eta} e^{\pi^2 \|D^{-1/2}\omega\|^2} |\mathcal{F}[f](\omega)|^2 d\omega \tag{C.6}$$

$$= \|f\|_{\mathcal{H}_\eta}^2 c_\eta (2\pi)^{2k} \sup_{t \in \mathbb{R}^d} \|t\|^{2k} e^{-\pi^2 \|D^{-1/2}t\|^2}. \tag{C.7}$$

Now, since $\sup_{z \in \mathbb{R}^d} \|z\|^{2k} e^{-\|z\|^2} = \sup_{r \geq 0} r^{2k} e^{-r^2} = k^k e^{-k} \leq k!$ and $\|D\| = \max_i \eta_i = \eta_+$,

$$\sup_{t \in \mathbb{R}^d} \|t\|^{2k} e^{-\pi^2 \|D^{-\frac{1}{2}}t\|^2} = \sup_{z \in \mathbb{R}^d} \|\tfrac{1}{\pi} D^{\frac{1}{2}} z\|^{2k} e^{-\|z\|^2} \leq \frac{\|D\|^k}{\pi^{2k}} \sup_{z \in \mathbb{R}^d} \|z\|^{2k} e^{-\|z\|^2} = \frac{\eta_+^k k!}{\pi^{2k}}. \tag{C.8}$$

Then,

$$\|f\|_{W_2^k(\mathbb{R}^d)} \leq \|f\|_{\mathcal{H}_\eta} c_\eta^{1/2} (4\eta_+)^{k/2} \sqrt{k!}. \tag{C.9}$$

By plugging the bound above in [(C.3)](#), we obtain that when $k, h_{\tilde{X},R}$ satisfy $kh_{\tilde{X},R} \leq RB_d'$ then,

$$\|f\|_{L^\infty(T)} \leq C \frac{(C_2 kh)^{k-d/2}}{\sqrt{k!}} \, \|f\|_{\mathcal{H}_\eta}, \tag{C.10}$$

where $C_2 = 2\sqrt{\eta_+}B_d$, and $C = (4B_d^2\pi)^{d/4}\det(\eta/\eta_+)^{-1/4}$. Let now $C_3 = 1/\max(B_d', 2B_d)$. Assume that $h \leq \frac{C_3}{2+d}\min(R, 1/\sqrt{\eta_+})$ and set $k = \lfloor s \rfloor$ and $s = \frac{C_3}{h}\min(R, 1/\sqrt{\eta_+})$. Note first, that with this choice of $h$ and $k$ we satisfy $hk \leq R/B_d'$ so we can apply Theorem 4.3 of [22]. Moreover, by construction $d/2 + 1 \leq \frac{s}{2} \leq s - 1 \leq \lfloor s \rfloor \leq s$. Then $C_2\lfloor s \rfloor h \leq 1$ and $\lfloor s \rfloor - d/2 \geq 0$, so $(C_2 h \lfloor s \rfloor)^{\lfloor s \rfloor - d/2} \leq 1$. Moreover $\frac{1}{\sqrt{k!}} \leq e^{-\frac{k}{2}\log\frac{k}{2}}$ so we have

$$\|f\|_{L^\infty(T)} \leq C(C_2 h \lfloor s \rfloor)^{\lfloor s \rfloor - d/2} e^{-\frac{1}{2}\lfloor s \rfloor \log \frac{1}{2}\lfloor s \rfloor} \|f\|_{\mathcal{H}_\eta} \tag{C.11}$$

$$\leq Ce^{-\frac{s-1}{2}\log\frac{s-1}{2}}\|f\|_{\mathcal{H}_\eta} \leq Ce^{-\frac{s}{4}\log\frac{s}{4}}\|f\|_{\mathcal{H}_\eta}. \tag{C.12}$$

The final result is obtained by writing $s/4 = c\sigma/h$ with $\sigma = \min(R, 1/\sqrt{\eta_+})$ and $c = C_3/4$ and by writing the assumption on $h$ as $h \leq \sigma/C'$ with $C' = (2+d)/C_3$. $\qquad\square$

**Lemma C.2** (Lemma 3, page 28 [20]). *Let $\mathcal{X} \subset \mathbb{R}^d$ with non-zero volume. Let $\mathcal{H}$ be a reproducing kernel Hilbert space on $\mathcal{X}$, associated to a continuous uniformly bounded feature map $\phi : \mathcal{X} \to \mathcal{H}$. Let $A : \mathcal{H} \to \mathcal{H}$ be a bounded linear operator. Then,*

$$\sup_{x \in \mathcal{X}} \|A\phi(x)\|_{\mathcal{H}} \leq \sup_{\|f\| \leq 1} \|A^* f\|_{C(\mathcal{X})}. \tag{C.13}$$

*In particular, if $\mathcal{X} \subset \mathbb{R}^d$ is a non-empty open set, then $\sup_{x \in \mathcal{X}} \|A\phi(x)\|_{\mathcal{H}} \leq \|A^* f\|_{L^\infty(\mathcal{X})}$.*

*Proof.* We recall the variational characterization of the norm $\|\cdot\|_{\mathcal{H}}$ in terms of the inner product $\langle\cdot,\cdot\rangle_{\mathcal{H}}$ as $\|v\|_{\mathcal{H}} = \sup_{\|f\| \leq 1}\langle f, v\rangle_{\mathcal{H}}$. We have the following

$$\sup_{x \in \mathcal{X}} \|A\phi(x)\|_{\mathcal{H}} = \sup_{x \in \mathcal{X}, \|f\| \leq 1} \langle f, A\phi(x)\rangle_{\mathcal{H}} \leq \sup_{x \in \mathcal{X}, \|f\| \leq 1} |\langle A^* f, \phi(x)\rangle_{\mathcal{H}}| \tag{C.14}$$

$$= \sup_{\|f\| \leq 1} \sup_{x \in \mathcal{X}} |(A^* f)(x)| = \sup_{\|f\| \leq 1} \|A^* f\|_{C(\mathcal{X})}. \tag{C.15}$$

Finally, note that when $\mathcal{X} \subset \mathbb{R}^d$ is a non-empty open set $\|A^* f\|_{C(\mathcal{X})} = \|A^* f\|_{L^\infty(\mathcal{X})}$, since $A^* f \in \mathcal{H}$ and all the functions in $\mathcal{H}$ are continuous and bounded due to the continuity of $\phi$. $\qquad\square$

**Theorem C.3** (Approximation properties of the projection). *Let $R > 0, \eta \in \mathbb{R}_{++}^d, m \in \mathbb{N}$. Let $\mathcal{X} \subseteq T = (-R, R)^d$ be a non-empty open set and let $\tilde{x}_1, \ldots, \tilde{x}_m$ be a set of distinct points. Let $h > 0$ be the fill distance associated to the points w.r.t $T$ (defined in (C.1)). Let $\tilde{P} : \mathcal{H}_\eta \to \mathcal{H}_\eta$ be the associated projection operator (see definition in (B.8)). There exists three constants $c, C, C'$, such that, when $h \leq \sigma/C'$,*

$$\sup_{x \in \mathcal{X}} \|(I - \tilde{P})\phi_\eta(x)\|_{\mathcal{H}_\eta} \leq Cq_\eta\, e^{-\frac{c\sigma}{h}\log\frac{c\sigma}{h}}. \tag{C.16}$$

*Here $q_\eta = \det(\frac{1}{\eta_+}\mathrm{diag}(\eta))^{-1/4}$ and $\sigma = \min(R, \frac{1}{\sqrt{\eta_+}})$, $\eta_+ = \max_i \eta_i$. The constants $c, C', C''$ depend only on $d$ and, in particular, are independent from $R, \eta, \tilde{x}_1, \ldots, \tilde{x}_m$.*

*Proof.* We first recall some basic properties of the projection operator $\tilde{P} : \mathcal{H}_\eta \to \mathcal{H}_\eta$ on the span of $\phi_\eta(\tilde{x}_1), \ldots, \phi_\eta(\tilde{x}_m)$, defined in (B.8). By construction $\tilde{P}\phi_\eta(\tilde{x}_i) = \phi_\eta(\tilde{x}_i)$ is of rank $m$ for any $i = 1, \ldots, m$. Now note that for any $f \in \mathcal{H}_\gamma$, the function $(\tilde{P}f)(\tilde{x}_i) = f(\tilde{x}_i)$, indeed, by the reproducing property of $\mathcal{H}_\eta$

$$(\tilde{P}u)(\tilde{x}_i) = \left\langle \tilde{P}u, \phi_\eta(\tilde{x}_i)\right\rangle_{\mathcal{H}_\eta} = \left\langle u, \tilde{P}\phi_\eta(\tilde{x}_i)\right\rangle_{\mathcal{H}_\eta} = \langle u, \phi_\eta(\tilde{x}_i)\rangle_{\mathcal{H}_\eta} = u(\tilde{x}_i). \tag{C.17}$$

Then $(f - \tilde{P}f)(\tilde{x}_i) = 0$ for any $i = 1, \ldots, m$. By Lemma C.1, we know that there exist three constants $c, C, C'$ depending only on $d$ such that when $h \leq \sigma/C'$ we have that the following holds $\|u\|_{L^\infty(T)} \leq Cq_\eta\, e^{-\frac{c\sigma}{h}\log\frac{c\sigma}{h}}$, for any $u \in \mathcal{H}_\eta$ such that $u(\tilde{x}_1) = \cdots = u(\tilde{x}_m) = 0$. Since, for any $f \in \mathcal{H}_\eta$, we have that $f - \tilde{P}f$ belongs to $\mathcal{H}_\eta$ and satisfies such property, we can apply Lemma C.1 with $u = (I - \tilde{P})f$, obtaining, under the same assumption on $h$,

$$\|(I - \tilde{P})f\|_{L^\infty(T)} \leq Cq_\eta\, e^{-\frac{c\sigma}{h}\log\frac{c\sigma}{h}}\|f\|_{\mathcal{H}_\eta}, \quad \forall f \in \mathcal{H}_\eta, \tag{C.18}$$

where we used the fact that $\|(I - \tilde{P})f\|_{\mathcal{H}_\eta} \leq \|I - \tilde{P}\|\|f\|_{\mathcal{H}_\eta}$ and $\|I - \tilde{P}\| \leq 1$, since $P$ is a projection operator and so also $I - P$ satisfies this property. The final result is obtained by applying Lemma C.2 with $A = I - \tilde{P}$, from which we have

$$\sup_{x \in \mathcal{X}} \|(I - \tilde{P})\phi(x)\|_{\mathcal{H}} \leq \sup_{\|f\| \leq 1} \|(I - \tilde{P})f\|_{L^\infty(T)} \leq \sup_{\|f\| \leq 1} Cq_\eta \, e^{-\frac{c\sigma}{h} \log \frac{c\sigma}{h}} \|f\|_{\mathcal{H}_\eta} \qquad \text{(C.19)}$$

$$= Cq_\eta \, e^{-\frac{c\sigma}{h} \log \frac{c\sigma}{h}}. \qquad \text{(C.20)}$$

$\square$

**Theorem C.4** (Compression of a PSD model). *Let $\eta \in \mathbb{R}^d_{++}$ and let $\mathsf{M} \in \mathbb{S}_+(\mathcal{H}_\eta)$. Let $\mathcal{X}$ be an open bounded subset with Lipschitz boundary of the cube $[-R, R]^d$, $R > 0$. Let $\tilde{x}_1, \ldots, \tilde{x}_m \in \mathcal{X}$ and $\tilde{X}_m$ be the base point matrix whose $j$-rows are the points $\tilde{x}_j$ with $j = 1, \ldots, m$. Consider the model $p = f(\,\cdot\,; \mathsf{M}, \phi_\eta)$ and the the compressed model $\tilde{p} = f(\,\cdot\,; A_m, \tilde{X}_m, \eta)$ with*

$$A_m = K^{-1}_{\tilde{X}, \tilde{X}, \eta} \, \tilde{Z}\mathsf{M}\tilde{Z}^* \, K^{-1}_{\tilde{X}, \tilde{X}, \eta}, \qquad \text{(C.21)}$$

*where $\tilde{Z} : \mathcal{H}_\eta \to \mathbb{R}^m$ is defined in (B.6) in terms of $\tilde{X}_m$. Let $h$ be the fill distance (defined in (C.1)) associated to the points $\tilde{x}_1, \ldots, \tilde{x}_m$. The there exist three constants $c, C, C'$ depending only on $d$ such that, when $h \leq \sigma/C'$, with $\sigma = \min(R, 1/\sqrt{\eta_+}), \eta_+ = \max_{i=1,\ldots,d} \eta_i, q_\eta = \det(\frac{1}{\eta_+}\mathrm{diag}(\eta))^{-1/4}$, then*

$$|p(x) - \tilde{p}(x)| \leq 2Cq_\eta \sqrt{\|\mathsf{M}\|p(x)} \, e^{-\frac{c\sigma}{h} \log \frac{c\sigma}{h}} + C^2 q_\eta^2 \|\mathsf{M}\| \, e^{-\frac{2c\sigma}{h} \log \frac{c\sigma}{h}}, \quad \forall x \in \mathcal{X}. \quad \text{(C.22)}$$

*Proof.* Let $\varepsilon \in (0, 1/e]$ and $\eta = \tau \mathbf{1}_d \in \mathbb{R}^d$ with $\tau > 0$. Consider the projection operator $\tilde{P} : \mathcal{H}_\eta \to \mathcal{H}_\eta$ associated to the points $\tilde{x}_1, \ldots, \tilde{x}_m$, defined in (B.8). Note that the adjoint $\tilde{Z}^*$ has range equal to $\mathrm{span}\{\phi_\eta(\tilde{x}_1), \ldots, \phi_\eta(\tilde{x}_1)\}$ and that by construction $\tilde{Z} \, \mathsf{M}_\varepsilon \tilde{Z}^* \in \mathbb{S}_+^m$ and so $A_m \in \mathbb{S}_+^m$.

**Step 1. Error induced by a projection.** By the reproducing property $k_\eta(x, x') = \phi_\eta(x)^\top \phi_\eta(x')$ (see Example 1 and (A.2)) and the fact that $\tilde{P} = \tilde{Z}^* K^{-1}_{\tilde{X}, \tilde{X}, \eta} \tilde{Z}$, (see (B.8)), then, for any $x \in \mathbb{R}^d$

$$f(x \,; A_m, \tilde{X}_m, \eta) = \sum_{i,j=1}^m (A_m)_{i,j} k_\eta(x, \tilde{x}_i) k_\eta(x, \tilde{x}_j) \qquad \text{(C.23)}$$

$$= \phi_\eta(x)^\top \Big( \sum_{i,j=1}^m (A_m)_{i,j} \phi_\eta(\tilde{x}_i) \phi_\eta(\tilde{x}_j)^\top \Big) \phi_\eta(x) \qquad \text{(C.24)}$$

$$= \phi_\eta(x)^\top \tilde{Z}^* A_m \tilde{Z} \phi_\eta(x) \qquad \text{(C.25)}$$

$$= \phi_\eta(x)^\top \tilde{Z}^* K^{-1}_{\tilde{X}, \tilde{X}, \eta} \tilde{Z} \, \mathsf{M}_\varepsilon \, \tilde{Z}^* K^{-1}_{\tilde{X}, \tilde{X}, \eta} \tilde{Z} \phi_\eta(x) \qquad \text{(C.26)}$$

$$= \phi_\eta(x)^\top \tilde{P}\mathsf{M}\tilde{P}\phi_\eta(x) \qquad \text{(C.27)}$$

$$= f(x \,; \tilde{P}\mathsf{M}\tilde{P}, \phi_\eta). \qquad \text{(C.28)}$$

This implies that, for all $x \in \mathbb{R}^d$ the following holds

$$f(x \,; A_m, \tilde{X}_m, \eta) - f(x \,; \mathsf{M}, \phi_\eta) = f(x \,; \tilde{P}\mathsf{M}\tilde{P}, \phi_\eta) - f(x \,; \mathsf{M}, \phi_\eta) \qquad \text{(C.29)}$$

$$= \phi_\eta(x)^\top (\tilde{P}\mathsf{M}\tilde{P} - \mathsf{M}_\varepsilon)\phi_\eta(x). \qquad \text{(C.30)}$$

**Step 2. Bounding** $|\phi_\eta(x)^\top (\tilde{P}\mathsf{M}_\varepsilon \tilde{P} - \mathsf{M})\phi_\eta(x)|$**.** Now, consider that

$$\tilde{P}\mathsf{M}\tilde{P} - \mathsf{M}_\varepsilon = \mathsf{M}(I - \tilde{P}) + (I - \tilde{P})\mathsf{M} + (I - \tilde{P})\mathsf{M}(I - \tilde{P}). \qquad \text{(C.31)}$$

Since $|a^\top ABAa| \leq \|Aa\|_{\mathcal{H}}^2 \|B\|$ and $a^\top ABa \leq \|Aa\|_{\mathcal{H}} \|B^{1/2}\|\|B^{1/2}a\|_{\mathcal{H}}$, for any $a$ in a Hilbert space $\mathcal{H}$ and for $A, B$ bounded symmetric linear operators with $B \in \mathbb{S}_+(\mathcal{H})$, by bounding the terms of the equation above, we have for any $x \in \mathbb{R}^d$,

$$|\phi_\eta(x)^\top (\tilde{P}\mathsf{M}\tilde{P} - \mathsf{M})\phi_\eta(x)| \leq 2\|(I - \tilde{P})\phi_\eta(x)\|_{\mathcal{H}_\eta} \|\mathsf{M}\|^{1/2} \|\mathsf{M}^{1/2}\phi_\eta(x)\|_{\mathcal{H}_\eta} \qquad \text{(C.32)}$$

$$+ \|(I - \tilde{P})\phi_\eta(x)\|_{\mathcal{H}_\eta}^2 \|\mathsf{M}\| \qquad \text{(C.33)}$$

$$= 2c_\mathsf{M}^{1/2} f(x \,; \mathsf{M}, \phi_\eta)^{1/2} Ce^{-\frac{c\sigma}{h} \log \frac{c\sigma}{h}} + c_\mathsf{M}\tilde{u}(x)^2, \qquad \text{(C.34)}$$

where $c_{\mathsf{M}} = \|\mathsf{M}\|$ and we denoted by $\tilde{u}(x)$ the quantity $\tilde{u}(x) = \|(I - \tilde{P})\phi_\eta(x)\|_{\mathcal{H}_\eta}$ and we noted that $\|\mathsf{M}^{1/2}\phi_\eta(x)\|_{\mathcal{H}_\eta}^2 = \phi_\eta(x)^\top \mathsf{M}\phi_\eta(x) = f(x\,;\,\mathsf{M},\phi_\eta)$.

**Step 3. Bounding $\tilde{u}$.** Now, by Thm. C.3 we have that when the fill distance $h$ (defined in (C.1)) satisfies $h \leq \sigma/C'$ with $\sigma = \min(R, 1/\sqrt{\tau})$, then

$$\|\tilde{u}\|_{L^\infty(\mathcal{X})} = \sup_{x \in \mathcal{X}} \|(I - \tilde{P})\phi_\eta(x)\|_{\mathcal{H}_\eta} \leq Ce^{-\frac{c\sigma}{h}\log\frac{c\sigma}{h}}. \tag{C.35}$$

with $c, C, C'$ depending only on $d$. $\qquad\qquad\qquad\qquad\qquad\qquad\qquad\qquad\qquad\qquad\qquad \Box$

## C.1 Proof of Thm. 8

Thm. 8 is a corollary of the next theorem, considering that $1/\sigma \leq (1 + \eta_+)^{1/2}$ and moreover $\det(\frac{1}{\eta_+}\mathrm{diag}(\eta)) = \prod_{j=1}^d \eta_i/\eta_+ \leq 1$, $\mathrm{Tr}(AK_{X,X,\eta}) \leq \|A\|\mathrm{Tr}(K_{X,X,\eta})$ since both $A, K_{X,X,\eta} \in \mathbb{S}_+^n$, and by construction $\mathrm{Tr}(K_{X,X,\eta}) = n$, then $c_{A,\eta} \leq \|A\|n$.

**Theorem C.5.** *Let $A \in \mathbb{S}_+^n$, $X \in \mathbb{R}^{n \times d}$, $\eta \in \mathbb{R}_{++}^d$. Let $\tilde{x}_1, \ldots, \tilde{x}_m$ be sampled independently and uniformly at random from $[-1,1]^d$. Let $\delta \in (0,1]$, $\eta_+ = \max(1, \max_{i=1,\ldots,d} \eta_i)$. When $m$ satisfies $m \geq Q'\eta_+^{d/2}(\log\frac{Q\|A\|n}{\varepsilon})^d \log(\frac{Q''(1+\eta_+)}{\delta}\log\frac{Q\|A\|n}{\varepsilon})$, then the following holds with probability at least $1 - \delta$,*

$$|p(x) - \tilde{p}(x)| \leq \varepsilon^2 + \varepsilon\sqrt{p(x)}, \qquad \forall x \in [-1,1]^d, \tag{C.36}$$

*Here the three constants $Q, Q', Q''$ depend only on $d$.*

*Proof.* First let us rewrite $f(\cdot\,;\,A, X, \eta)$ in the equivalent form $f(\cdot\,;\,\mathsf{M}, \phi_\eta)$ with $\mathsf{M} \in \mathbb{S}_+(\mathcal{H}_\eta)$ defined as $\mathsf{M} = \sum_{ij=1}^n A_{ij}\phi_\eta(x_i)\phi_\eta(x_j)$. In particular, by the cyclicity of the trace

$$\mathrm{Tr}(\mathsf{M}) = \sum_{ij=1}^n A_{i,j}\phi_\eta(x_i)^\top \phi_\eta(x_j) = \sum_{ij=1}^n A_{i,j}k_\eta(x_i, x_j) = \mathrm{Tr}(AK_{X,X,\eta}). \tag{C.37}$$

The proof of this theorem is an application of the approximation result in Thm. C.4 to the model $f(\cdot\,;\,\mathsf{M}, \phi_\eta)$ where we use as compression points, the points $\tilde{x}_1, \ldots, \tilde{x}_m$ sampled independently and uniformly at random from $[-1,1]^d$.

The result of the theorem depends on the fill distance $h$, defined in (C.1), and associated to the points $\tilde{x}_1, \ldots, \tilde{x}_m$. Let $c, C, C'$ be the constants depending only on $d$ from Thm. C.4. To apply Thm. C.4 we have to guarantee that $h \leq \sigma/C'$ with $\sigma = \min(1, 1/\sqrt{\eta_+})$, in particular, choosing $h$ such that $h \leq \min(c, 1/C')\sigma/(e\log(2Cc_{A,\eta}/\varepsilon))$ guarantees that $h \leq \sigma/C'$ and by applying the theorem we have, for all $x \in [-1,1]^d$

$$|p(x) - \tilde{p}(x)| \leq 2Cq_\eta\sqrt{\|\mathsf{M}\|p(x)}\,e^{-\frac{c\sigma}{h}\log\frac{c\sigma}{h}} + Cq_\eta^2\|\mathsf{M}\|\,e^{-\frac{2c\sigma}{h}\log\frac{c\sigma}{h}}, \tag{C.38}$$

$$\leq 2Cc_{A,\eta}\sqrt{p(x)}\,e^{-\frac{c\sigma}{h}\log\frac{c\sigma}{h}} + C^2c_{A,\eta}^2\,e^{-\frac{2c\sigma}{h}\log\frac{c\sigma}{h}}, \tag{C.39}$$

with $q_\eta = \det(\frac{1}{\eta_+}\mathrm{diag}(\eta))^{-1/4}$ where in the last step we used the fact that $\|\mathsf{M}\| \leq \mathrm{Tr}(\mathsf{M}) = \mathrm{Tr}(AK_{X,X,\eta})$ and so $\|\mathsf{M}\|q_\eta^2 \leq \mathrm{Tr}(AK_{X,X,\eta})q_\eta^2 = c_{A,\eta}^2$. Note now, that by the choice we made for $h$, we have $\log(c\sigma/h) \geq 1$ and so that $e^{-\frac{c\sigma}{h}\log\frac{c\sigma}{h}} \leq \varepsilon/(2Cc_{A,\eta})$. This implies

$$|p(x) - \tilde{p}(x)| \leq \varepsilon\sqrt{p(x)} + \varepsilon^2. \tag{C.40}$$

The final result is obtained by controlling the number of points $m$ such that $h$ satisfy the required bound in high probability. By, e.g. Lemma 12, page 19 of [37] and the fact that $[-1,1]^d$ is a convex set, we have that there exists two constants $C_1, C_2$ depending only on $d$ such that $h \leq C_1 m^{-1/d}(\log(C_2 m/\delta))^{1/d}$, with probability at least $1 - \delta$. In particular $m$ satisfying

$$m \geq \left(\frac{eC_1}{\min(c, 1/C')}\frac{1}{\sigma}\log\frac{2Cc_{A,\varepsilon}}{\varepsilon}\right)^d \log\frac{C_2 m}{\delta} \tag{C.41}$$

guarantees that $C_1 m^{-1/d}(\log \frac{C_2 m}{\delta})^{1/d} \leq \frac{\sigma \min(c,1/C')}{e} \log \frac{2Cc_{A,\eta}}{\varepsilon}$. Note that, given $A \geq e, B \geq e$, the inequality $m \geq B\log(Am)$ is satisfied by $m \geq 2B\log 2AB$, indeed $\log A \geq \log \log A$ and $\log 2B \geq \log \log 2B$ and so, when $m = m_0 = 2B\log 2AB$ we have

$$B\log(Am_0) = B\log(2AB\log(2AB)) \tag{C.42}$$

$$= B\log A + B\log 2B + B\log \log A + B\log \log 2B \tag{C.43}$$

$$\leq 2B\log A + 2B\log 2B = 2B\log 2AB = m_0, \tag{C.44}$$

and moreover $m - B\log(Am)$ is increasing for $m \geq B$. Then, to satisfy (C.41) we choose $m \geq 2B\log(2AB)$ with $B = \left(\frac{eC_1}{\min(c,1/C')} \frac{1}{\sigma} \log \frac{2Cc_{A,\eta}}{\varepsilon}\right)^d$ and $A = \frac{C_2}{\delta}$, in particular

$$m = Q\left(\frac{1}{\sigma} \log \frac{Q'c_{A,\eta}}{\varepsilon}\right)^d \log \left(\frac{Q''}{\delta\sigma} \log \frac{Q'c_{A,\eta}}{\varepsilon}\right). \tag{C.45}$$

with $Q = 2d(\frac{eC_1}{\min(c,1/C')})^d$, with $Q' = 2C$, $Q'' = 2C_2 Q^d$. $\qquad\square$

# D  Approximation of a probability via a PSD model

In this section we prove Prop. 5 and Thm. 6.

## D.1  Proof of Prop. 5

**Lemma D.1.** *Let $\beta > 0$ and $\tilde{p}$ be a continuous strictly positive function such that $\tilde{p} \in W_2^\beta(\mathcal{X}) \cap L^\infty(\mathcal{X})$, with $\beta > 0$, be a function such that $p > 0$. Let $\mathcal{X}$ be a compact set with Lipschitz boundary. Then there exist a function in $\tilde{f} \in W_2^\beta(\mathcal{X}) \cap L^\infty(\mathcal{X})$ such that $p(x) = f(x)^2$ for all $x \in \mathcal{X}$.*

*Proof.* First, let $\tilde{p} \in W_2^\beta(\mathbb{R}^d) \cap L^\infty(\mathbb{R}^d)$ be the extension of $p$ on all $\mathbb{R}^d$ (see Cor. A.3), i.e. $\tilde{p} = p$ on $\mathcal{X}$ and $\|\tilde{p}\|_{W_2^\beta(\mathbb{R}^d)} \leq C\|p\|_{W_2^\beta(\mathcal{X})}$ for a fixed constant $C$. Let $c = \inf_{x \in \mathcal{X}} p(x)$ and $C = \sup_{x \in \mathcal{X}} p(x)$, we have that $c > 0$ since $\mathcal{X}$ is compact and $p$ is continuous. Note that $g(z) = \sqrt{z}$ is $C^\infty$ on the open interval $(0, +\infty)$. Let $u \in C^\infty(\mathbb{R})$ be the bump function that is identically 0 on $J = (\infty, c/2] \cup [2C, \infty)$ and identically 1 on the interval $I = [c, C]$. Then the function $h(z) = u(z)g(z)$ is identically 0 on $J$ and $h(z) = \sqrt{z}$ on $I$ and $h \in C^\infty(\mathbb{R})$ since $h = 0$ on $J$ and both $u, g \in C^\infty([c/2, 2C])$. Now, denote by $f$ the function $f(x) = h(p(x))$ for all $x \in \mathbb{R}^d$. Since $p(x) \in I$ for any $x \in \mathcal{X}$, we have $f(x) = h(p(x)) = \sqrt{p(x)}$ for any $x \in \mathcal{X}$. Moreover, by Theorem 1, page 8, in [28], $f \in W_2^\beta(\mathbb{R}^d)$, since $h \in C^\infty(\mathbb{R}) \subset C^{\beta+1}(\mathbb{R})$ and $p \in W_2^\beta(\mathbb{R}^d) \cap L^\infty(\mathbb{R}^d)$. The fact that $f \in L^\infty(\mathbb{R}^d)$ derive from the fact that it is the composition of a compactly supported smooth function $h$ and an $L^\infty(\mathbb{R}^d)$ function $p$. The proof is concluded by taking $\tilde{f}$ to be the restriction of $f$ to $\mathcal{X}$. $\qquad\square$

**Lemma D.2** ([24] Corollary 2, page 23). *Let $\mathcal{X}$ be an open bounded subset of $\mathbb{R}^d$ with Lipschitz boundary. Let $p$ be a probability density that is $\beta + 2$-times differentiable on the closure of $\mathcal{X}$, with $\beta > 0$. The zeros of $p$ are isolated points with strictly positive Hessian and their number is finite. Moreover there are no zeros of $p$ on the boundary. Then there exist $q \in \mathbb{N}$ and $q$ functions $f_1 \ldots f_q \in W^\beta(\mathcal{X}) \cap L^\infty(\mathcal{X})$, such that*

$$p(x) = \sum_{i=1}^q f_i(x)^2, \quad \forall x \in \mathcal{X}. \tag{D.1}$$

*Proof.* Let $\tilde{p}$ be the $\beta + 2$-times differentiable extension to $\mathbb{R}^d$ of $p$ (via the Withney extension theorem [12]), i.e. $\tilde{p} = p$ on the closure of $\mathcal{X}$. We apply [24] Corollary 2, page 23 on $\tilde{p}$, obtaining $q$ functions $f_1, \ldots, f_q \in C^\beta(\mathbb{R}^d)$ such that $\tilde{p}(x) = \sum_i f_i(x)^2$ for all $x \in \mathcal{X}$. The result is obtained by applying Prop. A.4 on the restrictions $f_1, \ldots, f_q$ on $\mathcal{X}$. $\qquad\square$

Now we are ready to prove Prop. 5. We restate here fore convenience.

**Proposition 5** (Generality of Assumption 1). *The assumption above is satisfied by*

(a) *any probability density $p \in W_2^\beta(\mathcal{X}) \cap L^\infty(\mathcal{X})$ and strictly positive on $[-1,1]^d$,*

(b) *any exponential model $p(x) = e^{-v(x)}$ with $v \in W_2^\beta(\mathcal{X}) \cap L^\infty(\mathcal{X})$,*

(c) *any mixture model of Gaussians or, more generally, of exponential models from (b),*

(d) *any $p$ that is $\beta + 2$-times differentiable on $[-1,1]^d$, with a finite set of zeros, all in $(-1,1)^d$, and with positive definite Hessian in each zero. E.g. $p(x) \propto x^2 e^{-x^2}$.*

*Moreover if $p$ is $\beta$-times differentiable over $[-1,1]^d$, then it belongs to $W_2^\beta(\mathcal{X}) \cap L^\infty(\mathcal{X})$.*

*Proof.* Let $\mathcal{X} = (-1,1)^d$. The case (a) is proven in Lemma D.1. Note that (b) and (c) are a subcase of (a). For the case (b), let $\tilde{v} \in W_2^\beta(\mathbb{R}^d) \cap L^\infty(\mathbb{R}^d)$ be the extension of $v$ (see Cor. A.3), by Theorem 1, page 8, in [28] the function $e^{\tilde{v}} - 1 \in W_2^\beta(\mathbb{R}^d) \cap L^\infty(\mathbb{R}^d)$ since $\exp(\cdot) - 1$ is analytic and 0 in 0. Let $q = (e^{\tilde{v}} - 1)|_{\mathcal{X}}$, $q \in W_2^\beta(\mathcal{X}) \cap L^\infty(\mathcal{X})$ and so also $g = q + 1$ belongs to $W_2^\beta(\mathcal{X}) \cap L^\infty(\mathcal{X})$, since $\mathcal{X}$ is a bounded set. Finally note that $g = e^v$ on $\mathcal{X}$ and $\min_{x \in \mathcal{X}} g(x) = \min_{x \in \mathcal{X}} e^{-v(x)} \geq e^{-\|v\|_{L^\infty(\mathcal{X})}} > 0$, so it satisfies the point (a). The point (c) is a consequence of (b) indeed if $p = \sum_{i=1}^t \alpha_i e^{-v_i}$ and each $v_i$ satisfies (b), then $e^{-v_i} = \sum_{j=1}^{q_i} f_{i,j}^2$ with $f_{i,j} \in W_2^\beta(\mathbb{R}^d) \cap L^\infty(\mathbb{R}^d)$, so $p = \sum_{i=1}^t \sum_{j=1}^{q_i} g_{i,j}^2$, with $g_{i,j} = \sqrt{\alpha_i} f_{i,j} \in W_2^\beta(\mathbb{R}^d) \cap L^\infty(\mathbb{R}^d)$. Finally, (d) is proven in Lemma D.2. $\square$

### D.2 Additional results required to prove Thm. 6

We now focus on proving the result in Thm. 6. To this end, we first prove some preliminary result that will be useful in the following.

Let $S \subseteq \mathbb{R}^d$. We recall the definition of the function $\mathbf{1}_S$, that is $\mathbf{1}_S(\omega) = 1$ for any $\omega \in S$ and $\mathbf{1}_S(\omega) = 0$ for any $\omega \notin S$. Define moreover,

$$g(x) = \tfrac{1}{V_d} \|x\|^{-d} J_{d/2}(2\pi\|x\|) J_{d/2}(4\pi\|x\|) \tag{D.2}$$

where $J_{d/2}$ is the Bessel function of the first kind of order $d/2$ and $V_d = \int_{\|x\| \leq 1} dx = \frac{\pi^{d/2}}{\Gamma(d/2+1)}$.

**Lemma D.3.** *The function $g$ defined above satisfies $g \in L^1(\mathbb{R}^d) \cap L^2(\mathbb{R}^d)$ and $\int g(x)dx = 1$. Moreover, for any $\omega \in \mathbb{R}^d$, we have*

$$\mathbf{1}_{\{\|\omega\| < 1\}}(\omega) \leq \mathcal{F}[g](\omega) \leq \mathbf{1}_{\{\|\omega\| \leq 3\}}(\omega). \tag{D.3}$$

*Proof.* In this proof we will use the notation about the convolution and the Fourier transform in Prop. A.1. Define $b(x) = \|x\|^{-d/2} J_{d/2}(2\pi\|x\|)$ where $J_{d/2}$ is the Bessel function of the first kind of order $d/2$. Note that $b \in L^2(\mathbb{R}^d) \cap L^1(\mathbb{R}^d)$, since there exists a constant $c > 0$ $|J_{d/2}(z)| \leq c \min(z^{d/2}, z^{-1/2})$ for any $z \geq 0$ [33]. Moreover note that the Fourier transform of $b$ is $\mathcal{F}[b](\omega) = \mathbf{1}_{\{\|\omega\| < 1\}}$ (see [33], Thm. 4.15, page 171). Define now $g(x) = \frac{1}{V_d} b(x) b(2x) = \frac{1}{V_d} \|x\|^{-d} J_{d/2}(2\pi\|x\|) J_{d/2}(4\pi\|x\|)$. Note that $g \in L^1(\mathbb{R}^d)$ since

$$\|g\|_{L^1(\mathbb{R}^d)} = \|b(\cdot)b(2\cdot)\|_{L^1(\mathbb{R}^d)} \leq \|b(\cdot)\|_{L^2(\mathbb{R}^d)} \|b(2\cdot)\|_{L^2(\mathbb{R}^d)} < \infty, \tag{D.4}$$

and analogously

$$\|g\|_{L^2(\mathbb{R}^d)} = \|b(\cdot)b(2\cdot)\|_{L^2(\mathbb{R}^d)} \leq \|b(\cdot)\|_{L^\infty(\mathbb{R}^d)} \|b(2\cdot)\|_{L^2(\mathbb{R}^d)} < \infty. \tag{D.5}$$

By the properties of the Fourier transform, we have $\mathcal{F}[g] = \frac{1}{V_d} \mathcal{F}[b] \star \mathcal{F}[b(2\cdot)] = \frac{1}{V_d} \int \mathbf{1}_{\{\|z\| \leq 1\}} \mathbf{1}_{\{\|\omega - z\| \leq 2\}} dz$. Note that for any $\omega \in \mathbb{R}^d$, since $\mathbf{1}_{\{\|\omega - z\| \leq 2\}} \leq 1$,

$$\mathcal{F}[g](\omega) = \frac{1}{V_d} \int \mathbf{1}_{\{\|z\| \leq 1\}} \mathbf{1}_{\{\|\omega - z\| \leq 2\}} dz \leq \frac{1}{V_d} \int \mathbf{1}_{\{\|z\| \leq 1\}} dz \leq 1. \tag{D.6}$$

Now, note that when $\|\omega\|, \|z\| \leq 1$, then $\|\omega - z\| \leq \|\omega\| + \|z\| \leq 2$. So we have $\mathbf{1}_{\{\|\omega\| \leq 1\}} \mathbf{1}_{\{\|z\| \leq 1\}} \mathbf{1}_{\{\|\omega - z\| \leq 2\}} = \mathbf{1}_{\{\|z\| \leq 1\}}$. Then

$$\mathbf{1}_{\{\|\omega\| \leq 1\}} \mathcal{F}[g](\omega) = \frac{1}{V_d} \int \mathbf{1}_{\{\|\omega\| \leq 1\}} \mathbf{1}_{\{\|z\| \leq 1\}} \mathbf{1}_{\{\|\omega - z\| \leq 2\}} dz \tag{D.7}$$

$$= \frac{1}{V_d} \int \mathbf{1}_{\{\|z\| \leq 1\}} dz = \mathbf{1}_{\{\|\omega\| \leq 1\}}. \tag{D.8}$$

Moreover note that for all $\|\omega\| > 3, \|z\| \leq 1$ we have $\|\omega - z\| \geq |\|\omega\| - \|z\|| > 2$, then $\mathbf{1}_{\{\|\omega\|>3\}}\mathbf{1}_{\{\|z\|\leq1\}}\mathbf{1}_{\{\|\omega-z\|\leq2\}} = 0$. So for any $\|\omega\| > 3$

$$\mathbf{1}_{\{\|\omega\|>3\}}\mathcal{F}[g](\omega) = \frac{1}{V_d}\int \mathbf{1}_{\{\|\omega\|>3\}}\mathbf{1}_{\{\|z\|\leq1\}}\mathbf{1}_{\{\|\omega-z\|\leq2\}}dz = 0. \tag{D.9}$$

To conclude $\int g(x)dx = \int g(x)e^{-2\pi iw^\top 0}dx = \mathcal{F}[g](0) = 1$. $\qquad\square$

**Theorem D.4.** *Let $\beta > 0, q \in \mathbb{N}$. Let $f_1, \ldots, f_q \in W_2^\beta(\mathbb{R}^d) \cap L^\infty(\mathbb{R}^d)$ and denote by $p$ the function $p = \sum_{i=1}^q f_i^2$. Let $\varepsilon \in (0,1]$ and let $\eta \in \mathbb{R}_{++}^d$. Denote by $\eta_0 = \min_{j=1,\ldots,d} \eta_j$. Let $\phi_\eta$ be the feature map of the Gaussian kernel with bandwidth $\eta$ and let $\mathcal{H}_\eta$ be the associated RKHS. Then there exists $\mathsf{M}_\varepsilon \in \mathbb{S}_+(\mathcal{H}_\eta)$ with $\mathrm{rank}(\mathsf{M}_\varepsilon) \leq q$, such that*

$$\|f(\cdot\,;\mathsf{M}_\varepsilon,\phi_\eta) - p(\cdot)\|_{L^r(\mathbb{R}^d)} \leq \varepsilon, \qquad \mathrm{Tr}(\mathsf{M}_\varepsilon) \leq C|\eta|^{1/2}(1 + \varepsilon^2\exp(\tfrac{C'}{\eta_0}\varepsilon^{-\frac{2}{\beta}})), \tag{D.10}$$

*for all $r \in [1,2]$, where $|\eta| = \det(\mathrm{diag}(\eta))$ and $C, C'$ depend only on $\beta, d, \|f_i\|_{W_2^\beta(\mathbb{R}^d)}, \|f_i\|_{L^\infty(\mathbb{R}^d)}$.*

*Proof.* Let $t > 0$ (to be set later) and let $g$ be defined according to (D.2). Define $g_t(x) = t^{-d}g(x/t)$. Given the properties of $g$ in Lemma D.3, we have that $\int g_t(x)dx = 1$, $g_t \in L^1(\mathbb{R}^d) \cap L^2(\mathbb{R}^d)$, that $\mathcal{F}[g_t](\omega) = \mathcal{F}[g](t\omega)$ and so that $|\mathcal{F}[g_t](\omega)| = |\mathcal{F}[g](t\omega)| \leq \mathbf{1}_{\{t\|\omega\|\leq3\}}(\omega)$. Moreover we have that $|1 - \mathcal{F}[g_t](\omega)| = |1 - \mathcal{F}[g](t\omega)| \leq \mathbf{1}_{\{t\|\omega\|\geq1\}}(\omega)$.

Now, note that $\int(1 + \|\omega\|^2)^\beta|\mathcal{F}[f](\omega)|^2d\omega \leq 2^{2\beta}\|f\|_{W_2^\beta(\mathbb{R}^d)}^2$, as discussed in Prop. A.5.

**Step 1. Bounding $\|f - f \star g_t\|_{L^2(\mathbb{R}^d)}$.**
Since, we have seen that $|1 - \mathcal{F}[g_t](\omega)| \leq \mathbf{1}_{t\|\omega\|\geq1}$, then for any $f \in W_2^\beta(\mathbb{R}^d)$ we have

$$\|f - f \star g_t\|_{L^2(\mathbb{R}^d)}^2 = \|\mathcal{F}[f] - \mathcal{F}[f \star g_t]\|_{L^2(\mathbb{R}^d)}^2 = \|\mathcal{F}[f](1 - \mathcal{F}[g_t])\|_{L^2(\mathbb{R}^d)}^2 \tag{D.11}$$

$$= \int |1 - \mathcal{F}[g](t\omega)|^2|\mathcal{F}[f](\omega)|^2d\omega \leq \int_{t\|\omega\|\geq1}\mathcal{F}[f](\omega)^2d\omega \tag{D.12}$$

$$= \int_{t\|\omega\|\geq1}(1 + \|\omega\|^2)^{-\beta}(1 + \|\omega\|^2)^\beta|\mathcal{F}[f](\omega)|^2d\omega \tag{D.13}$$

$$\leq \sup_{t\|\omega\|\geq1}(1 + \|\omega\|^2)^{-\beta}\int(1 + \|\omega\|^2)^\beta|\mathcal{F}[f](\omega)|^2d\omega \tag{D.14}$$

$$= 2^{2\beta}\|f\|_{W_2^\beta(\mathbb{R}^d)}^2\frac{(2t)^{2\beta}}{(1 + t^2)^\beta} \leq \|f\|_{W_2^\beta(\mathbb{R}^d)}^2(2t)^{2\beta}. \tag{D.15}$$

**Step 2. Bounding $\|f \star g_t\|_{\mathcal{H}_\eta}$.**
However, the function $f \star g_t$ belongs to $\mathcal{H}_\eta$ for any $\eta \in \mathbb{R}_{++}^d$. Indeed, as discussed in Example 1, we have that $\|u\|_{\mathcal{H}_\eta}$ is characterized as

$$\|u\|_{\mathcal{H}_\eta}^2 = c_\eta\int|\mathcal{F}[u](\omega)|^2e^{\pi^2\omega^\top\mathrm{diag}(\eta)^{-1}\omega}d\omega,$$

and $u \in \mathcal{H}_\eta$ iff $\|u\|_{\mathcal{H}_\eta} < \infty$, with $c_\eta = \pi^{-d/2}\det(\mathrm{diag}(\eta))^{1/2}$. Now, let $\eta_0 = \min_{i=1..d}\eta_i$, since we have seen that $|\mathcal{F}[g_t](\omega)| \leq \mathbf{1}_{t\|\omega\|\leq3}(\omega)$, then we have that

$$\|f\star g_t\|_{\mathcal{H}_\eta}^2 = c_\eta\int|\mathcal{F}[f](\omega)\mathcal{F}[g](t\omega)|^2e^{\omega^\top\mathrm{diag}(\eta)^{-1}\omega}d\omega \tag{D.16}$$

$$\leq c_\eta\int|\mathcal{F}[f](\omega)\mathcal{F}[g](t\omega)|^2e^{\frac{\pi^2}{\eta_0}\|\omega\|^2}d\omega = c_\eta\int_{t\|\omega\|\leq3}|\mathcal{F}[f](\omega)|^2e^{\frac{4}{\eta_0}\|\omega\|^2}d\omega \tag{D.17}$$

$$= c_\eta\int_{t\|\omega\|\leq3}|\mathcal{F}[f](\omega)|^2(1 + \|\omega\|^2)^\beta\frac{e^{\frac{4}{\eta_0}\|\omega\|^2}}{(1 + \|\omega\|^2)^\beta}d\omega \tag{D.18}$$

$$\leq c_\eta\sup_{t\|\omega\|\leq3}\frac{e^{\frac{4}{\eta_0}\|\omega\|^2}}{(1+\|\omega\|^2)^\beta}\int|\mathcal{F}[f](\omega)|^2(1 + \|\omega\|^2)^\beta d\omega \tag{D.19}$$

$$\leq \|f\|_{W_2^\beta(\mathbb{R}^d)}^2c_\eta2^{2\beta}\sup_{r\leq3/t}\frac{e^{r^2\pi^2/\eta_0}}{(1+r^2)^\beta} \tag{D.20}$$

**Step 3. Bounding** $\mathrm{Tr}(\mathsf{M}_\varepsilon)$**.**

Note that the function $\frac{1}{(1+r^2)^\beta}\exp(\frac{r^2\pi^2}{\eta_0})$ has only one critical point in $r$ that is a minimum, then $\sup_{r\le 3/t}\frac{1}{(1+r^2)^\beta}\exp(\frac{r^2\pi^2}{\eta_0}) \le \max[1,\frac{t^{2\beta}}{(t^2+9)^\beta}\exp(\frac{9\pi^2}{\eta_0 t^2})] \le 1 + (t/3)^{2\beta}\exp(\frac{89}{\eta_0 t^2})$. Now let consider the functions $f_{i,t} = f_i \star g_t$ for $i \in \{1,\dots,q\}$ and note that, by the results above $\|f_{i,t}-f_i\|_{L^2(\mathbb{R}^d)} \le \|f_i\|_{W_2^\beta(\mathbb{R}^d)}(2t)^\beta$ and $\|f_{i,t}\|_{\mathcal{H}_\eta}^2 \le \|f_i\|_{W_2^\beta(\mathbb{R}^d)}^2 c_\eta 2^{2\beta}(1+(t/3)^{2\beta}\exp(\frac{89}{\eta_0 t^2}))$. Since $f_{i,t}$ belong to the reproducing kernel Hilbert space $\mathcal{H}_\eta$, define the operator $\mathsf{M}_\varepsilon$ as

$$\mathsf{M}_\varepsilon = \sum_{i=1}^q f_{i,t} f_{i,t}^\top.$$

First note that $\mathsf{M}_\varepsilon \in \mathbb{S}_+(\mathcal{H}_\eta)$, moreover $\mathrm{rank}(\mathsf{M}_\varepsilon) = q$ and

$$\mathrm{Tr}(\mathsf{M}_\varepsilon) = \sum_{i=1}^q \|f_{i,t}\|_{\mathcal{H}_\eta}^2 \le c_\eta 2^{2\beta}(1+(t/3)^{2\beta}e^{\frac{89}{\eta_0 t^2}})\sum_{i=1}^q \|f_i\|_{W_2^\beta(\mathbb{R}^d)}^2. \tag{D.21}$$

**Step 4. Bounding** $\|p - f(x\,;\,\mathsf{M}_\varepsilon,\phi_\eta)\|_{L^1(\mathbb{R}^d)}$**.**

Note that

$$f(x\,;\,\mathsf{M}_\varepsilon,\phi_\eta) = \phi_\eta(x)^\top \mathsf{M}_\varepsilon \phi_\eta(x) = \sum_{i=1}^q (f_{i,t}^\top \phi(x))^2 = \sum_{i=1}^q f_{i,t}(x)^2.$$

Then, since $a^2 - b^2 = (a-b)(a+b)$ for any $a,b \in \mathbb{R}$, by applying the Hölder inequality

$$\|p - f(x\,;\,\mathsf{M}_\varepsilon,\phi_\eta)\|_{L^1(\mathbb{R}^d)} = \|\sum_{i=1}^q f_i^2 - f_{i,t}^2\|_{L^1(\mathbb{R}^d)} = \|\sum_{i=1}^q (f_i - f_{i,t})(f_i + f_{i,t})\|_{L^1(\mathbb{R}^d)} \tag{D.22}$$

$$\le \sum_{i=1}^q \|f_i - f_{i,t}\|_{L^2(\mathbb{R}^d)}(\|f_i\|_{L^2(\mathbb{R}^d)} + \|f_{i,t}\|_{L^2(\mathbb{R}^d)}), \tag{D.23}$$

finally, by the Young convolution inequality,

$$\|f_{i,t}\|_{L^2(\mathbb{R}^d)} = \|f_i \star g_t\|_{L^2(\mathbb{R}^d)} \le \|f_i\|_{L^2(\mathbb{R}^d)}\|g_t\|_{L^1(\mathbb{R}^d)}.$$

By the change of variable $x = zt$, $dx = t^d dz$, we have

$$\|g_t\|_{L^1(\mathbb{R}^d)} = \int |g_t(tx)|dx = \int t^{-d}|g(x/t)|dx = \int |g(z)|dz = \|g\|_{L^1(\mathbb{R}^d)}. \tag{D.24}$$

then we obtain

$$\|p - f(x\,;\,\mathsf{M}_\varepsilon,\phi_\eta)\|_{L^1(\mathbb{R}^d)} \le (2t)^\beta (1+\|g\|_{L^1(\mathbb{R}^d)})\sum_{i=1}^q \|f_i\|_{W_2^\beta(\mathbb{R}^d)}\|f_i\|_{L^2(\mathbb{R}^d)}. \tag{D.25}$$

**Step 5. Bounding** $\|p - f(x\,;\,\mathsf{M}_\varepsilon,\phi_\eta)\|_{L^2(\mathbb{R}^d)}$**.**

With the same reasoning above, we have

$$\|p - f(x\,;\,\mathsf{M}_\varepsilon,\phi_\eta)\|_{L^2(\mathbb{R}^d)} = \|\sum_{i=1}^q f_i^2 - f_{i,t}^2\|_{L^2(\mathbb{R}^d)} = \|\sum_{i=1}^q (f_i - f_{i,t})(f_i + f_{i,t})\|_{L^2(\mathbb{R}^d)} \tag{D.26}$$

$$\le \sum_{i=1}^q \|f_i - f_{i,t}\|_{L^2(\mathbb{R}^d)}(\|f_i\|_{L^\infty(\mathbb{R}^d)} + \|f_{i,t}\|_{L^\infty(\mathbb{R}^d)}) \tag{D.27}$$

finally, by the Young convolution inequality,

$$\|f_{i,t}\|_{L^\infty(\mathbb{R}^d)} = \|f_i \star g_t\|_{L^\infty(\mathbb{R}^d)} \le \|f_i\|_{L^\infty(\mathbb{R}^d)}\|g_t\|_{L^1(\mathbb{R}^d)}.$$

Then,

$$\|p - f(x\,;\,\mathsf{M}_\varepsilon,\phi_\eta)\|_{L^2(\mathbb{R}^d)} \le (2t)^\beta(1+\|g\|_{L^1(\mathbb{R}^d)})\sum_{i=1}^q \|f_i\|_{W_2^\beta(\mathbb{R}^d)}\|f_i\|_{L^\infty(\mathbb{R}^d)} \tag{D.28}$$

**Step 6. Setting $t$ appropriately.**
Finally, noting that by construction $\|f\|_{L^2(\mathbb{R}^d)} \leq \|f\|_{W_2^\beta(\mathbb{R}^d)}$ and setting

$$t = \left(\frac{\varepsilon}{C_1}\right)^{\frac{1}{\beta}}, \quad C_1 = 2^\beta(1 + \|g\|_{L^1(\mathbb{R}^d)}) \sum_{i=1}^q \|f_i\|_{W_2^\beta(\mathbb{R}^d)} \max(\|f_i\|_{L^\infty(\mathbb{R}^d)}, \|f_i\|_{W_2^\beta(\mathbb{R}^d)}) \quad \text{(D.29)}$$

then, $\|p - f(x\,;\, \mathsf{M}_\varepsilon, \phi_\eta)\|_{L^j(\mathbb{R}^d)} \leq \varepsilon$, with $j = 1, 2$. By Littlewood's interpolation inequality, $\|\cdot\|_{L^r(\mathbb{R}^d)} \leq \|\cdot\|_{L^1(\mathbb{R}^d)}^{(2/r)-1} \|\cdot\|_{L^2(\mathbb{R}^d)}^{2-(2/r)}$ when $r \in [1, 2]$ (see, e.g, Thm. 8.5 pag 316 of [6]), we have

$$\|p - f(x\,;\, \mathsf{M}_\varepsilon, \phi_\eta)\|_{L^r(\mathbb{R}^d)} \leq \varepsilon, \quad \forall r \in [1, 2]. \quad \text{(D.30)}$$

By setting $C_2 = 2^{2\beta} \sum_{i=1}^q \|f_i\|_{W_2^\beta(\mathbb{R}^d)}^2$, we have

$$\text{Tr}(\mathsf{M}_\varepsilon) \leq c_\eta C_2(1 + e^{\frac{89}{\eta_0 t^2}}(t/3)^{2\beta}) \leq c_\eta C_2(1 + \tfrac{3^{-2\beta}}{C_1^2}\varepsilon^2 e^{\frac{89}{\eta_0}(\frac{C_1}{\varepsilon})^{2/\beta}}) \leq C|\eta|^{1/2}(1 + \varepsilon^2 e^{\frac{C'}{\eta_0 \varepsilon^{2/\beta}}}),$$

where $|\eta| = \det(\text{diag}(\eta))$ and $C' = 89 C_1^{2/\beta}$ and $C = \pi^{-d/2} C_2 \max(1, 3^{-2\beta}/C_1^2)$. $\qquad \square$

## D.3   Proof of Thm. 6

We can now prove Thm. 6. We will prove a more general result Thm. D.5, from which Thm. 6 follows when $R = 1$ and $\mathcal{X} = (-1, 1)^d$ applied to $\tilde{f}_1, \ldots, \tilde{f}_q \in W_2^\beta(\mathbb{R}^d) \cap L^\infty(\mathbb{R}^d)$ that are the extension to $\mathbb{R}^d$ of the functions $f_1, \ldots, f_q$ characterizing $p$ via Assumption 1. The details of the extension are in Cor. A.3

**Theorem D.5.** *Let $R > 0$ and let $\mathcal{X} \subseteq T = (R, R)^d$ be a non-empty open set with Lipschitz boundary. Let $f_1, \ldots, f_q \in W_2^\beta(\mathbb{R}^d) \cap L^\infty(\mathbb{R}^d)$ and let $p = \sum_{i=1}^q \tilde{f}_i^2$. Then, for any $\varepsilon \in (0, 1/e]$, there exists $m \in \mathbb{N}$, $\eta \in \mathbb{R}_{++}^d$, a base point matrix $\tilde{X} \in \mathbb{R}^{m \times d}$ and a matrix $A \in \mathbb{S}_+^m$ such that $\|f(\cdot\,;\, A, \tilde{X}, \eta) - p\|_{L^2(\mathcal{X})} \leq 2\varepsilon$, with*

$$m^{1/d} \leq C + C' \log \tfrac{1+R}{\varepsilon} + C'' R \varepsilon^{-\frac{1}{\beta}}(\log \tfrac{(1+R)}{\varepsilon})^{\frac{1}{2}} \quad \text{(D.31)}$$

*where $C, C', C''$ depend only on $\mathcal{X}, \beta, d, \|f_j\|_{W_2^\beta(\mathbb{R}^d)}, \|f_j\|_{L^\infty(\mathbb{R}^d)}$ for $j = 1, \ldots, q$. This implies that there exists a model of dimension $m$ such that $\|f(\cdot\,;\, A, \tilde{X}, \eta) - p\|_{L^2(\mathcal{X})} \leq \varepsilon$,*

$$m = O\left(R^d \varepsilon^{-d/\beta}(\log \tfrac{1+R}{\varepsilon})^{d/2}\right). \quad \text{(D.32)}$$

*Proof.* Let $\varepsilon \in (0, 1/e]$ and $\eta = \tau \mathbf{1}_d \in \mathbb{R}^d$ with $\tau > 0$ and $m \in \mathbb{N}$. Let $\mathsf{M}_\varepsilon \in \mathbb{S}_+(\mathcal{H}_\eta)$ be the operator constructed in Thm. D.4. We consider the compression of the model $p_\varepsilon = f(\cdot\,;\, \mathsf{M}_\varepsilon, \phi_\eta)$ as in Thm. C.4. In particular, let $\tilde{x}_1, \ldots, \tilde{x}_m$ be a covering of $T$ with $\ell_2$. We consider the following model $\tilde{p}_m = f(\cdot\,;\, A_m, \tilde{X}_m, \eta)$ where $\tilde{X}_m \in \mathbb{R}^{m \times d}$ is the base point matrix whose $j$-th row is the point $\tilde{x}_j$ for $j = 1, \ldots, m$, and where $A_m \in \mathbb{S}_+^m$ is defined as

$$A_m = K_{\tilde{X}, \tilde{X}, \eta}^{-1} \tilde{Z} \mathsf{M}_\varepsilon \tilde{Z}^* K_{\tilde{X}, \tilde{X}, \eta}^{-1}, \quad \text{(D.33)}$$

where $\tilde{Z} : \mathcal{H}_\eta \to \mathbb{R}^m$ is defined in (B.6) and its adjoint $\tilde{Z}^*$ has range equal to $\text{span}\{\phi_\eta(\tilde{x}_1), \ldots, \phi_\eta(\tilde{x}_1)\}$. Note that, by construction $\tilde{Z}\,\mathsf{M}_\varepsilon \tilde{Z}^* \in \mathbb{S}_+^m$ and so $A_m \in \mathbb{S}_+^m$.

**Step 1. Approximation error decomposition.** We will split the approximation error as follows,

$$\|\tilde{p}_m - p\|_{L^2(\mathcal{X})} \leq \|\tilde{p}_m - p_\varepsilon\|_{L^2(\mathcal{X})} + \|p_\varepsilon - p\|_{L^2(\mathcal{X})}. \quad \text{(D.34)}$$

Note that for the second term, by Thm. D.4, we have

$$\|p_\varepsilon - p\|_{L^r(\mathcal{X})} \leq \|p_\varepsilon - p\|_{L^r(\mathbb{R}^d)} \leq \varepsilon, \qquad \forall\, r \in [1, 2] \quad \text{(D.35)}$$

**Step 2. Error induced by a projection.** Since an $h$-covering of a set has fill distance $h$, by definition of fill distance (C.1), then we will choose the $m$ base points $\tilde{x}_1, \ldots, \tilde{x}_m$ to be an $h$-covering of the hypercube $T$. Since the $\ell_2$-ball of diameter 1 contains a cube of side $1/\sqrt{d}$, it is possible to cover a

cube of side $2R$ with $m \leq (1 + 2R\sqrt{d}/h)^d$ balls of diameter $2h$ (and so of radius $h$), see, e.g., Thm. 5.3, page 76 of [8]. Now, by Thm. C.4 applied to $\mathsf{M}_\varepsilon$, we have that when the fill distance $h$ (defined in (C.1)) satisfies $h \leq \sigma/C'$ with $\sigma = \min(R, 1/\sqrt{\tau})$, then

$$|p_\varepsilon(x) - \tilde{p}_m(x)| \leq 2C\sqrt{\|\mathsf{M}_\varepsilon\| p_\varepsilon(x)}\, e^{-\frac{c\sigma}{h}\log\frac{c\sigma}{h}} + C^2\|\mathsf{M}_\varepsilon\|e^{-\frac{2c\sigma}{h}\log\frac{c\sigma}{h}}, \quad \forall x \in \mathcal{X} \quad (\text{D.36})$$

with $c, C, C'$ depending only on $d$ and $c_\mathsf{M} =$. Now denoting by $\alpha = 2C\sqrt{\|\mathsf{M}_\varepsilon\|}e^{-\frac{c\sigma}{h}\log\frac{c\sigma}{h}}$ and $\beta = C^2\|\mathsf{M}_\varepsilon\|e^{-\frac{2c\sigma}{h}\log\frac{c\sigma}{h}}$, we have

$$\|p_\varepsilon - \tilde{p}_m\|_{L^2(\mathcal{X})} \leq \|\alpha\sqrt{p_\varepsilon} + \beta\|_{L^2(\mathcal{X})} \leq \alpha\|\sqrt{p_\varepsilon}\|_{L^2(\mathcal{X})} + \beta\|1\|_{L^2(\mathcal{X})} \quad (\text{D.37})$$

$$\leq (2R)^{d/2}\beta + \alpha\|p_\varepsilon^{1/2}\|_{L^2(\mathcal{X})}^2. \quad (\text{D.38})$$

where we used the fact that $\|1\|_{L^2(\mathcal{X})}^2 \leq \|1\|_{L^2(T)}^2 = \int_T dx = (2R)^d$.

**Step 3. Final bound.** First, note that by Thm. D.4

$$\text{Tr}(\mathsf{M}_\varepsilon) \leq C_1\tau^{d/2}(1 + \varepsilon^2\exp(\tfrac{C_2}{\tau}\varepsilon^{-\frac{2}{\beta}})) \quad (\text{D.39})$$

where $C_1, C_2$ are independent on $\varepsilon, \tau$ and depend only on $\beta, d, \|f_i\|_{W_2^\beta(\mathbb{R}^d)}, \|f_i\|_{L^\infty(\mathbb{R}^d)}$. By setting $\tau = \frac{C_2\varepsilon^{-2/\beta}}{2\log\frac{1+R}{\varepsilon}}$ we have

$$\|\mathsf{M}_\varepsilon\| \leq \text{Tr}(\mathsf{M}_\varepsilon) \leq C_1\tau^{d/2}(1 + \varepsilon^2\exp(\tfrac{C_2}{\tau}\varepsilon^{-\frac{2}{\beta}})) \leq (1+R)^2 C_3\varepsilon^{-d/\beta}. \quad (\text{D.40})$$

with $C_3 = 2^{-d/2}C_1C_2^{d/2}$. Then, note that $\|p_\varepsilon^{1/2}\|_{L^2(\mathcal{X})} = \|p_\varepsilon\|_{L^1(\mathcal{X})}^{1/2}$, so, using (D.35), we have

$$\|p_\varepsilon\|_{L^1(\mathcal{X})} \leq \|p_\varepsilon - p\|_{L^1(\mathcal{X})} + \|p\|_{L^1(\mathcal{X})} \leq 1 + \varepsilon \leq 2. \quad (\text{D.41})$$

By choosing $h = c\sigma/s$ with $s = \max(C', (1 + \frac{d}{2\beta})\log\frac{1}{\varepsilon} + (1 + \frac{d}{4})\log(1+R) + \log(C\sqrt{C_3}) + e)$, since $s \geq e$, then $\log s \geq 1$, so

$$Ce^{-\frac{c\sigma}{h}\log\frac{c\sigma}{h}} = Ce^{-s\log s} \leq Ce^{-s} \leq \tfrac{1}{15\sqrt{C_3}}(1+R)^{-d/4}\varepsilon^{1+\frac{d}{2\beta}}. \quad (\text{D.42})$$

Gathering the results from the previous steps, we have

$$\|\tilde{p}_m - p\|_{L^2(\mathcal{X})} \leq \varepsilon + 4\|\mathsf{M}_\varepsilon\|^{1/2}Ce^{-\frac{c\sigma}{h}\log\frac{c\sigma}{h}} + \|\mathsf{M}_\varepsilon\|R^{d/2}C^2e^{-\frac{c\sigma}{h}\log\frac{c\sigma}{h}} \quad (\text{D.43})$$

$$\leq \varepsilon + \tfrac{4}{15}(1+R)^{-(d/4)}\varepsilon + \tfrac{1}{225}\big(\tfrac{R}{R+1}\big)^{d/2}\varepsilon^2 \quad (\text{D.44})$$

$$\leq 2\varepsilon. \quad (\text{D.45})$$

To conclude we recall the fact that $\tilde{x}_1, \ldots, \tilde{x}_m$ is a $h$-covering of $T$, guarantees that the number of centers $m$ in the covering satisfies

$$m \leq (1 + \tfrac{2R\sqrt{d}}{h})^d. \quad (\text{D.46})$$

Then, since $h \geq c\sigma/(C_4\log\frac{C_5\log(1+R)}{\varepsilon})$ with $C_4 = 1 + d/\min(2\beta, 4)$ and $C_5 = (C\sqrt{C_3}e)^{1/C_4}$, and since $\sigma = \min(R, 1/\sqrt{\tau})$, then $R/\sigma = \max(1, R\sqrt{\tau}) \leq 1 + \sqrt{C_2/2}\varepsilon^{-1/\beta}(\log\frac{1+R}{\varepsilon})^{-1/2}$, so we have

$$m^{\frac{1}{d}} \leq 1 + 2R\sqrt{d}/h \leq 1 + C_4\big(1 + R\sqrt{d}(\tfrac{C_2}{2})^{1/2}\varepsilon^{-\frac{1}{\beta}}(\log\tfrac{1+R}{\varepsilon})^{-\frac{1}{2}}\big)\log\tfrac{C_5(1+R)}{\varepsilon} \quad (\text{D.47})$$

$$= (1 + C_4\log C_5) + C_4\log\tfrac{1+R}{\varepsilon} + RC_4\sqrt{d}(\tfrac{C_2}{2})^{\frac{1}{2}}\varepsilon^{-\frac{1}{\beta}}(\log\tfrac{1+R}{\varepsilon})^{-\frac{1}{2}}\log\tfrac{C_5(1+R)}{\varepsilon} \quad (\text{D.48})$$

$$\leq C_6 + C_4\log\tfrac{1+R}{\varepsilon} + RC_7\varepsilon^{-\frac{1}{\beta}}(\log\tfrac{(1+R)}{\varepsilon})^{\frac{1}{2}} \quad (\text{D.49})$$

with $C_6 = 1 + C_4\log C_5$, $C_7 = C_4\sqrt{d}(\tfrac{C_2}{2})^{1/2}\log(eC_5)$, since $\log(eC_5) \geq 1$, then

$$\log\tfrac{C_5(1+R)}{\varepsilon} = \log eC_5 + \log\tfrac{(1+R)}{e\varepsilon} \leq (\log eC_5)(1 + \tfrac{\log\frac{(1+R)}{e\varepsilon}}{\log eC_5}) \quad (\text{D.50})$$

$$\leq (\log eC_5)(1 + \log\tfrac{(1+R)}{e\varepsilon}) \leq (\log eC_5)(1 + \log\tfrac{(1+R)}{\varepsilon}). \quad (\text{D.51})$$

The constants $C, C', C''$ in the statement of the theorem correspond to $C = C_6, C' = C_4, C'' = C_7$. $\qquad\square$

# E Learning a PSD model from examples

In this section we provide a proof for Thm. 7, which characterizes the learning capabilities of PSD models. We first provide intermediate results that will be useful for the proof.

Let $\mathcal{X}$ be a compact space and let $p : \mathcal{X} \to \mathbb{R}$ be a probability density which we assume to belong to $p \in L^2(\mathcal{X})$. Let $x_1, \ldots, x_n$ sampled i.i.d. according to $p$. We will study an estimator for $p$ in terms of the squared L2 norm $\| \cdot \|_{L^2(\mathcal{X})}$. Let $\eta = \eta_0 1_d$ with $\eta_0 > 0$ and $\tilde{X} \in \mathbb{R}^{m \times d}$ the base point matrix whose rows are some points $\tilde{x}_1, \ldots, \tilde{x}_m$. We will consider the following estimator $\hat{p}$ for $p$

$$\hat{p}(x) = f(x\,;\,\hat{A}, \tilde{X}, \eta), \qquad \hat{A} = \min_{A \in \mathbb{S}_+^m} \hat{L}_\lambda(A), \tag{E.1}$$

and, denoting by $\tilde{R}$ the Cholesky decomposition of $K_{\tilde{X}, \tilde{X}, \eta}$, i.e. the upper triangular matrix such that $K_{\tilde{X}, \tilde{X}, \eta} = \tilde{R}^\top \tilde{R}$, we define

$$\hat{L}_\lambda(A) = \int_{\mathcal{X}} f(x\,;\,A, \tilde{X}, \eta)^2 dx - \frac{2}{n} \sum_{i=1}^n \mathbf{1}_{\mathcal{X}}(x_i) f(x_i\,;\,A, \tilde{X}, \eta) + \lambda \|\tilde{R} A \tilde{R}^\top\|_F^2. \tag{E.2}$$

Denote by $L_\lambda(A)$ the following functional

$$L_\lambda(A) = \int_{\mathcal{X}} f(x\,;\,A, \tilde{X}, \eta)^2 dx - \int_{\mathcal{X}} f(x\,;\,A, \tilde{X}, \eta) p(x) dx + \lambda \|\tilde{R} A \tilde{R}^\top\|_F^2. \tag{E.3}$$

and by $\bar{A}_{\eta, \lambda} \in \mathbb{S}_+^m$ the matrix $\bar{A}_{\eta, \lambda} = \min_{A \in \mathbb{S}_+} L_\lambda(A)$. We have the following result

## E.1 Operatorial characterization of $\hat{L}_\lambda, L_\lambda$

We can now rewrite the loss functions as follows

**Lemma E.1** (Characterization of $\hat{L}_\lambda, L_\lambda$ in terms of $\hat{v}, v$). *For any $\lambda \geq 0$ the following holds*

$$\hat{L}_\lambda(A) = \|S\mathrm{vec}(\tilde{Z}^* A \tilde{Z})\|_{L^2(\mathcal{X})}^2 + \lambda \|\mathrm{vec}(\tilde{Z}^* A \tilde{Z})\|_{\mathcal{G}_\eta}^2 - 2 \left\langle \hat{v}, \mathrm{vec}(\tilde{Z}^* A \tilde{Z}) \right\rangle_{\mathcal{G}_\eta} \tag{E.4}$$

$$L_\lambda(A) = \|S\mathrm{vec}(\tilde{Z}^* A \tilde{Z})\|_{L^2(\mathcal{X})}^2 + \lambda \|\mathrm{vec}(\tilde{Z}^* A \tilde{Z})\|_{\mathcal{G}_\eta}^2 - 2 \left\langle v, \mathrm{vec}(\tilde{Z}^* A \tilde{Z}) \right\rangle_{\mathcal{G}_\eta}. \tag{E.5}$$

*Proof.* With the notation Appendix A and by using the operators defined in Appendix B for any $\mathsf{M} \in \mathbb{S}_+(\mathcal{H}_\eta)$ we have

$$f(x\,;\,\mathsf{M}, \phi_\eta) = \langle \psi_\gamma(x), \mathrm{vec}(\mathsf{M}) \rangle_{\mathcal{G}_\eta}, \quad \forall x \in \mathbb{R}^d \tag{E.6}$$

and in particular for any $A \in \mathbb{S}_+^m$, we have

$$f(x\,;\,A, \tilde{X}, \eta) = f(x\,;\,\tilde{Z}^* A \tilde{Z}, \phi_\eta) = \left\langle \psi_\gamma(x), \mathrm{vec}(\tilde{Z}^* A \tilde{Z}) \right\rangle_{\mathcal{G}_\eta}, \quad \forall x \in \mathbb{R}^d \tag{E.7}$$

Now note that, by ciclicity of the trace, for any matrix $A, B \in \mathbb{R}^{m \times m}$ we have

$$\|B^{1/2} A B^{1/2}\|_F^2 = \mathrm{Tr}(B^{1/2} A B^{1/2} B^{1/2} A B^{1/2}) = \mathrm{Tr}(ABAB). \tag{E.8}$$

This implies that $\|K_{\tilde{X}, \tilde{X}, \eta}^{1/2} A K_{\tilde{X}, \tilde{X}, \eta}^{1/2}\|_F^2 = \mathrm{Tr}(A K_{\tilde{X}, \tilde{X}, \eta} A K_{\tilde{X}, \tilde{X}, \eta})$. Moreover, by ciclicity of the trace, definition of Frobenious norm and since $\tilde{Z} \tilde{Z}^* = K_{\tilde{X}, \tilde{X}, \eta}$ we have

$$\mathrm{Tr}(A K_{\tilde{X}, \tilde{X}, \eta} A K_{\tilde{X}, \tilde{X}, \eta}) = \mathrm{Tr}(A \tilde{Z} \tilde{Z}^* A \tilde{Z} \tilde{Z}^*) = \mathrm{Tr}(\tilde{Z}^* A \tilde{Z} \tilde{Z}^* A \tilde{Z}) \tag{E.9}$$

$$= \left\langle \mathrm{vec}(\tilde{Z}^* A \tilde{Z}), \mathrm{vec}(\tilde{Z}^* A \tilde{Z}) \right\rangle_{\mathcal{G}_\eta} = \|\mathrm{vec}(\tilde{Z}^* A \tilde{Z})\|_{\mathcal{G}_\eta}^2. \tag{E.10}$$

By linearity of the integral and the inner product and since $\phi_\eta$ and so $\psi_\eta$ are uniformly bounded,

$$\int_{\mathcal{X}} f(x\,;\,\mathsf{M}, \phi_\eta)^2 dx = \int_{\mathcal{X}} \left\langle \mathrm{vec}(\mathsf{M}), (\psi_\eta(x) \psi_\eta(x)^\top) \mathrm{vec}(\mathsf{M}) \right\rangle_{\mathcal{G}_\eta} dx \tag{E.11}$$

$$= \left\langle \mathrm{vec}(\mathsf{M}), \left( \int_{\mathcal{X}} \psi_\eta(x) \psi_\eta(x)^\top dx \right) \mathrm{vec}(\mathsf{M}) \right\rangle_{\mathcal{G}_\eta} \tag{E.12}$$

$$= \langle \mathrm{vec}(\mathsf{M}), Q \mathrm{vec}(\mathsf{M}) \rangle_{\mathcal{G}_\eta} = \langle \mathrm{vec}(\mathsf{M}), S^* S \mathrm{vec}(\mathsf{M}) \rangle_{\mathcal{G}_\eta} \tag{E.13}$$

$$= \langle S \mathrm{vec}(\mathsf{M}), S \mathrm{vec}(\mathsf{M}) \rangle_{\mathcal{G}_\eta} = \|S \mathrm{vec}(\mathsf{M})\|_{L^2(\mathcal{X})}^2 \tag{E.14}$$

Then, we have

$$\hat{L}_\lambda(A) = \|S\mathrm{vec}(\tilde{Z}^* A\tilde{Z})\|^2_{L^2(\mathcal{X})} - 2\left\langle \hat{v}, \mathrm{vec}(\tilde{Z}^* A\tilde{Z})\right\rangle_{\mathcal{G}_\eta} + \lambda\|\mathrm{vec}(\tilde{Z}^* A\tilde{Z})\|^2_{\mathcal{G}_\eta}. \qquad (E.15)$$

The identical reasoning holds for $L_\lambda(A)$, with respect to $v$. $\qquad\square$

**Theorem E.2** (Error decomposition). *Let $\hat{A}$ be a minimizer of $\hat{L}_\lambda$ over a set $S \subseteq \mathbb{R}^{m\times m}$ (non-necessarily convex). Denote by $\mu(A)$ the vector $\mu(A) = \mathrm{vec}(\tilde{Z}^* A\tilde{Z}) \in \mathcal{G}_\eta$ for any $A \in S$. Then for any $A \in S$ the following holds*

$$\left(\|S\mu(\hat{A}) - p\|^2_{L^2(\mathcal{X})} + \lambda\|\mu(\hat{A})\|^2_{\mathcal{G}_\eta}\right)^{1/2} \leq \left(\|S\mu(A) - p\|^2_{L^2(\mathcal{X})} + \lambda\|\mu(A)\|^2_{\mathcal{G}_\eta}\right)^{1/2}$$
$$+ 3\sqrt{2}\|(Q + \lambda I)^{-1/2}(\hat{v} - v)\|_{\mathcal{G}_\eta}. \qquad (E.16)$$

*Proof.* We start noting that since $\hat{A}$ is the minimizer over $S$ of $\hat{L}_\lambda$, then $\hat{L}_\lambda(\hat{A}) \leq \hat{L}_\lambda(A)$ for any $A \in S$. In particular, since $\bar{A} \in S$ this means that $\hat{L}_\lambda(\hat{A}) \leq \hat{L}_\lambda(\bar{A})$, then $\hat{L}_\lambda(\hat{A}) - \hat{L}_\lambda(\bar{A}) \leq 0$, this implies that

$$L_\lambda(\hat{A}) - L_\lambda(A) = L_\lambda(\hat{A}) - \hat{L}_\lambda(\hat{A}) + \hat{L}_\lambda(\hat{A}) - \hat{L}_\lambda(A) + \hat{L}_\lambda(A) - L_\lambda(A) \qquad (E.17)$$

$$\leq L_\lambda(\hat{A}) - \hat{L}_\lambda(\hat{A}) + \hat{L}_\lambda(A) - L_\lambda(A). \qquad (E.18)$$

Denote $\hat{\mu} = \mathrm{vec}(\tilde{Z}^* \hat{A}\tilde{Z})$ and $\bar{\mu} = \mathrm{vec}(\tilde{Z}^* \bar{A}\tilde{Z})$. Now note that, by the characterization of $L_\lambda, \hat{L}_\lambda$ in [Lemma E.1](#), we have

$$L_\lambda(\hat{A}) - \hat{L}_\lambda(\hat{A}) + \hat{L}_\lambda(A) - L_\lambda(A) = 2\left\langle \hat{v} - v, \hat{\mu} - \bar{\mu}\right\rangle_{\mathcal{G}_\eta}. \qquad (E.19)$$

**Step 1. Decomposing the error.**
Note that, since $v = S^* p$, then $\langle v, w\rangle_{\mathcal{G}_\eta} = \langle S^* p, w\rangle_{\mathcal{G}_\eta} = \langle p, Sw\rangle_{L^2(\mathcal{X})}$ for any $w \in \mathcal{G}_\eta$. Then, for any $A \in \mathbb{S}^m_+$, denoting by $\mu = \mathrm{vec}(\tilde{Z}^* A\tilde{Z})$, and substituting $\langle v, \mu\rangle_{\mathcal{G}_\eta}$ with $\langle p, S\mu\rangle_{L^2(\mathcal{X})}$ in the definition of $L_\lambda(A)$, we have

$$L_\lambda(A) + \|p\|^2 = \|S\mu\|^2_{L^2(\mathcal{X})} - 2\langle p, S\mu\rangle_{L^2(\mathcal{X})} + \|p\|^2_{L^2(\mathcal{X})} + \lambda\|\mu\|^2_{\mathcal{G}_\eta} \qquad (E.20)$$

$$= \|S\mu - p\|^2_{L^2(\mathcal{X})} + \lambda\|\mu\|^2_{\mathcal{G}_\eta}. \qquad (E.21)$$

From [(E.17)](#) and [(E.19)](#) and the equation above, we obtain

$$\|S\hat{\mu} - p\|^2_{L^2(\mathcal{X})} + \lambda\|\hat{\mu}\|^2_{\mathcal{G}_\eta} = L_\lambda(\hat{A}) + \|p\|^2_{L^2(\mathcal{X})}$$
$$\leq L_\lambda(\bar{A}) + \|p\|^2_{L^2(\mathcal{X})} + 2\left\langle \hat{v} - v, \hat{\mu} - \bar{\mu}\right\rangle_{\mathcal{G}_\eta} \qquad (E.22)$$
$$= \|S\bar{\mu} - p\|^2_{L^2(\mathcal{X})} + \lambda\|\bar{\mu}\|^2_{\mathcal{G}_\eta} + 2\left\langle \hat{v} - v, \hat{\mu} - \bar{\mu}\right\rangle_{\mathcal{G}_\eta}$$

Now, note that

$$\|S\hat{\mu} - p\|^2_{L^2(\mathcal{X})} = \|S(\hat{\mu} - \bar{\mu}) + (S\bar{\mu} - p)\|^2_{L^2(\mathcal{X})} \qquad (E.23)$$

$$= \|S(\hat{\mu} - \bar{\mu})\|^2_{L^2(\mathcal{X})} + 2\left\langle S(\hat{\mu} - \bar{\mu}), (S\bar{\mu} - p)\right\rangle_{L^2(\mathcal{X})} + \|S\bar{\mu} - p\|^2_{L^2(\mathcal{X})} \qquad (E.24)$$

$$\|\hat{\mu}\|^2_{\mathcal{G}_\eta} = \|(\hat{\mu} - \bar{\mu}) + \bar{\mu}\|^2_{\mathcal{G}_\eta} = \|\hat{\mu} - \bar{\mu}\|^2_{\mathcal{G}_\eta} + 2\left\langle \bar{\mu}, \hat{\mu} - \bar{\mu}\right\rangle_{\mathcal{G}_\eta} + \|\bar{\mu}\|^2_{\mathcal{G}_\eta}. \qquad (E.25)$$

Expanding $\|S\hat{\mu} - p\|^2_{L^2(\mathcal{X})}$ and $\|\hat{\mu}\|^2_{\mathcal{G}_\eta}$ in [(E.22)](#) and reorganizing the terms, we obtain

$$\|S(\hat{\mu} - \bar{\mu})\|^2_{L^2(\mathcal{X})} + \lambda\|\hat{\mu} - \bar{\mu}\|^2_{\mathcal{G}_\eta} \leq -2\left\langle S(\hat{\mu} - \bar{\mu}), (S\bar{\mu} - p)\right\rangle_{L^2(\mathcal{X})}$$
$$- 2\lambda\left\langle \bar{\mu}, \hat{\mu} - \bar{\mu}\right\rangle_{\mathcal{G}_\eta} \qquad (E.26)$$
$$+ 2\left\langle \hat{v} - v, \hat{\mu} - \bar{\mu}\right\rangle_{\mathcal{G}_\eta}.$$

**Step 2. Bounding the three terms of the decomposition.**
The proof is concluded by bounding the three terms of the right hand side of the equation above, indeed

$$-2\left\langle S(\hat{\mu} - \bar{\mu}), (S\bar{\mu} - p)\right\rangle_{L^2(\mathcal{X})} \leq 2\|S(\hat{\mu} - \bar{\mu})\|_{L^2(\mathcal{X})}\|S\bar{\mu} - p\|_{L^2(\mathcal{X})} \qquad (E.27)$$

$$-2\lambda\left\langle \bar{\mu}, \hat{\mu} - \bar{\mu}\right\rangle_{\mathcal{G}_\eta} \leq 2\lambda\|\bar{\mu}\|_{\mathcal{G}_\eta}\|\hat{\mu} - \bar{\mu}\|_{\mathcal{G}_\eta}. \qquad (E.28)$$

For the third term we multiply and divide by $Q + \lambda$ (it is invertible since $Q \in \mathbb{S}_+(\mathcal{G}_\eta)$ and $\lambda > 0$), so

$$2 \langle \hat{v} - v, \hat{\mu} - \bar{\mu} \rangle_{\mathcal{G}_\eta} = 2 \left\langle (Q + \lambda)^{-1/2}(\hat{v} - v), (Q + \lambda)^{1/2}(\hat{\mu} - \bar{\mu}) \right\rangle_{\mathcal{G}_\eta} \tag{E.29}$$
$$\leq 2\|(Q + \lambda)^{-1/2}(\hat{v} - v)\|_{\mathcal{G}_\eta}\|(Q + \lambda)^{1/2}(\hat{\mu} - \bar{\mu})\|_{\mathcal{G}_\eta}.$$

Note that for any $w \in \mathcal{G}_\eta$, since $Q$ is characterized as $Q = S^*S$, we have

$$\|(Q + \lambda)^{1/2}w\|_{\mathcal{G}_\eta}^2 = \langle w, (Q + \lambda)w \rangle_{\mathcal{G}_\eta} = \langle w, Qw \rangle_{\mathcal{G}_\eta} + \lambda \langle w, w \rangle_{\mathcal{G}_\eta} \tag{E.30}$$
$$= \langle Sw, Sw \rangle_{\mathcal{G}_\eta} + \lambda \langle w, w \rangle_{\mathcal{G}_\eta} = \|Sw\|_{\mathcal{G}_\eta}^2 + \lambda\|w\|_{\mathcal{G}_\eta}^2.$$

By applying the equation above to $w = \hat{\mu} - \bar{\mu}$, since $\sqrt{a^2 + b^2} \leq a + b, \forall a, b \geq 0$, we have

$$\|(Q + \lambda)^{1/2}(\hat{\mu} - \bar{\mu})\|_{\mathcal{G}_\eta} \leq \|S(\hat{\mu} - \bar{\mu})\|_{L^2(\mathcal{X})} + \sqrt{\lambda}\|(\hat{\mu} - \bar{\mu})\|_{\mathcal{G}_\eta}. \tag{E.31}$$

Combining (E.26) with the bounds in (E.27) and (E.28) for the first two terms of its right hand side and with the bounds in (E.29) and (E.31) for the third term, we have

$$\|S(\hat{\mu} - m)\|_{L^2(\mathcal{X})}^2 + \lambda\|\hat{\mu} - \bar{\mu}\|_{\mathcal{G}_\eta}^2 \leq 2\alpha\|S(\hat{\mu} - m)\|_{L^2(\mathcal{X})} + 2\beta\sqrt{\lambda}\|\hat{\mu} - \bar{\mu}\|_{\mathcal{G}_\eta}, \tag{E.32}$$

with $\alpha = \|S\bar{\mu} - p\|_{L^2(\mathcal{X})} + \|(Q + \lambda)^{-1/2}(\hat{v} - v)\|_{\mathcal{G}_\eta}, \beta = \sqrt{\lambda}\|\bar{\mu}\|_{\mathcal{G}_\eta} + \|(Q + \lambda)^{-1/2}(\hat{v} - v)\|_{\mathcal{G}_\eta}.$

**Step 3. Solving the inequality associated to the bound of the three terms.**
Now denoting by $x = \|S(\hat{\mu} - \bar{\mu})\|_{L^2(\mathcal{X})}$ and $y = \sqrt{\lambda}\|\hat{\mu} - \bar{\mu}\|_{\mathcal{G}_\eta}$, the inequality above becomes

$$x^2 + y^2 \leq 2\alpha x + 2\beta y. \tag{E.33}$$

By completing the squares it is equivalent to $(x - \alpha)^2 + (y - \beta)^2 \leq \alpha^2 + \beta^2$, from which we derive that $(x - \alpha)^2 \leq (x - \alpha)^2 + (y - \beta)^2 \leq \alpha^2 + \beta^2$. This implies $x \leq \alpha + \sqrt{\alpha^2 + \beta^2} \leq 2\alpha + \beta$. With the same reasoning we derive $y \leq \alpha + 2\beta$, that corresponds to

$$\|S(\hat{\mu} - \bar{\mu})\|_{L^2(\mathcal{X})} \leq 2\|S\bar{\mu} - p\|_{L^2(\mathcal{X})} + 3\|(Q + \lambda)^{-1/2}(\hat{v} - v)\|_{\mathcal{G}_\eta} + \sqrt{\lambda}\|\bar{\mu}\|_{\mathcal{G}_\eta} \tag{E.34}$$

$$\|\hat{\mu} - \bar{\mu}\|_{\mathcal{G}_\eta} \leq \tfrac{1}{\sqrt{\lambda}}\|S\bar{\mu} - p\|_{L^2(\mathcal{X})} + \tfrac{3}{\sqrt{\lambda}}\|(Q + \lambda)^{-1/2}(\hat{v} - v)\|_{\mathcal{G}_\eta} + 2\|\bar{\mu}\|_{\mathcal{G}_\eta} \tag{E.35}$$

**Step 4. The final bound.**
The final result is obtained by bounding the term $\langle \hat{v} - v, \hat{\mu} - \bar{\mu} \rangle_{\mathcal{G}_\eta}$ in (E.22). In particular, we will bound it by using (E.29), and by bounding the resulting term $\|(Q + \lambda)^{1/2}(\hat{\mu} - \bar{\mu})\|_{\mathcal{G}_\eta}$ with (E.31) and the resulting terms $\|S(\hat{\mu} - \bar{\mu})\|_{L^2(\mathcal{X})}, \|\hat{\mu} - \bar{\mu}\|_{\mathcal{G}_\eta}$ via (E.34). This leads to

$$\|S\hat{\mu} - p\|_{L^2(\mathcal{X})}^2 + \lambda\|\hat{\mu}\|_{\mathcal{G}_\eta}^2 \leq \|S\bar{\mu} - p\|_{L^2(\mathcal{X})}^2 + \lambda\|\bar{\mu}\|_{\mathcal{G}_\eta}^2 \tag{E.36}$$
$$+ 6\tau(\|S\bar{\mu} - p\|_{L^2(\mathcal{X})} + \sqrt{\lambda}\|\bar{\mu}\|_{\mathcal{G}_\eta} + 2\tau).$$

with $\tau = \|(Q + \lambda)^{-1/2}(\hat{v} - v)\|_{\mathcal{G}_\eta}$. We can optimize the writing of the theorem by noting that since $2ab \leq a^2 + b^2$ for any $a, b \in \mathbb{R}$, we have $a + b \leq \sqrt{2(a^2 + b^2)}$ and so $\|S\bar{\mu} - p\|_{L^2(\mathcal{X})} + \sqrt{\lambda}\|\bar{\mu}\|_{\mathcal{G}_\eta} \leq \sqrt{2(\|S\bar{\mu} - p\|_{L^2(\mathcal{X})}^2 + \lambda\|\bar{\mu}\|_{\mathcal{G}_\eta}^2)}$ and the bound in (E.36) becomes

$$\|S\hat{\mu} - p\|_{L^2(\mathcal{X})}^2 + \lambda\|\hat{\mu}\|_{\mathcal{G}_\eta}^2 \leq z^2 + 6\tau(\sqrt{2}z + 2\tau) \leq (z + 3\sqrt{2}\tau)^2 \tag{E.37}$$

where $z^2 = \|S\bar{\mu} - p\|_{L^2(\mathcal{X})}^2 + \lambda\|\bar{\mu}\|_{\mathcal{G}_\eta}^2$. The final result is obtained by noting that, since $\bar{A}$ is the minimizer of $L_\lambda$, then for any $A \in \mathbb{S}_+^m$ the following holds

$$z^2 = \|p\|^2 + L_\lambda(\bar{A}) = \|p\|^2 + \min_{A' \in \mathbb{S}_+^m} L_\lambda(A') \tag{E.38}$$

$$\leq \|p\|^2 + L_\lambda(A) = \|S\mu(A) - p\|_{L^2(\mathcal{X})}^2 + \lambda\|\mu(A)\|_{\mathcal{G}_\eta}^2. \tag{E.39}$$

$\square$

**Lemma E.3.** *Let $\mathcal{X} \subseteq \mathbb{R}^d$ and let $\eta \in \mathbb{R}_{++}^d$. Let $\tilde{x}_1, \ldots, \tilde{x}_m$ be a set of points in $\mathbb{R}^d$ and let $\tilde{X} \in \mathbb{R}^{m \times d}$ be the associated base point matrix, i.e., the $j$-th row of $\tilde{X}$ corresponds to $\tilde{x}_j$. With the notation and the definitions of [Appendix B](#) denote by $\mu(A) = \text{vec}(\tilde{Z}^* A \tilde{Z})$. Then, for any $A \in \mathbb{S}_+^m$ and any $p \in L^2(\mathcal{X})$*

$$\|S\mu(A) - p\|_{L^2(\mathcal{X})}^2 = \|f(\cdot\,; A, \tilde{X}, \eta) - p\|_{L^2(\mathcal{X})}^2, \tag{E.40}$$

*and moreover*

$$\|\mu(A)\|_{\mathcal{G}_\eta}^2 = \|\tilde{Z}^* A \tilde{Z}\|_F^2. \tag{E.41}$$

*Proof.* Denote by $\mu(A) = \text{vec}(\tilde{Z}^* A \tilde{Z})$. We recall that the operator used in the rest of the section are defined in [Appendix B](#). First we recall from [(E.20)](#) that, by definition of $S : \mathcal{G}_\eta \to L^2(\mathcal{X})$, we have $S\mu(A) = \langle \psi_\eta(\cdot), \mu(A) \rangle_{\mathcal{G}_\eta} \in L^2(\mathcal{X})$. In particular, by definition of vec, for any $x \in \mathcal{X}$

$$(S\mu(A))(x) = \langle \mu(A), \psi_\eta(x) \rangle_{\mathcal{G}_\eta} = \left\langle \text{vec}(\tilde{Z}^* A \tilde{Z}), \phi_\eta(x) \otimes \phi_\eta(x) \right\rangle_{\mathcal{G}_\eta} = \phi_\eta(x)^\top \tilde{Z}^* A \tilde{Z} \phi_\eta(x). \tag{E.42}$$

Now, since $\tilde{Z}^* A \tilde{Z} = \sum_{i,j=1}^m A_{i,j} \phi_\eta(x_i) \phi_\eta(x_j)^\top$, and, by the representer property of the kernel $k_\eta$ we have $k_\eta(x, x') = \phi_\eta(x)^\top \phi_\eta(x')$ (see [(A.2)](#) and [Example 1](#)), then

$$\phi_\eta(x)^\top \tilde{Z}^* A \tilde{Z} \phi_\eta(x) = \sum_{i,j=1}^m A_{i,j}(\phi_\eta(x)^\top \phi_\eta(x_i))(\phi_\eta(x_j)^\top \phi_\eta(x)) \tag{E.43}$$

$$= \sum_{i,j=1}^m A_{i,j} k_\eta(x, x_i) k_\eta(x, x_j) = f(x\,; A, \tilde{X}, \eta). \tag{E.44}$$

Then $S\mu(A) = f(\cdot\,; A, \tilde{X}, \eta) \in L^2(\mathcal{X})$. So

$$\|S\mu(A) - p\|_{L^2(\mathcal{X})}^2 = \|f(\cdot\,; A, \tilde{X}, \eta) - p\|_{L^2(\mathcal{X})}^2. \tag{E.45}$$

To conclude note that, by the properties of vec (see [Appendix B](#)), we have

$$\|\mu(A)\|_{\mathcal{G}_\eta}^2 = \left\langle \text{vec}(\tilde{Z}^* A \tilde{Z}), \text{vec}(\tilde{Z}^* A \tilde{Z}) \right\rangle_{\mathcal{G}_\eta} = \|\tilde{Z}^* A \tilde{Z}\|_F^2 \tag{E.46}$$

$\square$

**Lemma E.4.** *Let $s \in \mathbb{N}$ and $\delta \in (0, 1]$, $\lambda > 0$ let $\mathcal{X} \subset \mathbb{R}^d$ be an open set with Lipschitz boundary and let $p \in L^1(\mathcal{X}) \cap L^\infty(\mathcal{X})$. Then the following holds with probability $1 - \delta$*

$$\|(Q + \lambda I)^{-1/2}(\hat{v} - v)\|_{\mathcal{G}_\eta} \leq \frac{C\tau^{d/4} \log \frac{2}{\delta}}{n(\lambda \tau^d)^{d/4s}} + \sqrt{\frac{2\text{Tr}(Q_\lambda^{-1} Q) \log \frac{2}{\delta}}{n}}, \tag{E.47}$$

*where $C$ is a constant that depends only on $\mathcal{X}, s, d$.*

*Proof.* We are going to use here a Bernstein inequality for random vectors in separable Hilbert spaces (see, e.g., Thm 3.3.4 of [41]). Define the random variable $\zeta = Q_\lambda^{-1/2} \psi_\eta(x) \mathbf{1}_\mathcal{X}(x)$ with $x$ distributed according to $\rho$ and $Q_\lambda = Q + \lambda I$. To apply such inequality, we need to control the second moment and the norm of $\zeta$. First note that $\zeta$

**Step 1. Bounding the variance of $\zeta$.** We have

$$\mathbb{E}\|\zeta\|^2 = \text{Tr}\left(\int \zeta \zeta^\top p(x)dx\right) = \text{Tr}\left(\int_\mathcal{X} Q_\lambda^{-1/2}(\psi_\eta(x)\psi_\eta(x)^\top)Q_\lambda^{-1/2} p(x)dx\right) \tag{E.48}$$

$$\leq \|p\|_{L^\infty(\mathcal{X})} \text{Tr}\left(Q_\lambda^{-1/2}\left(\int_\mathcal{X} \psi_\eta(x)\psi_\eta(x)^\top dx\right) Q_\lambda^{-1/2}\right) \tag{E.49}$$

$$= \|p\|_{L^\infty(\mathcal{X})} \text{Tr}(Q_\lambda^{-1/2} Q Q_\lambda^{-1/2}) = \text{Tr}(Q Q_\lambda^{-1}). \tag{E.50}$$

**Bounding the norm of** $\zeta$**.** Applying Lemma C.2 the the operator $Q_\lambda^{-1/2}$ we have that

$$\text{ess-sup}\|\zeta\|_{\mathcal{G}_\eta} \leq \sup_{x \in \mathcal{X}} \|Q_\lambda^{-1/2}\phi_\eta(x)\| \leq \sup_{\|f\|_{\mathcal{G}_\eta} \leq 1} \|Q_\lambda^{-1/2}f\|_{L^\infty(\mathcal{X})}. \tag{E.51}$$

Now, according to the definitions in Appendix B, note that the reproducing kernel Hilbert space $\mathcal{G}_\eta$ is associated to the kernel $h(x, x') = \psi_\eta(x)^\top \psi_\eta(x)^\top$, that corresponds to

$$h(x, x') = \psi_\eta(x)^\top \psi_\eta(x)^\top = \langle \phi_\eta(x) \otimes \phi_\eta(x), \phi_\eta(x') \otimes \phi_\eta(x') \rangle_{\mathcal{H}_\eta \otimes \mathcal{H}_\eta} \tag{E.52}$$

$$= (\phi_\eta(x)^\top \phi_\eta(x'))^2 = k_\eta(x, x')^2 = k_{2\eta}(x, x'). \tag{E.53}$$

Then $\mathcal{G}_\eta$ is still a RKHS associated to a Gaussian kernel, in particular $k_{2\eta}$, so $\mathcal{G}_\eta \subset W_2^s(\mathbb{R}^d)$ for any $s \geq 0$. In particular, by (C.9) and the fact that $\mathcal{X} \subset \mathbb{R}^d$, we have

$$\|f\|_{W_2^s(\mathcal{X})} \leq \|g\|_{W_2^s(\mathbb{R}^d)} \leq C\|f\|_{\mathcal{G}_\eta}\tau^{(s-d)/2}, \quad \forall f \in \mathcal{G}_\eta, \tag{E.54}$$

where $C$ depends only on $s, d$. Now by the interpolation inequality for Sobolev spaces (see, e.g. Thm 5.9, page 139 of [1]), we have that for any $g \in W_2^s(\mathcal{X})$ the following holds

$$\|g\|_{L^\infty(\mathcal{X})} \leq C'\|g\|_{W_2^s(\mathcal{X})}^{d/(2s)}\|g\|_{L^2(\mathcal{X})}^{1-d/(2s)}. \tag{E.55}$$

Applying the inequality above to the function $g = Q_\lambda^{-1/2}f$, with $f \in \mathcal{G}_\eta$ and $\|f\|_{\mathcal{G}_\eta} \leq 1$, we have

$$\|Q_\lambda^{-1/2}f\|_{L^\infty(\mathcal{X})} \leq C'\|Q_\lambda^{-1/2}f\|_{W_2^s(\mathcal{X})}^{d/(2s)}\|Q_\lambda^{-1/2}f\|_{L^2(\mathcal{X})}^{1-d/(2s)} \tag{E.56}$$

$$\leq C'C^{d/2s}\tau^{\frac{(s-d)d}{4s}}\|Q_\lambda^{-1/2}f\|_{\mathcal{G}_\eta}^{d/2s}\|Q_\lambda^{-1/2}f\|_{L^2(\mathcal{X})}^{1-d/(2s)}, \tag{E.57}$$

$$\leq C'C^{d/2s}\tau^{\frac{(s-d)d}{4s}}\lambda^{-d/4s}\|Q_\lambda^{-1/2}f\|_{L^2(\mathcal{X})}^{1-d/(2s)}. \tag{E.58}$$

Finally note that, since by the reproducing property $g(x) = \langle \psi_\eta(x), g \rangle_{\mathcal{G}_\eta}$ for any $g \in \mathcal{G}_\eta$, we have that for any $f \in \mathcal{G}_\eta$ such that $\|f\|_{\mathcal{G}_\eta} \leq 1$, we have

$$\|Q_\lambda^{1/2}f\|_{L^2(\mathcal{X})}^2 = \int_{\mathcal{X}} (Q_\lambda^{1/2}f)(x)^2 dx \tag{E.59}$$

$$= \int_{\mathcal{X}} \left\langle Q_\lambda^{1/2}f, (\psi_\eta(x)\psi_\eta(x)^\top Q_\lambda^{1/2}f) \right\rangle_{\mathcal{G}_\eta} \tag{E.60}$$

$$= \int \left\langle f, Q_\lambda^{1/2}\left(\int_{\mathcal{X}} \psi_\eta(x)\psi_\eta(x)^\top dx\right) Q_\lambda^{1/2}f \right\rangle_{\mathcal{G}_\eta} \tag{E.61}$$

$$\leq \|f\|^2\|Q_\lambda^{-1/2}QQ_\lambda^{-1/2}\| \leq 1. \tag{E.62}$$

Then, to recap

$$\text{ess-sup}\|\zeta\|_{\mathcal{G}_\eta} \leq \sup_{\|f\|_{\mathcal{G}_\eta} \leq 1}\|Q_\lambda^{-1/2}f\|_{L^\infty(\mathcal{X})} \leq \sup_{\|f\|_{\mathcal{G}_\eta} \leq 1}\|Q_\lambda^{-1/2}f\|_{W_2^s(\mathcal{X})}^{d/(2s)}\|Q_\lambda^{-1/2}f\|_{L^2(\mathcal{X})}^{1-d/(2s)} \tag{E.63}$$

$$\leq C'C^{d/2s}\tau^{\frac{(s-d)d}{4s}}\lambda^{-d/4s}. \tag{E.64}$$

**Step 3. Bernstein inequality for random vectors in separable Hilbert spaces.** The points $x_1, \ldots, x_n$ are independently and identically distributed according to $p$. Define the random variables $\zeta_i = Q_\lambda^{-1/2}\phi_\eta(x_i)\mathbf{1}_{\mathcal{X}}(x_i)$ for $i = 1, \ldots, n$. Note that $\zeta_i$ are independent and identically distributed with the same distribution as $\zeta$. Now note that

$$\mathbb{E}\zeta_i = \mathbb{E}\zeta = Q_\lambda^{-1/2}\int_{\mathcal{X}} p(x)\psi_\eta(x)dx = Q_\lambda^{-1/2}v, \tag{E.65}$$

moreover $\frac{1}{n}\sum_{i=1}^n \zeta_i = Q_\lambda^{-1/2}\hat{v}$. Then, given the bounds on the variance and on the norm for $\zeta$, by applying a Bernstein inequality for random vectors in separable Hilbert spaces, as, e.g., Thm 3.3.4 of [41] (we will use the notation of Prop. 2 of [25]), the following holds with probability $1 - \delta$

$$\|Q_\lambda^{-1/2}(\hat{v} - v)\|_{\mathcal{G}_\eta} = \|\frac{1}{n}\sum_{i=1}^n \zeta_i - \mu\|_{\mathcal{G}_\eta} \leq \frac{M\log\frac{2}{\delta}}{n} + \sqrt{\frac{S\log\frac{2}{\delta}}{n}}, \tag{E.66}$$

with $M = C'C^{d/2s}\tau^{\frac{(s-d)d}{4s}}\lambda^{-d/4s}$ and $S = 2\text{Tr}(Q_\lambda^{-1}Q)$. $\qquad\square$

**Lemma E.5.** *Let $\eta = \tau \mathbf{1}_d$ with $\tau \geq 1$ and $\lambda \leq 1/2$. Let $\mathcal{X} \subset [-1,1]^d$. Then*

$$\mathrm{Tr}((Q + \lambda I)^{-1} Q) \leq C \tau^{d/2} \left(\log \tfrac{1}{\lambda}\right)^d, \tag{E.67}$$

*Proof.* Let $\sigma_j(Q)$ with $j \in \mathbb{N}$ be the sequence of singular values of $Q$ in non-increasing order. First note that for any $k \in \mathbb{N}$,

$$\sigma_k(Q) \leq \sum_{j \in \mathbb{N}} \sigma_j(Q) \leq \mathrm{Tr}(Q) = \mathrm{Tr}\left(\int_{\mathcal{X}} \psi_\eta(x) \psi_\eta(x)^\top dx\right) = \int_{\mathcal{X}} \|\psi_\eta(x)\|_{\mathcal{G}_\eta}^2 dx \leq 2^d, \quad \text{(E.68)}$$

since $\|\psi_\eta(x)\|_{\mathcal{G}_\eta}^2 = 1$ and $\mathcal{X} \subseteq [-1,1]^d$. Moreover, for $k \in \mathbb{N}$, let $\bar{x}_{1,k}, \dots, \bar{x}_{k,k}$ be a minimal covering of $[-1,1]^d$. Let $\tilde{P}_k$ be the projection operator whose range is $\mathrm{span}\{\bar{x}_{1,k}, \dots, \bar{x}_{k,k}\}$. Note that $\tilde{P}_k$ has rank $k$. By the Eckart-Young theorem, we have that $\sigma_{k+1}(Q) = \inf_{\mathrm{rank}(A)=k} \|Q - A\|_F^2$, then

$$\sigma_{k+1}(Q) = \inf_{\mathrm{rank}(A)=k} \|Q - A\|_F^2 \leq \|Q - \tilde{P}_k Q\|_F \tag{E.69}$$

$$\leq \left\| \int_{\mathcal{X}} (I - \tilde{P}_k) \psi_\eta(x) \psi_\eta(x) dx \right\|_F \leq \int_{\mathcal{X}} \|(I - \tilde{P}_k) \psi_\eta(x) \psi_\eta(x)\|_F dx \tag{E.70}$$

$$= \int_{\mathcal{X}} \|(I - \tilde{P}_k) \psi_\eta(x)\|_{\mathcal{G}_\eta} \|\psi_\eta(x)\|_{\mathcal{G}_\eta} dx \leq 2^d \sup_{x \in \mathcal{X}} \|(I - \tilde{P}_k) \psi_\eta(x)\|_{\mathcal{G}_\eta}. \tag{E.71}$$

Since $\bar{x}_{1,k}, \dots, \bar{x}_{k,k}$ is a minimal covering of $[-1,1]^d$ and since the $\ell_2$-ball of diameter 1 contains a cube of side $1/\sqrt{d}$, it is possible to cover a cube of side 2 with $k$ balls of diameter $2h$ (and so of radius $h$) with $h \leq 2\sqrt{d}/(k^{1/d} - 1)$, see, e.g., Thm. 5.3, page 76 of [8]. Since $\mathcal{G}_\eta$ is a reproducing kernel Hilbert space with Gaussian kernel $k_{2\eta}$ as discussed in (E.52), by applying Thm. C.3, we have that when $h \leq 1/(C' \sqrt{\tau})$ and

$$\sup_{x \in \mathcal{X}} \|(I - \tilde{P}_k) \psi_\eta(x)\|_{\mathcal{G}_\eta} \leq C e^{-\frac{c}{\sqrt{\tau} h} \log \frac{c}{\sqrt{\tau} h}} \leq C e^{-\frac{c(k^{1/d} - 1)}{2\sqrt{d}\sqrt{\tau}} \log \frac{c(k^{1/d}-1)}{2\sqrt{d}\sqrt{\tau}}}, \tag{E.72}$$

with $c, C, C'$ depending only on $d$. Take $k_\tau \geq (1 + 2\max(1, e/c, C')\sqrt{d\tau})^d$. When $k \geq k_\tau$, we have $h \leq 1/(C'\sqrt{\tau})$, $c/(h\sqrt{t}) \leq e$, and $k \geq 2^d$, so $(k^{1/d} - 1) \geq k^{1/d}/2$, then

$$\sup_{x \in \mathcal{X}} \|(I - \tilde{P}_k) \psi_\eta(x)\|_{\mathcal{G}_\eta} \leq C e^{-\frac{c_1 k^{1/d}}{\sqrt{\tau}}}, \qquad \forall k \geq k_\tau. \tag{E.73}$$

with $c_1 = c/(4\sqrt{d})$. Let $g(x) = C e^{-\frac{c_1}{\sqrt{\tau}} x^{1/d}}$ for $x \geq 0$. Since $g$ is non-increasing then $g(n + 1) \geq \int_n^{n+1} g(x) dx$. Since $\int_k^\infty e^{-\frac{c_1}{\sqrt{\tau}} x^{1/d}} \leq d(\frac{c_1}{\sqrt{\tau}})^{-d} \Gamma(d, k^{1/d} \frac{c_1}{\sqrt{\tau}})$ where $\Gamma(a, x)$ is the *incomplete Gamma function* and is bounded by $\Gamma(a, x) \leq 2x^a e^{-x}$ for any $x, a \geq 1$ and $x \geq 2a$ (see Lemma P, page 31 of [2]). The condition to apply the bound on the incomplete Gamma function in our case corresponds to require $k$ to satisfy $k \geq (2d/c_1)^d \tau^{d/2}$. Then, for any $k \in \mathbb{N}$, we have

$$\sum_{t > k+2} \sigma_t(Q) \leq 2^d \begin{cases} t & t \leq k_\tau' \\ dk e^{-\frac{c_1}{\sqrt{\tau}} k^{1/d}} & t > k_\tau', \end{cases} \tag{E.74}$$

where $k_\tau' = \max(k_\tau, 2 + (2d/c_1)^d \tau^{d/2})$. The bound on $\mathrm{Tr}((Q+\lambda I)^{-1}Q)$ is obtained by considering the characterization of the trace of an operator in terms of its singular values and the fact that $\frac{z}{z+\lambda} \leq \min(1, \frac{z}{\lambda})$. For any $k \geq k_\tau'$, we have

$$\mathrm{Tr}((Q + \lambda)^{-1} Q) = \sum_{t \in \mathbb{N}} \frac{\sigma_t(Q)}{\sigma_t(Q) + \lambda} = \sum_{t \leq (k+1)} \frac{\sigma_t(Q)}{\sigma_t(Q) + \lambda} + \sum_{t \geq k+2} \frac{\sigma_t(Q)}{\sigma_t(Q) + \lambda} \tag{E.75}$$

$$\leq 1 + k + \frac{1}{\lambda} \sum_{t \geq k+2} \sigma_t(Q) \leq 1 + k(1 + \tfrac{1}{\lambda}) e^{-\frac{c_1}{\sqrt{\tau}} k^{1/d}}. \tag{E.76}$$

In particular, choosing $\bar{k}_\tau = (\sqrt{\tau} C_2 \log(1 + \frac{1}{\lambda}))^d$ with $C_2 = 2d/c_1 + \max(1, e/c, C')\sqrt{d}$, then $\bar{k} \geq k_\tau'$ and so

$$\mathrm{Tr}((Q + \lambda)^{-1} Q) \leq 1 + (\sqrt{\tau} C_2 \log(1 + \tfrac{1}{\lambda}))^d \leq (\sqrt{\tau} 4 C_2 \log(\tfrac{1}{\lambda}))^d. \tag{E.77}$$

$\square$

## E.2 Proof of Thm. 7

We are finally ready to prove Thm. 7. We restate here the theorem for convenience.

**Theorem 7.** *Let $n, m \in \mathbb{N}, \lambda > 0, \eta \in \mathbb{R}^d_{++}$ and $p$ be a density satisfying Assumption 1. With the definitions above, let $\hat{p}_{n,m}$ be the model $\hat{p}_{n,m}(x) = f(x\,; \hat{A}, \tilde{X}, \eta)$, with $\hat{A}$ the minimizer of (12). Let $\eta = n^{\frac{2}{2\beta+d}}\,\mathbf{1}_d$ and $\lambda = n^{-\frac{2\beta+2d}{2\beta+d}}$. When $m \geq C'n^{\frac{d}{2\beta+d}}(\log n)^d \log(C''n(\log n))$, the following holds with probability at least $1 - \delta$,*

$$\|p - \hat{p}_{n,m}\|_{L^2(\mathcal{X})} \leq Cn^{-\frac{\beta}{2\beta+d}}(\log n)^{d/2}, \tag{13}$$

*where constant $C$ depends only on $\beta, d$ and $p$ and the constants $C', C''$ depend only on $\beta, d$.*

*Proof.* Let $\varepsilon \in (0,1], h > 0$ and $\eta = \tau\mathbf{1}_d$ with $\tau \in [1, \infty)$, to be fixed later. Denote by $\hat{p}$ the model associated to matrix $\hat{A} \in \mathbb{S}^m_+$ that minimizes $\hat{L}_\lambda$ over the set $\mathbb{S}^m_+$, i.e., $\hat{p} = f(\cdot\,; \hat{A}, \tilde{X}, \eta)$. The goal is then to bound $\|\hat{p} - p\|^{L^2(\mathbb{R}^d)}$. First we introduce the probabilities $p_\varepsilon, \tilde{p}_\varepsilon$, useful to perfom the analysis. Let $\mathsf{M}_\varepsilon \in \mathbb{S}_+(\mathcal{H}_\eta)$ be the operator such that $p_\varepsilon = f(\cdot\,; \mathsf{M}_\varepsilon, \phi_\eta)$ approximates $p$ with error $\varepsilon$ as defined in Thm. D.4, on the functions $\tilde{f}_1, \ldots, \tilde{f}_q \in W^\beta_2(\mathbb{R}^d) \cap L^\infty(\mathbb{R}^d)$ that are the extension to $\mathbb{R}^d$ of the functions $f_1, \ldots, f_q$ characterizing $p$ via Assumption 1. The details of the extension are in Cor. A.3. Now, consider the model the compressed model $\tilde{p}_\varepsilon = f(\cdot\,; \tilde{A}_\varepsilon, \tilde{X}_m, \eta)$ with

$$\tilde{A}_\varepsilon = K^{-1}_{\tilde{X},\tilde{X},\eta}\tilde{Z}\mathsf{M}_\varepsilon\tilde{Z}^* K^{-1}_{\tilde{X},\tilde{X},\eta}, \tag{E.78}$$

where $\tilde{Z}: \mathcal{H}_\eta \to \mathbb{R}^m$ is defined in (B.6) in terms of $\tilde{X}_m$.

**Step 1. Decomposition of the error.**
By applying Thm. E.2 with $A = \tilde{A}_\varepsilon$ and Lemma E.3 to simplify the notation, we derive

$$\|\hat{p} - p\|_{L^2(\mathcal{X})} \leq \|\tilde{p}_\varepsilon - p\|_{L^2(\mathcal{X})} + \sqrt{\lambda}\|\tilde{M}_\varepsilon\|_F + 5\|Q^{-\frac{1}{2}}_\lambda(\hat{v} - v)\|_{\mathcal{G}_\eta}, \tag{E.79}$$

where $Q_\lambda = Q + \lambda I$ and $\tilde{M}_\varepsilon = \tilde{Z}^*\tilde{A}_\varepsilon\tilde{Z}_\varepsilon$. Note that

$$\tilde{M}_\varepsilon = \tilde{Z}^*\tilde{A}_\varepsilon\tilde{Z}_\varepsilon = \tilde{Z}^* K^{-1}_{\tilde{X},\tilde{X},\eta}\tilde{Z}\mathsf{M}_\varepsilon\tilde{Z}^* K^{-1}_{\tilde{X},\tilde{X},\eta}\tilde{Z}_\varepsilon = \tilde{P}\mathsf{M}_\varepsilon\tilde{P}, \tag{E.80}$$

where $\tilde{P}: \mathcal{H}_\eta \to \mathcal{H}_\eta$ is defined in Appendix B and is the projection operator on the range of $\tilde{Z}^*$, so

$$\|\tilde{M}_\varepsilon\|_F = \|\tilde{P}\mathsf{M}_\varepsilon\tilde{P}\|_F \leq \|\tilde{P}\|^2\|\mathsf{M}_\varepsilon\|_F \leq \|\mathsf{M}_\varepsilon\|_F. \tag{E.81}$$

Bounding $\|\tilde{p}_\varepsilon - p\|_{L^2(\mathcal{X})}$ with $\|\tilde{p}_\varepsilon - p\|_{L^2(\mathcal{X})} \leq \|\tilde{p}_\varepsilon - p_\varepsilon\|_{L^2(\mathcal{X})} + \|p_\varepsilon - p\|_{L^2(\mathcal{X})}$, we obtain

$$\|\hat{p} - p\|_{L^2(\mathcal{X})} \leq \|\tilde{p}_\varepsilon - p_\varepsilon\|_{L^2(\mathcal{X})} + \|p_\varepsilon - p\|_{L^2(\mathcal{X})} + \sqrt{\lambda}\|\mathsf{M}_\varepsilon\|_F + 5\|Q^{-\frac{1}{2}}_\lambda(\hat{v} - v)\|_{\mathcal{G}_\eta}, \tag{E.82}$$

**Step 2. Bounding the terms of the decomposition.**
Let $h$ be the fill distance (defined in (C.1)) associated to the points $\tilde{x}_1, \ldots, \tilde{x}_m$. By Thm. C.4, there exist three constants $c, C, C'$ depending only on $d$ such that, when $h \leq \sigma/C'$, with $\sigma = \min(1, 1/\sqrt{\tau})$, then for any $x \in \mathcal{X}$

$$|\tilde{p}_\varepsilon(x) - p_\varepsilon(x)| \leq 2C\sqrt{\|\mathsf{M}_\varepsilon\|p_\varepsilon(x)}\,e^{-\frac{c\sigma}{h}\log\frac{c\sigma}{h}} + C^2\|\mathsf{M}_\varepsilon\|e^{-\frac{2c\sigma}{h}\log\frac{c\sigma}{h}}. \tag{E.83}$$

Since $q_\eta = 1, p_\varepsilon(x) = \phi_\eta(x)^\top M_\varepsilon\phi_\eta(x) \leq \|M_\varepsilon\|\|\phi_\eta(x)\|^2_{\mathcal{H}_\eta} = \|M_\varepsilon\|$ for any $x \in \mathbb{R}^d$, since $\|\phi_\eta(x)\|^2_{\mathcal{H}_\eta} = \phi_\eta(x)^\top\phi_\eta(x) = k_\eta(x,x) = 1$, then

$$\|\tilde{p}_\varepsilon - p_\varepsilon\|_{L^2(\mathcal{X})} \leq \text{vol}(\mathcal{X})(2C + C^2)\|\mathsf{M}_\varepsilon\|e^{-\frac{c\sigma}{h}\log\frac{c\sigma}{h}} \tag{E.84}$$

where $\text{vol}(\mathcal{X})$ is the volume of $\mathcal{X}$ and is $\text{vol}(\mathcal{X}) = 2^d$. By Thm. D.4 we know also that $\|p_\varepsilon - p\|_{L^2(\mathcal{X})} \leq \|p_\varepsilon - p\|_{L^2(\mathbb{R}^d)} \leq \varepsilon$. Moreover, we also know that there exists two constants $C_1, C_2$ depending on $\mathcal{X}, \beta, d$ and the norms of $\tilde{f}_1, \ldots, \tilde{f}_q$ (and so on the norms of $f_1, \ldots, f_q$ via Cor. A.3) such that

$$\|\mathsf{M}_\varepsilon\| \leq \|\mathsf{M}_\varepsilon\|_F \leq \text{Tr}(\mathsf{M}_\varepsilon) \leq C_1\tau^{d/2}(1 + \varepsilon^2\exp(\frac{C_2}{\tau}\varepsilon^{-\frac{2}{\beta}})) \leq 2C_1\tau^{d/2}\exp(\frac{C_2}{\tau}\varepsilon^{-\frac{2}{\beta}}) \tag{E.85}$$

By bounding $\|Q_\lambda^{-\frac{1}{2}}(\hat{v}-v)\|_{\mathcal{G}_\eta}$ via Lemma E.4, with $s=d$ and Lemma E.5, then (E.82) is bounded by

$$\|\hat{p}-p\|_{L^2(\mathcal{X})} \leq \varepsilon + C_3 \tau^{d/2} e^{\frac{C_2}{\tau}\varepsilon^{-\frac{2}{\beta}}}\left(\sqrt{\lambda}+e^{-\frac{c\sigma}{h}\log\frac{c\sigma}{h}}\right) \tag{E.86}$$

$$+ \frac{C_4 \log\frac{2}{\delta}}{n\lambda^{1/4}} + \left(\frac{C_5\tau^{d/2}(\log\frac{1}{\lambda})^d \log\frac{2}{\delta}}{n}\right)^{1/2} \tag{E.87}$$

$C_3 = 2C_1 \mathrm{vol}(\mathcal{X})(2C+C^2)$, $C_4 = 5C_4'$ where $C_4'$ is from Lemma E.4 and depends only on $d$, while $C_5 = 50C_5'$, where $C_5'$ is from Lemma E.5 and depends only on $d$. Setting $\varepsilon = n^{-\frac{\beta}{2\beta+d}}$, $\tau = \varepsilon^{-2/\beta}$ and $\lambda = \varepsilon^{2+2d/\beta} = n^{-(2\beta+2d)/(2\beta+d)}$, since $1/(n\lambda^{1/4}) = \varepsilon$ and $\varepsilon^{d/\beta}/n = \varepsilon^2$, then

$$\|\hat{p}-p\|_{L^2(\mathcal{X})} \leq (1+C_3 e^{C_2})\varepsilon + C_3 e^{C_2}\varepsilon^{d/\beta}e^{-\frac{c\sigma}{h}\log\frac{c\sigma}{h}} \tag{E.88}$$

$$+ C_4 \log\frac{2}{\delta}\varepsilon + C_5^{1/2}(\frac{2\beta+2d}{2\beta+d})^{d/2}\varepsilon(\log n)^{d/2}(\log\frac{2}{\delta})^{1/2}. \tag{E.89}$$

**Step 3. Controlling the number $m$ in terms of $h$.**

The final result is obtained by controlling the number of points $m$ such that $h \leq \frac{1}{C'\sqrt{\tau}}$ (in order to be able to apply Thm. C.4) and such that $h \leq \frac{c}{\sqrt{\tau}}\max(e,(1+\frac{d}{\beta})\log\frac{1}{\varepsilon})^{-1}$, so $\log\frac{c\sigma}{h} \geq 1$ and

$$\varepsilon^{d/\beta}e^{-\frac{c\sigma}{h}\log\frac{c\sigma}{h}} \leq \varepsilon^{d/\beta}e^{-\frac{c\sigma}{h}} \leq \varepsilon. \tag{E.90}$$

By, e.g. Lemma 12, page 19 of [37] and the fact that $[-1,1]^d$ is a convex set, we have that there exists two constants $C_6, C_7$ depending only on $d$ such that $h \leq C_6 m^{-1/d}(\log(C_7 m/\delta))^{1/d}$, with probability at least $1-\delta$. In particular, we want to find $m$ that satisfy

$$C_6 C_8^d \tau^{d/2}(\log 1/\varepsilon)^d m \geq \log\frac{C_7 m}{\delta}, \tag{E.91}$$

with $C_8 = \max(\frac{1}{c}, C', e, 1+d/\beta)$. With the same reasoning as in (C.42), we see that any $m$ satisfying $m \geq 2B\log(2AB)$ with $A = C_6 C_8^d \tau^{d/2}(\log 1/\varepsilon)^d$ and $B = C_7/\delta$ suffices to guarantee the inequality above. In particular, since $\varepsilon \leq n$ and $\varepsilon^{d/\beta} = n^{\frac{d}{2\beta+d}} \leq n^d$ we choose

$$m = C_9 n^{\frac{d}{2\beta+d}}(\log n)^d\, d\log(C_{10}^{1/d}n(\log n)). \tag{E.92}$$

with $C_9 = C_6 C_8^d$, $C_{10} = 2C_9 C_7$. With this choice, we have

$$\|\hat{p}-p\|_{L^2(\mathcal{X})} \leq (1+2C_3 e^{C_2})\varepsilon + C_4\varepsilon\log\frac{2}{\delta} + C_5^{\frac{1}{2}}(\frac{2\beta+2d}{2\beta+d})^{\frac{d}{2}}\varepsilon(\log n)^{\frac{d}{2}}(\log\frac{2}{\delta})^{\frac{1}{2}} \tag{E.93}$$

$$\leq (1+2C_3 e^{C_2}+C_4+C_5^{\frac{1}{2}}(\frac{2\beta+2d}{2\beta+d})^{\frac{d}{2}})\varepsilon\,(\log n)^{\frac{d}{2}}(\log\frac{2}{\delta}) \tag{E.94}$$

$$= C_{11}\, n^{-\frac{\beta}{2\beta+d}}\,(\log n)^{\frac{d}{2}}(\log\frac{2}{\delta}). \tag{E.95}$$

with $C_{11} = 1 + 2C_3 e^{C_2} + C_4 + C_5^{\frac{1}{2}}(\frac{2\beta+2d}{2\beta+d})^{\frac{d}{2}}$. $\qquad\square$

# F  Operations

We report here the derivation of the operations discussed in Sec. 2.1 for PSD models. For simplicity in the following, given a vector $x \in \mathbb{R}^d$ and a positive vector $\eta \in \mathbb{R}_{++}^n$, with some abuse of notation, in the following we will denote $\eta\|x\|^2$ for $x^\top \mathrm{diag}(\eta)x$ when clear from the context.

## F.1  Properties of the Gaussian Function

We recall the following classical properties of Gaussian functions, which are key to derive the results in the following. For any two points $x_1, x_2 \in \mathbb{R}^d$ and $\eta_1, \eta_2 \in \mathbb{R}_{++}^d$, with the notation introduced above, let $c_\eta = \pi^{d/2}\det(\mathrm{diag}(\eta))^{-1/2}$ and $x_3 = \frac{\eta_1 x_1 + \eta_2 x_2}{\eta_1+\eta_2}$ and $\eta_3 = \frac{\eta_1\eta_2}{\eta_1+\eta_2}$. We have

$$k_{\eta_1}(x,x_1)k_{\eta_2}(x,x_2) = k_{\eta_1+\eta_2}(x,x_3)k_{\eta_3}(x_1,x_2) \quad\text{and}\quad \int k_\eta(x,x_1)dx = c_\eta. \tag{F.1}$$

Additionally, we recall that the joint Gaussian kernel corresponds to the product kernels in the two variables, namely $k_{(\eta_1, \eta_2)}((x, y), (x', y')) = k_{\eta_1}(x, x') k_{\eta_2}(y, y')$.

We begin by recalling how the equalities in (F.1) can be derived. First, we recall that for any positive definite matrix $A \in \mathbb{S}_{++}^n$ (namely an invertible positive semi-definite matrix), the integral of the Gaussian function $e^{-\langle x, Ax \rangle}$ is

$$\int_{\mathbb{R}^d} e^{-\eta \|x\|^2} \, dx = \pi^{d/2} \det A^{-1/2}, \tag{F.2}$$

which yields the required equality in (F.1) for $A = \mathrm{diag}(\eta)$ and $\eta \in \mathbb{R}_{++}^d$.

For the second property in (F.1), let $x_1, x_2 \in \mathbb{R}^d$ and $\eta_1, \eta_2 \in \mathbb{R}_{++}^d$. For any $x \in \mathbb{R}^d$ we have

$$k_{\eta_1}(x, x_1) k_{\eta_2}(x, x_2) = e^{-\eta_1 \|x - x_1\|^2 - \eta_2 \|x - x_2\|^2}. \tag{F.3}$$

By expanding the argument in the exponent we have

$$\eta_1 \|x - x_1\|^2 + \eta_2 \|x - x_2\|^2 \tag{F.4}$$

$$= (\eta_1 + \eta_2) \|x\|^2 - 2 \langle x, \eta_1 x_2 + \eta_2 x_2 \rangle + \eta_1 \|x_1\|^2 + \eta_2 \|x_2\|^2 \tag{F.5}$$

$$= (\eta_1 + \eta_2) \left( \|x\|^2 - 2 \left\langle x, \frac{\eta_1 x_1 + \eta_2 x_2}{\eta_1 + \eta_2} \right\rangle \pm \left\| \frac{\eta_1 x_1 + \eta_2 x_2}{\eta_1 + \eta_2} \right\|^2 \right) \tag{F.6}$$

$$+ \eta_1 \|x_1\|^2 + \eta_2 \|x_2\|^2 \tag{F.7}$$

$$= (\eta_1 + \eta_2) \|x - x_3\|^2 + \eta_1 \|x_1\|^2 + \eta_2 \|x_2\|^2 - (\eta_1 + \eta_2) \|x_3\|^2, \tag{F.8}$$

where $x_3 = \frac{\eta_1 x_1 + \eta_2 x_2}{\eta_1 + \eta_2}$. Now,

$$\eta_1 \|x_1\|^2 + \eta_2 \|x_2\|^2 - (\eta_1 + \eta_2) \|x_3\|^2 \tag{F.9}$$

$$= \eta_1 \|x_1\|^2 + \eta_2 \|x_2\|^2 - \frac{\|\eta_1 x_1 + \eta_2 x_2\|^2}{\eta_1 + \eta_2} \tag{F.10}$$

$$= \eta_1 \|x_1\|^2 + \eta_2 \|x_2\|^2 - \frac{\eta_1^2 \|x_1\|^2 + 2\eta_1 \eta_2 \langle x_1, x_2 \rangle + \eta_2^2 \|x_2\|^2}{\eta_1 + \eta_2} \tag{F.11}$$

$$= \frac{\eta_1 \eta_2 \|x_1\|^2 - 2\eta_1 \eta_2 \langle x_1, x_2 \rangle + \eta_1 \eta_2 \|x_2\|^2}{\eta_1 + \eta_2} \tag{F.12}$$

$$= \frac{\eta_1 \eta_2}{\eta_1 + \eta_2} \|x_1 - x_2\|. \tag{F.13}$$

We conclude that, for $\eta_3 = \frac{\eta_1 \eta_2}{\eta_1 + \eta_2}$, we have

$$k_{\eta_1}(x, x_1) k_{\eta_2}(x, x_2) = e^{-\eta_1 \|x - x_1\|^2 - \eta_2 \|x - x_2\|^2} \tag{F.14}$$

$$= e^{-(\eta_1 + \eta_2) \|x - x_3\|^2 - \frac{\eta_1 \eta_2}{\eta_1 + \eta_2} \|x_1 - x_2\|^2} \tag{F.15}$$

$$= k_{(\eta_1 + \eta_2)}(x, x_3) k_{\eta_3}(x_1, x_2), \tag{F.16}$$

as required.

Finally, we recall that for any vectors $x, x' \in \mathbb{R}^{d_1}$ and $y, y' \in \mathbb{R}^{d_2}$ and positive weights $\eta_1 \in \mathbb{R}_{++}^{d_1}, \eta_2 \in \mathbb{R}_{++}$, we will make use of the fact that

$$k_{(\eta_1, \eta_2)}((x, y), (x', y')) = e^{-(x - x', y - y')^\top \mathrm{diag}(\eta_1, \eta_2)(x - x', y - y')} \tag{F.17}$$

$$= e^{-(x - x')^\top \mathrm{diag}(\eta_1) x - y^\top \mathrm{diag}(\eta_2) y'} \tag{F.18}$$

$$= k_{\eta_1}(x, x') k_{\eta_2}(y, y') \tag{F.19}$$

### F.2 Evaluation

We recall that, given a PSD model $f(x \,;\, A, X, \eta)$ evaluating it in a point $x_0$ writes as

$$f(x = x_0 \,;\, A, X, \eta) = K_{X, x_0, \eta}^\top A K_{X, x_0, \eta}. \tag{F.20}$$

Given a PSD model of the form $f(x, y\,;\, A, [X, Y], (\eta_1, \eta_2))$ we denote partial evaluation in a vector $y_0 \in \mathcal{Y}$ as

$$f(x, y = y_0\,;\, A, [X, Y], (\eta_1, \eta_2)) = \sum_{i,j=1}^{n} A_{i,j} k_{\eta_1}(x_i, x) k_{\eta_1}(x_j, x) k_{\eta_2}(y_i, y_0) k_{\eta_2}(y_j, y_0) \quad \text{(F.21)}$$

$$= \sum_{i,j=1}^{n} \left[ k_{\eta_2}(y_i, y_0) A_{ij} k_{\eta_2}(y_j, y_0) \right] k_{\eta_1}(x_i, x) k_{\eta_1}(x_j, x) \,\text{(F.22)}$$

$$= f(x\,;\, A \circ (K_{Y,y_0,\eta_2} K_{Y,y_0,\eta_2}^{\top}), X, \eta_1) \quad \text{(F.23)}$$

### F.3 Integration and Marginalization

We begin by showing the result characterizing the marginalization of a PSD model with respect to a number of random variables.

**Proposition 1** (Sum Rule – Marginalization). *Let $X \in \mathbb{R}^{n \times d}$, $Y \in \mathbb{R}^{n \times d'}$, $A \in \mathbb{S}_+(\mathbb{R}^n)$ and $\eta \in \mathbb{R}_{++}^d, \eta' \in \mathbb{R}_{++}^{d'}$. Then, the following integral is a PSD model*

$$\int f(x, y\,;\, A, [X, Y], (\eta, \eta'))\, dx = f(y\,;\, B, Y, \eta'), \qquad \text{with} \qquad B = c_{2\eta}\, A \circ K_{X,X,\frac{\eta}{2}}, \quad (5)$$

*Proof.* The result is obtained as follows

$$\int f(x, y\,;\, A, [X, Y], (\eta, \eta'))\, dy = \sum_{i,j=1}^{n} A_{ij} k_{\eta'}(y_i, y) k_{\eta'}(y_j, y) \int k_\eta(x_i, y) k_\eta(x_j, y)\, dx \quad \text{(F.24)}$$

$$= \sum_{i,j=1}^{n} A_{ij} k_{\frac{\eta}{2}}(x_i, x_j) k_{\eta'}(y_i, y) k_{\eta'}(y_j, y) \int k_{2\eta}(\tfrac{x_i + x_j}{2}, x)\, dx$$

$$\text{(F.25)}$$

$$= c_{2\eta} \sum_{i,j=1}^{n} \left[ A_{ij} k_{\frac{\eta_2}{2}}(x_i, x_j) \right] k_{\eta'}(y_i, y) k_{\eta'}(t_j, y) \quad \text{(F.26)}$$

$$= f(y\,;\, c_{2\eta}(A \circ K_{X,X,\frac{\eta}{2}}), Y, \eta') \quad \text{(F.27)}$$

$\square$

**Integration.** Analogously, we can write in matrix form the full integral of a PSD model (with respect to all its variables): let $f(x\,;\, A, X, \eta)$, we have

$$\int f(x\,;\, A, X, \eta)\, dx = \sum_{i,j=1}^{n} A_{ij} \int k_\eta(x_i, x) k_\eta(x_j, x)\, dx \quad \text{(F.28)}$$

$$= \sum_{i,j=1}^{n} A_{ij} k_{\frac{\eta}{2}}(x_i, x_j) \int k_{2\eta}(\tfrac{x_i + x_j}{2}, x)\, dx \quad \text{(F.29)}$$

$$= c_{2\eta} \sum_{i,j=1}^{n} A_{ij} k_{\frac{\eta}{2}}(x_i, x_j) \quad \text{(F.30)}$$

$$= c_{2\eta} \text{Tr}(A K_{X,X,\frac{\eta}{2}}), \quad \text{(F.31)}$$

which yields (4).

**Integration on the Hypercube.** In Remark 3 we commented upon restricting integration and marginalization on the hypercube $H = \prod_{t=1}^{d} [a_t, b_t]$. Both operations can be performed by slightly changing the integrals above. In particular, let $G \in \mathbb{R}^{n \times n}$ be the matrix with $i, j$-th entry equal to

$$G_{ij} = c_{2\eta} \prod_{t=1}^{d} \text{erf}\left( \sqrt{2\eta_t}\left(b_t - \frac{x_{i,t} + x_{j,t}}{2}\right) \right) - \text{erf}\left( \sqrt{2\eta_t}\left(a_t - \frac{x_{i,t} + x_{j,t}}{2}\right) \right) \quad \text{(F.32)}$$

Then, integration becomes

$$\int_H f(X; A, X, \eta\,; =)\mathrm{Tr}((A \circ K_{X,X,\frac{\eta}{2}})G) \tag{F.33}$$

and marginalization

$$\int f(x,y\,;\, A, [X,Y], (\eta, \eta'))\, dy = f(y\,;\, c_{2\eta}A \circ K_{X,X,\frac{\eta}{2}} \circ G.) \tag{F.34}$$

This result is a corollary of Prop. 9 that we prove below.

**Proposition 9.** *Let $p(x) = f(x\,;\, A, X, \eta)$ with $X \in \mathbb{R}^{n \times d}$, $A \in \mathbb{S}_+^n$, $\eta \in \mathbb{R}_{++}^d$. Let $g : \mathbb{R}^d \to \mathbb{R}$ and define $c_{g,\eta}(z) = \int g(x)e^{-\eta\|x-z\|^2}\, dx$ for any $z \in \mathbb{R}^d$. Then*

$$\mathbb{E}_{x \sim p}\, g(x) = \mathrm{Tr}((A \circ K_{X,X,\eta/2})\, G) \qquad \text{with} \qquad G_{ij} = c_{g,2\eta}\big(\tfrac{x_i+x_j}{2}\big). \tag{18}$$

*Proof.* Following the same proof for Prop. 1, we have

$$\int g(x)f(x\,;\, A, [X,Y], (\eta))\, dy = \sum_{i,j=1}^n A_{ij} \int g(x)k_\eta(x_i, y)k_\eta(x_j, y)\, dx \tag{F.35}$$

$$= \sum_{i,j=1}^n A_{ij}k_{\frac{\eta}{2}}(x_i, x_j) \int g(x)k_{2\eta}(\tfrac{x_i+x_j}{2}, x)\, dx \tag{F.36}$$

$$= \sum_{i,j=1}^n \Big[A_{ij}k_{\frac{\eta_2}{2}}(x_i, x_j)\Big]c_{g,2\eta}\big(\tfrac{x_i+x_j}{2}\big) \tag{F.37}$$

$$= \mathrm{Tr}((A \circ K_{X,X,\eta/2})G). \tag{F.38}$$

$\square$

## F.4 Multiplication

**Proposition 2** (Multiplication). *Let $X \in \mathbb{R}^{n \times d_1}$, $Y \in \mathbb{R}^{n \times d_2}$, $Y' \in \mathbb{R}^{m \times d_2}$, $Z \in \mathbb{R}^{m \times d_3}$, $A \in \mathbb{S}_+^n$, $B \in \mathbb{S}_+^m$ and $\eta_1 \in \mathbb{R}_{++}^{d_1}$, $\eta_2, \eta_2' \in \mathbb{R}_{++}^{d_2}$, $\eta_3 \in \mathbb{R}_{++}^{d_3}$. Then*

$$f(x,y\,;\, A, [X,Y], (\eta_1, \eta_2))f(y,z\,;\, B, [Y', Z], (\eta_2', \eta_3)) = f(x,y,z\,;\, C, W, \eta), \tag{6}$$

*is a PSD model, where $C = (A \otimes B) \circ \big(\mathrm{vec}(K_{Y,Y',\tilde\eta_2})\mathrm{vec}(K_{Y,Y',\tilde\eta_2})^\top\big)$, with $\tilde\eta_2 = \frac{\eta_2\eta_2'}{\eta_2+\eta_2'}$, base matrix $W = [X \otimes \mathbf{1}_m,\ \frac{\eta_2}{\eta_2+\eta_2'}Y \otimes \mathbf{1}_m + \frac{\eta_2'}{\eta_2+\eta_2'}\mathbf{1}_n \otimes Y',\ \mathbf{1}_n \otimes Z]$ and $\eta = \big(\eta_1, \eta_2 + \eta_2', \eta_3\big)$.*

*Proof.* We begin by explicitly writing the product between the two PSD models

$$f(x,y\,;\, A, [X,Y], (\eta_1, \eta_2))f(y,z\,;\, B, [Y', Z], (\eta_2', \eta_3)) \tag{F.39}$$

$$= \Big( \sum_{i,j=1}^n A_{ij}k_{\eta_1}(x_i, x)k_{\eta_1}(x_j, x)k_{\eta_2}(y_i, x)k_{\eta_2}(y_j, y) \Big) \tag{F.40}$$

$$\Big( \sum_{\ell,h=1}^n B_{\ell h}k_{\eta_2'}(y_\ell', y)k_{\eta_2'}(y_h', y)k_{\eta_3}(z_\ell, z)k_{\eta_3}(z_h, z) \Big) \tag{F.41}$$

$$= \sum_{i,j,\ell,h=1}^{n,m} \Big[A_{ij}B_{\ell h}k_{\tilde\eta_2}(y_i, y_\ell')k_{\tilde\eta_2}(y_j, y_h')\Big]k_{\eta_2+\eta_2'}\big(\tfrac{\eta_2 y_i+\eta_2' y_\ell'}{\eta_2+\eta_2'}, y\big)k_{\eta_2+\eta_2'}\big(\tfrac{\eta_2 y_j+\eta_2' y_h'}{\eta_2+\eta_2'}, y\big) \tag{F.42}$$

$$k_{\eta_1}(x_i, x)k_{\eta_1}(x_j, x)k_{\eta_3}(z_\ell, z)k_{\eta_3}(z_h, z), \tag{F.43}$$

where we have coupled together the pairs $(i, \ell)$ and $(j, h)$ using the product rule between Gaussian functions. Let

$$C = (A \otimes B) \circ (\mathrm{vec}(K_{Y,Y',\tilde\eta_2})\mathrm{vec}(K_{Y,Y',\tilde\eta_2})^\top), \tag{F.44}$$

and denote by $\mathfrak{i} : \mathbb{N} \times \mathbb{N} \to \mathbb{N}$ now the indexing function such that $\mathfrak{i}(i, \ell) = (i - 1)n + \ell$. It follows that the term

$$A_{ij}B_{\ell h}k_{\tilde{\eta}_2}(y_i, y'_\ell)k_{\tilde{\eta}_2}(y_j, y'_h) = C_{\mathfrak{i}(i,\ell)\mathfrak{i}(j,h)}, \tag{F.45}$$

corresponds to the $(\mathfrak{i}(i, \ell)\mathfrak{i}(j, h))$-th entry of the matrix $C$. Analogously, let:

- $\tilde{y}_{\mathfrak{i}(i,\ell)} = \frac{\eta_2 y_i + \eta'_2 y'_\ell}{\eta_2 + \eta'_2}$ is the $\mathfrak{i}(i, \ell)$-th row of the matrix $\widetilde{Y} = \frac{\eta_2}{\eta_2 + \eta'_2}Y \otimes \mathbf{1}_m + \frac{\eta'_2}{\eta_2 + \eta'_2}\mathbf{1}_n \otimes Y'$, which is the $nm \times d_2$ matrix whose rows correspond to all possible pairs from $Y$ and $Y'$ respectively.

- $\tilde{x}_{\mathfrak{i}(i,\ell)} = x_i$ is the $\mathfrak{i}(i, \ell)$-th row of the matrix $\widetilde{X} = X \otimes \mathbf{1}_m$, namely the $nm \times d_1$ matrix containing $m$ copies of each row of $X$.

- $\tilde{z}_{\mathfrak{i}(i,\ell)} = z_\ell$ is the $\mathfrak{i}(i, \ell)$-th row of the matrix $\widetilde{Z}\mathbf{1}_n \otimes Z$, namely the $nm \times d_3$ matrix containing $n$ copies of $Z$.

Then we have that

$$\sum_{i,j,\ell,h=1}^{n,m}\left[A_{ij}B_{\ell h}k_{\tilde{\eta}_2}(y_i, y'_\ell)k_{\tilde{\eta}_2}(y_j, y'_h)\right]k_{\eta_2+\eta'_2}\left(\frac{\eta_2 y_i + \eta'_2 y'_\ell}{\eta_2 + \eta'_2}, y\right)k_{\eta_2+\eta'_2}\left(\frac{\eta_2 y_j + \eta'_2 y'_h}{\eta_2 + \eta'_2}, y\right) \tag{F.46}$$

$$k_{\eta_1}(x_i, x)k_{\eta_1}(x_j, x)k_{\eta_3}(z_\ell, z)k_{\eta_3}(z_h, z) \tag{F.47}$$

$$= \sum_{i,j,\ell,h=1}^{n,m}C_{\mathfrak{i}(i,\ell)\mathfrak{i}(j,h)}k_{\eta_2+\eta'_2}(\tilde{y}_{\mathfrak{i}(i,\ell)}, y)k_{\eta_2+\eta'_2}(\tilde{y}_{\mathfrak{i}(j,h)}, y)k_{\eta_1}(\tilde{x}_{\mathfrak{i}(i,\ell)}, x)k_{\eta_1}(\tilde{x}_{\mathfrak{i}(j,h)}, x) \tag{F.48}$$

$$k_{\eta_3}(\tilde{z}_{\mathfrak{i}(i,\ell)}, x)k_{\eta_3}(\tilde{z}_{\mathfrak{i}(j,h)}, x) \tag{F.49}$$

$$= \sum_{s,t}^{nm}C_{st}k_{\eta_2+\eta'_2}(\tilde{y}_s, y)k_{\eta_2+\eta'_2}(\tilde{y}_t, y)k_{\eta_1}(\tilde{x}_s, x)k_{\eta_1}(\tilde{x}_t, x)k_{\eta_3}(\tilde{z}_s, x)k_{\eta_3}(\tilde{z}_t, x) \tag{F.50}$$

$$= f(x, y, z \, ; \, C, [\widetilde{X}, \widetilde{Y}, \widetilde{Z}], (\eta_1, \eta_2 + \eta'_2, \eta_3)), \tag{F.51}$$

as desired. $\qquad\square$

## F.5 Reduction

The reduction operation leverages the structure of the base matrix $X \otimes \mathbf{1}_m$ to simplify the PSD model. To this end, fenote by $\widetilde{X} = X \otimes \mathbf{1}_m$ and consider again the indexing function $\mathfrak{i}(i, \ell) = (i - 1)n + \ell$. Then (see also the proof of Prop. 2 we have that the $\mathfrak{i}(i, \ell)$-th row of $\widetilde{X}$ is $\tilde{x}_{\mathfrak{i}(i,\ell)} = x_i$, the $i$-th row of $X$. Therefore we have

$$f(x \, ; \, A, \widetilde{X}, \eta) = \sum_{s,t=1}^{nm}A_{st}k_\eta(\tilde{x}_s, x)k_\eta(\tilde{x}_t, x) \tag{F.52}$$

$$= \sum_{i,j,\ell,h=1}^{n,m}A_{\mathfrak{i}(i,\ell)\mathfrak{i}(j,h)}k_\eta(\tilde{x}_{\mathfrak{i}(i,\ell)}, x)k_\eta(\tilde{x}_{\mathfrak{i}(j,h)}, x) \tag{F.53}$$

$$= \sum_{i,j,\ell,h=1}^{n,m}A_{\mathfrak{i}(i,\ell)\mathfrak{i}(j,h)}k_\eta(x_i, x)k_\eta(x_j, x) \tag{F.54}$$

$$= \sum_{i,j=1}^{n}\left[\sum_{\ell,h=1}^{m}A_{\mathfrak{i}(i,\ell)\mathfrak{i}(j,h)}\right]k_\eta(x_i, x)k_\eta(x_j, x) \tag{F.55}$$

$$= \sum_{i,j=1}^{n}B_{ij}k_\eta(x_i, x)k_\eta(x_j, x), \tag{F.56}$$

where $B$ is a $n \times n$ PSD matrix and each of its entries is the sum of the entries of $A$ corresponding to the repeated rows in $X \otimes \mathbf{1}_m$. Therefore $B = (I_n \otimes \mathbf{1}_m^\top)A(I_n \otimes \mathbf{1}_m)$ as required.

We give here the explicit form for the Markov transition in Cor. 3

**Corollary 3** (Markov Transition). *Let $X \in \mathbb{R}^{n \times d_1}$, $Y \in \mathbb{R}^{n \times d_2}$, $Y' \in \mathbb{R}^{m \times d_2}$, $A \in \mathbb{S}_+^n$, $B \in \mathbb{S}_+^m$ and $\eta_1 \in \mathbb{R}_{++}^{d_1}$, $\eta_2, \eta_2' \in \mathbb{R}_{++}^{d_2}$. Then*

$$\int f(x, y \,;\, A, [X, Y], (\eta_1, \eta_2)) f(y \,;\, B, Y', \eta_2') \, dy = f(x \,;\, C, X, \eta_1), \tag{8}$$

*with $C \in \mathbb{S}_+^n$ obtained by applying in order, Prop. 2, Prop. 1 and reduction (7).*

*Proof.* We first multiply the two PSD model to obtain, by Prop. 2

$$f(x, y \,;\, A, [X, Y], (\eta_1, \eta_2)) f(y \,;\, B, Y', \eta_2') = f(x, y \,;\, A_1, [\widetilde{X}, \widetilde{Y}], (\eta, \eta_2 + \eta_2')), \tag{F.57}$$

with

$$A_1 = (A \otimes B) \circ (\text{vec}(K_{Y,Y',\tilde{\eta}_2}) \text{vec}(K_{Y,Y',\tilde{\eta}_2})^\top) \qquad \tilde{\eta}_2 = \frac{\eta_2 \eta_2'}{\eta_2 + \eta_2'}, \tag{F.58}$$

and $\tilde{X} = X \otimes \mathbf{1}_m$ and $\widetilde{Y} = \frac{\eta_2}{\eta_2 + \eta_2'} Y \otimes + \frac{\eta_2'}{\eta_2 + \eta_2'}$. Then, we proceed with marginalization. By Prop. 1, we have

$$\int f(x, y \,;\, A_1, [\widetilde{X}, \widetilde{Y}], (\eta, \eta_2 + \eta_2')) \, dy = f(x \,;\, A_2, \widetilde{X}, \eta), \tag{F.59}$$

where $A_2 = c_{2(\eta_2 + \eta_2')} A_1 \circ K_{\widetilde{Y}, \widetilde{Y}, (\eta_2 + \eta_2')/2}$. Finally, since $\widetilde{X} = X \otimes \mathbf{1}_m$, by reduction (7), we have

$$f(x \,;\, A_2, \widetilde{X}, \eta) = f(x \,;\, C, X, \eta) \tag{F.60}$$

with $C = (I_n \otimes \mathbf{1}_m^\top) A_2 (I_n \otimes \mathbf{1}_m)$, which concludes the derivation. $\qquad\square$

## F.6 Hidden Markov Models

We conclude this section by providing the derivation of the HMM inference in Sec. 4.3.

**Proposition 10** (PSD Hidden Markov Models (HMM)). *Let $X_0 \in \mathbb{R}^{n_0 \times d}$, $X_+, X \in \mathbb{R}^{n \times d}$, $X' \in \mathbb{R}^{m \times d}$, $Y \in \mathbb{R}^{m \times d'}$, $A_0 \in \mathbb{S}_+^{n_0}$, $A \in \mathbb{S}_+^n$, $B \in \mathbb{S}_+^m$ and $\eta_0, \eta, \eta', \eta_+ \in \mathbb{R}_{++}^d$, $\eta_{obs} \in \mathbb{R}_{++}^{d'}$. Let*

$$\hat{\tau}(x_+, x) = f(x_+, x \,;\, B, [X_+, X], (\eta_+, \eta)), \qquad \hat{\omega}(y, x) = f(y, x \,;\, C, [Y, X'], (\eta_{obs}, \eta')), \tag{21}$$

*be approximate transition and observation functions. Then, given the initial state probability $\hat{p}(x_0) = f(x_0 \,;\, A_0, X_0, \eta_0)$, for any $t \geq 1$, the estimate $\hat{p}$ in (20) is a PSD model of the form*

$$\hat{p}(x_t | y_{t:1}) = f(x_t \,;\, A_t, \tilde{X}, \eta' + \eta_+), \tag{22}$$

*where $\tilde{X} = \frac{\eta'}{\eta' + \eta_+} X' \otimes \mathbf{1}_n + \frac{\eta_+}{\eta' + \eta_+} \mathbf{1}_m \otimes X_+$ and $A_t$ is recursively obtained from $A_{t-1}$ as in Alg. 1.*

*Proof.* Let $f(x_t \,;\, A_{t-1}, X_{t-1}, \eta_{t-1})$ be the estimate $\hat{p}(x_{t-1} | y_{1:t-1})$ obtained at the previous step, with $A_{t-1} \in \mathbb{S}_+^{n_{t-1}}$ and $X_{t-1} \in \mathbb{R}^{n_{t-1}}$ We then proceed by performing the operations in (20).

*Observation $\hat{\omega}_t(x)$.* A new observation $y_t$ is received. By (3) we have

$$\hat{\omega}_t(x) = \hat{\omega}(y = y_t, x) = f(y = y_t, x_+ \,;\, C, [Y, X'], (\eta_{obs}, \eta')) = f(x \,;\, C_t, X', \eta'), \tag{F.61}$$

with

$$C_t = C \circ (K_{Y, y_t, \eta_{obs}} K_{Y, y_t, \eta_{obs}}^\top). \tag{F.62}$$

*Product $\hat{\beta}_t(x_+, x) = \hat{\tau}(x_+, x) \hat{p}(x | y_{1:t-1})$.* We perform the product between the transition function and the previous state estimation

$$\hat{\beta}(x_+, x) = f(x_+, x \,;\, B, [X_+, X], (\eta_+, \eta)) f(x \,;\, A_{t-1}, X_{t-1}, \eta_{t-1}) \tag{F.63}$$

$$= f(x_+, x \,;\, B_t, [X_+ \otimes \mathbf{1}_{n_{t-1}}, \widetilde{X}_t], (\eta_+, \eta + \eta_t)), \tag{F.64}$$

with

$$B_t = (B \otimes A_{t-1}) \circ (\text{vec}(K_{X, X_t, \tilde{\eta}_t}) \text{vec}(K_{X, X_t, \tilde{\eta}_t})^\top) \qquad \tilde{\eta}_t = \tilde{\eta} \frac{\eta \eta_t}{\eta + \eta_t}, \tag{F.65}$$

and

$$\widetilde{X}_t \frac{\eta}{\eta + \eta_t} X \otimes \mathbf{1}_{n_{t-1}} + \frac{\eta_t}{\eta + \eta_t} \mathbf{1}_n \otimes X_t. \tag{F.66}$$

*Marginalization (+ Reduction)* $\hat{\beta}_t(x_+) = \int \hat{\beta}_t(x_+ +, x) \, dx$. We perform marginalization by [Prop. 1](#) to obtain

$$\hat{\beta}_t(x_+) = \int \hat{\beta}_t(x_+, x) \, dx \tag{F.67}$$

$$= \int f(x_+, x \, ; \, B_t, [X_+ \otimes \mathbf{1}_{n_{t-1}}, \widetilde{X}_t], (\eta_+, \eta + \eta_t)) \, dx \tag{F.68}$$

$$= f(x_+ \, ; \, D_t', X_+ \otimes \mathbf{1}_{n_{t-1}}, \eta_+) \tag{F.69}$$

with

$$D_t' = c_{2(\eta + \eta_t)} B_t \circ K_{\widetilde{X}_t, \widetilde{X}_t, \tilde{\eta}_t/2}. \tag{F.70}$$

Since the PSD model has a redundant base point matrix, we can apply reduction from [(7)](#), to obtain

$$\hat{\beta}_t(x_+) = f(x_+ \, ; \, D_t', X_+ \otimes \mathbf{1}_{n_{t-1}}, \eta_+) \tag{F.71}$$

$$= f(x_+ \, ; \, D_t, X_+, \eta_+), \tag{F.72}$$

where

$$D_t = (I_n \otimes \mathbf{1}_{n_{t-1}}^\top) D_t' (I_n \otimes \mathbf{1}_{n_{t-1}}). \tag{F.73}$$

*Multiplication* $\hat{\pi}_t(x_+) = \hat{\omega}_t(x_+) \hat{\beta}_t(x_+)$. We now multiply the observation term with the state estimation to obtain

$$\hat{\pi}_t(x_+) = \hat{\omega}_t(x_+) \hat{\beta}_t(x_+) \tag{F.74}$$

$$= f(x_+ \, ; \, C_t, X', \eta') f(x_+ \, ; \, D_t, X_+, \eta_+) \tag{F.75}$$

$$= f(x_+ \, ; \, E_t, \widetilde{X}, \eta' + \eta_+), \tag{F.76}$$

with

$$E_t = (C_t \otimes D_t) \circ (\text{vec}(K_{X', X, \tilde{\eta}'}) \text{vec}(K_{X', X, \tilde{\eta}'})^\top) \qquad \tilde{\eta}' = \frac{\eta' \eta_+}{\eta' + \eta_+} \tag{F.77}$$

and $\tilde{X} = \frac{\eta'}{\eta' + \eta_+} X' \otimes \mathbf{1}_n + \frac{\eta_+}{\eta' + \eta_+} \mathbf{1}_m \otimes X_+$.

*Normalization* $\hat{p}(x_t | y_{1:t}) = \hat{\pi}_t(x_+) / \int \hat{\pi}_t(x_+) \, dx_+$. We finally integrate $\hat{\pi}_t(x_+)$ in order to normalize it. By [(4)](#) we have

$$c_t = \int \hat{\pi}_t(x_+) \, dx_+ = \int f(x_+ \, ; \, E_t, \widetilde{X}, \eta' + \eta_+) \, dx_+ = c_{2(\eta' + \eta_+)} \text{Tr}(E_t K_{\tilde{X}, \tilde{X}, (\eta' + \eta_+)/2}), \tag{F.78}$$

and therefore

$$\hat{p}(x_t | y_{1:t}) = \frac{\hat{\pi}_t(x_+)}{\int \hat{\pi}_t(x_+) \, dx_+} = f(x_t \, ; \, A_t, \widetilde{X}, \eta' + \eta_+), \tag{F.79}$$

with

$$A_t = E_t / c_t. \tag{F.80}$$

This concludes the proof showing that, at every step, $\hat{p}(x_t | y_{1:t})$ has always same base point matrix $\tilde{X}$ and parameters $\eta' + \eta_+$. Note that the proof above also recovers explicitly the steps in [Alg. 1](#). $\quad\square$