# OpenReview forum: "PSD Representations for Effective Probability Models"
_NeurIPS.cc/2021/Conference — NeurIPS 2021 Poster_

### Official Review · Reviewer_ME1H · 2021-07-16

**Rating:** 7
**Confidence:** 3

**Summary:**

The authors propose using the class of PSD functions in probabilistic modeling. They show that this class of functions is closed under multiplication and marginalization, and that it allows for concise approximation of a wide range of densities.

**Limitations And Societal Impact:**

Yes.

**Main Review:**

The paper is well-written. Even though I am not an expert in the topic, I could follow the exposition without major problems.

The experimental section of the paper is very limited, and seems to just serve as a light proof of concept for the ideas discussed in the paper.

That being said, the theoretical part of the paper is an interesting and relevant contribution to probabilistic modeling. The authors consider using the class of PSD functions [1] to represent probability distributions, in particular, to generalize mixture models. They first show that this class is closed under common operations, such as multiplication, marginalization, and conditioning. They also show that these operations can be performed fairly efficiently. Then, they discuss the representational power of this class, and characterize the number of "base points" required to get an $\epsilon$-approximation of a density. I think both parts are solid theoretical contributions, and the class of PSD densities seems like an interesting object to be investigated further, especially in practical applications.

I have one concern regarding the use of such models in applications that require repeated multiplication operations. Based on the discussion of section 2.2, the authors propose using a Nystrom projection after each multiplication, in order to avoid increasing the number of base points used in the representation. If this is to be done multiple times, could it potentially lead to a gradually accumulating approximation error?

&nbsp;
### References
[1] Ulysse Marteau-Ferey, Francis Bach, and Alessandro Rudi. Non-parametric models for non-negative functions. NeurIPS, 2020.

**Time Spent Reviewing:**

3

---

> ### Author Response · Authors · 2021-08-09
> **Reply to Reviewer ME1H**
>
> We are glad the reviewer found the paper easy to navigate. We thank the reviewer for considering our work to be a relevant and solid contribution for the community and to pointing out that our preliminary experiments offer promising evidence in support of further future empirical investigation.
>
> We thank the reviewer for the relevant question about the potential accumulating error of subsequent Nystrom projections. We have two comments on this:
>
> - As the reviewer pointed out, the subsequent projections do induce an accumulation of the approximation error. However this accumulation is only in terms of a multiplicative constant, but does not affect the rate of the approximation with respect to key parameters such as the number $m$ of base points.
>
> - Additionally, note that in many relevant applications, the approximation effect might accumulate more favorably. In particular, we are currently investigating how the approximation error propagates in the contexts of filtering for hidden Markov models (Sec 4.3). In these contexts, if the Markov transition kernel is contractive, it might be possible to show that the impact of past approximations is weighted by an exponentially decreasing factor depending on the number of steps, leading in the long run to a weighted cumulative approximation error corresponding to a truncated geometric series bounded by $\epsilon/(1-c)$ where $c < 1$ is the contraction constant of the Markov kernel.
> We will add the discussion above in our paper, to better clarify the impact of subsequent contraction-multiplications operations with PSD models.

---

### Official Review · Reviewer_zSeK · 2021-07-16

**Rating:** 7
**Confidence:** 2

**Summary:**

This paper is based on the positive semi-definite model from reference [15], and propose a positive semi-definite (PSD) models for probability distribution. The paper then focuses on the family of Gaussian PSD, which can be written as a sum of weighted pairwise multiplicative kernels over sample points. Most importantly, the weights do not need to be positive, and it only requires that the matrix of weights is a symmetric positive semidefinite matrix to ensure that the function is positive everywhere. The authors then show that the PSD model facilitates the sum and product rules; and as the number of sample grows large, one can use projection to compress the model. Further, the authors show the approximation capacity of the PSD model and demonstrate the usefulness of the PSD model on some simple tasks.

**Limitations And Societal Impact:**

not relevant

**Main Review:**

The theoretical contribution of this paper is solid, and I enjoy reading most parts of it. On the downside, the paper is rather dense: the appendix is 43 pages, which is rather too long for me to follow all the derivations. I believe that a journal publication may be a better fit for this paper. The paper also lacks strong empirical results to show that the PSD models outperforms other methods on decent ML-related tasks.

Some minor comments:

- Please comment on how can solve the estimation problem (12) to get \hat{A}.
- Using a dense matrix A is cumbersome when the sample size gets large. Can we use some sparse models of A, such as Toeplitz matrix or a connectivity matrix of a graph?


**Time Spent Reviewing:**

10

---

> ### Author Response · Authors · 2021-08-09
> **Reply to Reviewer zSeK**
>
> We are glad the reviewer appreciated reading our paper and we are convinced as well that our contribution will be of interest for the community. We believe that our preliminary experiments, albeit limited, offer promising evidence supporting the proposed framework, which will be investigated in the future from a more empirical perspective. We address the reviewer’s comment below:
>
> **About the estimation of the weight matrix $A$.** As mentioned in Line 212-214, equation (12) corresponds to a quadratic convex optimization problem with semidefinite constraints, for which a wide range of options are available to minimize it. A standard approach that is also reasonably efficient in practice is Frank-Wolfe on the Spectrahedron [a], that requires only to compute the first eigenvector of each iterate with low precision, e.g. via power method. In [b] there is a large-scale implementation of a similar idea where they run up to $n \times n$ matrices with $n = 10^7$. We will add the discussion above to expand the comments on the algorithms to estimate $\hat{A}$ in Section 3.
>
> **About leveraging structure on $A$ to lower the computational complexity of solving Equation (12).** We thank the reviewer for the very on point question about potential structures for the coefficient matrix A that could be exploited for improved efficiency. We have two comments on this:
>
> - Clearly, leveraging sparsity or using specific structured matrices for A can be very beneficial in terms of computational complexity. In particular, we are currently investigating the case where the data are structured, for example when the base points x_i are on a grid. In this setting it is possible to show that the resulting matrix can benefit from a Henkel-type form (similar to Toeplitz), drastically reducing the required memory and the computational cost.
>
> - Another viable structure that can be exploited is to constrain A to be low-rank, i.e. $A = BB^\top$, where B is a n x r matrix with r << n. Albeit non-convex, this is a well-established approach to (approximately) tackle semidefinite programming  and has been extensively studied in previous works for problems related to matrix completion [c, d]. Preliminary investigation has shown that this method might be also promising in the context of this work.
>
>
> **References**
>
> [a] Martin Jaggi. Revisiting Frank-Wolfe: Projection-Free Sparse Convex Optimization. 2013.
>
> [b] Yurtsever, Tropp, Fercoq, Udell, Cevher.  Scalable Semidefinite Programming.  2019 https://arxiv.org/pdf/1912.02949.pdf
>
> [c] Burer, Samuel, and Renato DC Monteiro. "A nonlinear programming algorithm for solving semidefinite programs via low-rank factorization." Mathematical Programming 95, no. 2 (2003): 329-357.
>
> [d] Waldspurger, Waters. Rank optimality for the Burer-Monteiro factorization. 2019. https://arxiv.org/pdf/1812.03046.pdf

---

### Official Review · Reviewer_JxqL · 2021-07-17

**Rating:** 7
**Confidence:** 3

**Summary:**

In this paper, the authors propose a positive semidefinite (PSD) parameterization to model probabilities, which generalizes mixture models by allowing negative weights. They demonstrate that the PSD parameterization enjoys the same computational advantages of mixture models and could approximate a wide class of target densities.

**Limitations And Societal Impact:**

- Interpretation of the weights in the PSD model is not very clear to me. Is there a correspondence between PSD coefficients and the membership weights in the mixture model?
- Does the parameter estimation consider the kernel parameter?


**Main Review:**

The paper is well-written and the key idea is clear. The proposed PSD parameterization is a family of nonnegative functions in the reproducing kernel Hilbert space generated by a given kernel. This provides a natural approximation to a rich class of target densities. The authors show that common operations such marginalization and integration can be easily performed in terms of the PSD model. In particular, the PSD model supports a compression operation that reduces the number of base points leveraging the Nystrom projection in reproducing kernel Hilbert space. In addition, the PSD model has the computational advantage that the exact sum and product rules can be performed efficiently. This is demonstrated in an application of the PSD parameterization to the hidden Markov model.

There are two sets of parameters in the PSD parameterization, the coefficient matrix as well as the kernel parameter. It is stated that a benefit of the PSD model involves possible negative weights, i.e., entries of the coefficient matrix. This is a bit confusing as the interpretation of the weights is different from the weights in the mixture model setting where the latter is typically interpreted as membership probabilities.

Another concern is in the parameter estimation. It seems that that only the coefficient matrix will be estimated. This could be done efficiently due to the linear structure. The kernel parameter may also need to be inferred, which could involve expensive non-convex optimization.


**Time Spent Reviewing:**

2

---

> ### Author Response · Authors · 2021-08-09
> **Reply to Reviewer JxqL**
>
> We are glad the reviewer found the paper well written and clear to follow. We address the reviewer’s comment below:
>
> **On the interpretation of model weights.** In our model, weights don’t have a direct probabilistic interpretation. However, this is not required to achieve strong theoretical and computational guarantees. In particular, we discuss in Sec. 3 that allowing negative weights is crucial to achieve substantially better approximation guarantees than models using only non-negative weights. In this work, we succeed in finding the delicate balance of allowing for negative weights (and so achieving optimal approximation), while still guaranteeing the resulting model to be non-negative everywhere (a fundamental condition to define a probability).
>
> **Does the parameter estimation consider the kernel parameter?** As the reviewer mentioned, finding the best kernel parameter is a non-convex optimization problem. In line with standard literature on kernel methods, we assume the bandwidth parameter $\eta$ to be a hyperparameter and to find it by means of cross-validation. In our formulation $\eta$ is a vector in $\mathbb{R}^d$. However, our results (see e.g. Theorem 7) show that already taking $\eta = 1_d * \eta_0$ is enough to achieve optimal approximation properties (here $1_d$ is the vector of all ones and $\eta_0$ is a real number). In this case, cross validation becomes significantly easier since we have to tune only the parameter $\eta_0$. This is still a non-convex problem, but a much simpler one, since it is in one dimension.  In other words, practically speaking, it is possible to also tune $\eta$ as a parameter together with the weight matrix, especially in this last form. We will add the discussion above to further clarify how to choose $\eta$, at the end of Section 3.

---

> > ### Comment · Reviewer_JxqL · 2021-08-29
> > **After rebuttal**
> >
> > Dear authors, thank you for your response. I've raised my score and am happy to recommend acceptance.

---

### Official Review · Reviewer_1NBG · 2021-07-19

**Rating:** 6
**Confidence:** 4

**Summary:**

The authors discuss an application of recent work in nonnegative function estimation to the problem of density estimation.  In particular, they argue that under mild assumptions, PSD models can concisely model a large range of interesting densities and that these densities can be (more or less) efficiently learned.  The work concludes with a qualitative comparison against existing density estimation strategies.

**Limitations And Societal Impact:**

Appropriate.

**Main Review:**

Originality:  The work builds heavily off of [15], but there is more than enough novelty in this specific case to justify a separate work.

Quality:  1) For strictly positive densities is there a KL-divergence analog of Thm. 6 (which is a more typical error metric for densities -- though it has issues of its own)?

2) How computationally efficient really is Proposition 8?  In particular, if n is large, there are O(n^2)) integrals that need to be estimated.  The answer may just be that it depends on g, but there doesn't really seem to be any mechanism proposed here to help reduce the complexity with a large n, is there?

3)  The experiments feel very preliminary compared to a typical NeurIPS submission.  While I think the approach has a lot of merit and would be of significant interest, the limited experimental results by themselves are enough for me not to vote for acceptance. In particular, there is no quantitative comparison and no experimental comparison on real data.

Clarity: 1) There are a significant number of typos that should be corrected via a thorough proofreading.
2) The notation is a bit cumbersome in places, e.g., x for the functional dependence and x_i for the rows of the matrix X.  Consider making some notational changes to improve readability.

Significance:  Efficient density estimation techniques are of general interest.

**Time Spent Reviewing:**

3 hours

---

> ### Author Response · Authors · 2021-08-09
> **Reply to Reviewer 1NBG**
>
> We thank the reviewer for pointing out that our paper is original and well-situated within the previous literature (in particular connected to [15]). We address the reviewer’s outstanding comments below and hope our argument will make the reviewer re-evaluate our paper:
>
> **About a KL-divergence analog of Thm. 6.** The reviewer’s question is relevant and interesting. We first note that, given the relation between metrics on probability measures, Thm. 6 implies also a bound on the excess risk with KL-divergence: since by hypotheses of Thm. 6 we are on a compact domain and the two probabilities are bounded from below, then, by the reverse Pinsker inequality, the KL-divergence is bounded by the total variation, which is in turn bounded by the L2 distance [a]. Clearly, the resulting bound is possibly significantly less tight than what could be achieved by directly studying the KL-divergence loss (perhaps by introduced also a more suitable novel estimator for it). However this question is beyond the scope of this work since it poses significantly different challenges and calls for a different set of analytic tools to be introduced. We will add this remark in the discussion after Thm. 6.
>
> **About the computational efficiency of Prop. 8.** As the reviewer points out, there are two sources of complexity in Prop. 8. Firstly, the dependency on the possibly large number n of integrals to be computed and secondly the cost of computing each of such integrals of $g$ multiplied with a Gaussian function. We have two comments on this:
>
> - Regarding the first source of complexity, a viable approach to mitigate the dependency on a large $n$ is to perform a compression operation based on the Nystrom approximation (introduced in Section 2.2). This approach can greatly reduce the dimensionality of the model (and then consequently the computational cost of integrating it with g) by incurring in a small approximation error (as discussed in more depth in Appendix C).
> - Regarding the second source of complexity, we care to point out that the problem of integrating a function multiplied by a Gaussian has been thoroughly studied in the applied mathematics literature. In many interesting cases - depending on $g$ as the reviewer suggested - the integral can be obtained in closed form (typically using special functions that are commonly implemented in scientific libraries, like scipy). Examples of such $g$ include algebraic functions that are combinations of polynomials, absolute values, exponentials, logarithmic, trigonometric and hypergeometric functions. Additionally, for more general functions $g$ and for moderate dimensions, there exist a wide range of very efficient methods to integrate a smooth function in terms of a Gaussian, such the Gauss-Hermite quadrature type methods. More generally, a simple approach is to approximate the function g via an RBF network of Gaussians (in this case the resulting model is a linear combination of gaussians) and then integrating it corresponds to the sum of integrals between two Gaussian, that is known in closed form. We note that this is often the case where $g$ is not known a priori, but rather learned from data as part of a larger learning pipeline/problem. To speed up the computation we can compute the integrals only of gaussians whose means are closer than a given number of times the variance by incurring in only a small approximation error. We can use kd-trees for this purpose.
>
> We thank the reviewer for the interesting question. We will add the discussion above, under Proposition 8, to comment on the integration process.
>
> **About Experiments.** It would seem that the sole reason for the low score attributed by the reviewer to our paper is the lack of real world experiments. However, we argue that this should not be a disqualifying factor:
>
> - Contrary to the reviewer’s opinion, a good number of well ranked NeurIPS submissions that are oriented towards theory do not report real world experiments. This is natural since the NeurIPS community is very diverse and encompasses a wide range of research interests from more empirical investigations to purely theoretical inquiries. For example, many papers that were deemed worthy of an Oral or Spotlight presentation for NeurIPS 2019 or NeurIPS 2020 from tracks related to kernel methods, probability modeling and optimization (to which this paper is related) offer only simulations / toy experiments or none. For example, see e.g. [b,c,d,e] (a similar situation holds for other tracks, such as “Learning Theory”, in which the majority of papers has mostly proof of concept experiments in support of the theory). In particular, [15] - on which this paper is built upon - reports only simulations of similar nature to ours, yet it was given the spotlight at NeurIPS 2020.
>
> - While our paper clearly leans towards the theoretical end of the spectrum of NeurIPS submissions, we believe that our simulations and toy experiments offer promising evidence in support of our theoretical findings. In particular, we recall that the paper is equipped with simulations and toy experiments that provide a qualitative comparison with other existing density estimation strategies (see Section 5 and also Appendix G), as the reviewer notes in the Summary of this review.
> Finally we address the two points related to clarity:
> We thank the reviewer. We will amend the paper to address all typos.
> We note that the notation is coherent and follows naturally: x and x_i have the same semantic meaning, they are points belonging to the space $\cal{X}$. This is in line with standard machine learning settings, where, for example, a function f(x) is learned from the input points x_i with i=1,...,n, contained in the training set. We will make this more clear at the beginning of Section 2.
>
>
> **References**
>
> [a] Sason. On reverse Pinsker’s inequalities. 2015 https://arxiv.org/pdf/1503.07118.pdf
>
> [b] Singh, Rahul, Maneesh Sahani, and Arthur Gretton. "Kernel instrumental variable regression." Neural Information Processing Systems, 2019. https://proceedings.neurips.cc/paper/2019/file/17b3c7061788dbe82de5abe9f6fe22b3-Paper.pdf
>
> [c] Shuxiao Chen, Edgar Dobriban, Jane H. Lee. "A Group-Theoretic Framework for
> Data Augmentation" Neural Information Processing Systems, 2020 https://proceedings.neurips.cc/paper/2020/file/f4573fc71c731d5c362f0d7860945b88-Paper.pdf
>
> [d] Bolte, Jerome, and Edouard Pauwels. "A mathematical model for automatic differentiation in machine learning." In Conference on Neural Information Processing Systems. 2020. https://proceedings.neurips.cc/paper/2020/file/7a674153c63cff1ad7f0e261c369ab2c-Paper.pdf
>
> [e] Gatmiry, K., Aliakbarpour, M. and Jegelka, S. "Testing Determinantal Point Processes." Advances in Neural Information Processing Systems 2020, https://proceedings.neurips.cc/paper/2020/file/964d1775b722eff11b8ecd9e9ed5bd9e-Paper.pdf

---

> > ### Comment · Reviewer_1NBG · 2021-09-01
> > **Weak experiments**
> >
> > I appreciate your comments.  I'd be happy to vote to accept theoretical papers without real-world experiments, and I have many times in the past -- it's not my first NeurIPS ;).  I guess my biggest problem with the experiments is that they felt like an afterthought to me -- like you just tossed them in to have something (judging by some of the other reviewers' comments they felt similarly).  As a result, they detract more from the paper than they add (in my opinion).  I don't disagree that they comport with your theoretical results, but I do like to see experiments on real data that provide some evidence of the practical utility.  Still, considering the paper as a whole, I may have been too critical in my first review.  If I would probably vote to accept this paper without this section, I guess that I could vote to accept it with this section.

---

### Decision · Program_Chairs · 2021-09-27

**Decision:**

Accept (Poster)

**Comment:**

This article considers using a positive semi-definite (PSD) construction for flexible probability models. The basic idea is to parameterize a probability density as $f(x; M, \phi) = \phi(x)^\top M \phi(x)$ where $x\mapsto\phi(x)$ is a mapping to a Hilbert space and $M$ is a positive semi-definite operator (Equation 1).  In particular, this provides a generalization of mixture models that allows negative mixture weights. The paper focuses on the special case of Gaussian PSD models, in which $\phi(x)$ is the feature map associated with the Gaussian kernel.

This class of PSD models has some attractive properties: it is closed under multiplication (product rule, see Prop 2) and marginalization over a subset of variables (sum rule, see Prop 1).  Further, it exhibits universal consistency: it can approximate any probability density arbitrarily well; see Prop 4.  This paper builds upon recent work by Marteau-Ferey et al (2020), which showed the properties above.

The paper extends this previous work by showing that PSD models obtain the optimal rate of convergence for density approximation (similar to kernel  density estimation with positive and negative weights); see Theorem 7.  This is in contrast with the slow, sub-optimal rate exhibited by kernel density estimation using non-negative weights only. Additionally, the paper applies the PSD framework to several nontrivial applications in decision theory, classification and regression, and hidden Markov models.

The paper is well-written and clear.  The reviewers and I found the ideas to be interesting and well-developed.  The PSD framework seems potentially very useful for nonparametric modeling.

In my view, the main limitations of the paper are as follows.

1) Novelty. Since many of the results are closely related to the previous work of Marteau-Ferey et al (2020), it would be helpful for readers if the paper could more clearly delineate the novel contributions of this work.  Nonetheless, the reviewers and I found the work to contain many innovative contributions.

2) Computational complexity.  The computational cost can be large, particularly under the multiplication rule.  The article discusses using Nystrom projections as a way of mitigating this, but this is an approximation that may hinder performance.

3) Limited experiments.  The experimental results shown are compelling, but the experiments section is quite brief.  More experimental investigation would help demonstrate the utility of the methodology.

Marteau-Ferey, U., Bach, F., & Rudi, A. (2020). *Non-parametric models for non-negative functions.* arXiv preprint arXiv:2007.03926.